# Broad-scale variation in human genetic diversity levels is predicted by purifying selection on coding and non-coding elements

David A Murphy[1,2]*, Eyal Elyashiv[1,3], Guy Amster[1,4], Guy Sella[1,5]*

[1]Department of Biological Sciences, Columbia University, New York, United States; [2]Genes and Human Disease Research Program, Oklahoma Medical Research Foundation, Oklahoma City, Oklahoma City, United States; [3]MyHeritage, Or Yehuda, Israel; [4]Flatiron Health Inc, New York, United States; [5]Program for Mathematical Genomics, Columbia University, New York, United States

**Abstract** Analyses of genetic variation in many taxa have established that neutral genetic diversity is shaped by natural selection at linked sites. Whether the mode of selection is primarily the fixation of strongly beneficial alleles (selective sweeps) or purifying selection on deleterious mutations (background selection) remains unknown, however. We address this question in humans by fitting a model of the joint effects of selective sweeps and background selection to autosomal polymorphism data from the 1000 Genomes Project. After controlling for variation in mutation rates along the genome, a model of background selection alone explains ~60% of the variance in diversity levels at the megabase scale. Adding the effects of selective sweeps driven by adaptive substitutions to the model does not improve the fit, and when both modes of selection are considered jointly, selective sweeps are estimated to have had little or no effect on linked neutral diversity. The regions under purifying selection are best predicted by phylogenetic conservation, with ~80% of the deleterious mutations affecting neutral diversity occurring in non-exonic regions. Thus, background selection is the dominant mode of linked selection in humans, with marked effects on diversity levels throughout autosomes.

*For correspondence:
david-murphy@omrf.org (DAM);
gs2747@columbia.edu (GS)

## Editor's evaluation

This paper uses state-of-the-art methods and the latest data to answer the question of whether variation in polymorphism levels along the human genome is mostly driven by linked purifying selection or selective sweeps. It makes a very strong case for the former. The paper is exceptionally well written and should be of interest to anyone wishing to understand patterns of polymorphism.

## Introduction

Selection at a given locus in the genome affects diversity levels at sites linked to it (*Hill and Robertson, 1966*; *Smith and Haigh, 1974*; *Kaplan et al., 1989*; *Begun and Aquadro, 1992*; *Charlesworth et al., 1993*; *Hudson and Kaplan, 1995*; *Nordborg et al., 1996*; *Charlesworth, 2013*; *Cutter and Payseur, 2013*). When a new, strongly beneficial mutation increases in frequency to fixation in the population, it carries with it the haplotype on which it arose, thus reducing levels of neutral diversity nearby, in what is sometimes called a 'hard selective sweep' (*Smith and Haigh, 1974*; *Kaplan et al., 1989*). 'Soft sweeps', particularly those in which an allele segregates at low frequency before becoming beneficial and sweeping to fixation, and 'partial sweeps', in which a beneficial mutation rapidly increases

to an intermediate frequency, also reduce neutral diversity levels near the selected sites (*Hermisson and Pennings, 2005*; *Przeworski et al., 2005*; *Pennings and Hermisson, 2006a*; *Pennings and Hermisson, 2006b*; *Coop and Ralph, 2012*; *Berg and Coop, 2015*). Similarly, when deleterious mutations are eliminated from the population by selection, so are the haplotypes on which they lie. This process too reduces diversity levels near selected sites, in a phenomenon known as 'background selection' (*Charlesworth et al., 1993*; *Hudson and Kaplan, 1995*; *Nordborg et al., 1996*; *Comeron and Kreitman, 2002*; *Good et al., 2014*; *Cvijović et al., 2018*). Because the lengths of the haplotypes associated with selected alleles depend on the recombination rate, selection causes a greater reduction in levels of linked neutral genetic diversity in regions with lower rates of recombination or a greater density of selected sites. These predicted relationships have been observed in numerous taxa, including plants, *Drosophila*, rodents, and primates, establishing that the effects of linked selection are widespread (*Begun and Aquadro, 1992*; *Nachman, 1997*; *Payseur and Nachman, 2002*; *Nordborg et al., 2005*; *Wright et al., 2006*; *Andolfatto, 2007*; *Begun et al., 2007*; *Macpherson et al., 2007*; *Wright and Andolfatto, 2008*; *Cai et al., 2009*; *Sella et al., 2009*; *Cutter and Payseur, 2013*).

More recently, the advent of large genomic datasets and detailed functional annotations have made it possible to infer the effects of linked selection and build maps that predict levels of diversity along the genome (*McVicker et al., 2009*; *Elyashiv et al., 2016*; also see *Hudson and Kaplan, 1995*; *Nordborg et al., 1996*; *Comeron, 2014*). The first effort predated the availability of genome-wide resequencing data, relying instead on information about incomplete lineage sorting among human, chimpanzee and gorilla, which reflects variation in diversity levels along the genome in the common ancestor of humans and chimpanzees (*McVicker et al., 2009*). This pioneering paper showed that a model of background selection fits variation in human-chimpanzee divergence levels along the genome remarkably well, with only a few parameters.

What remained unclear is whether this remarkable fit should be attributed to the effects of background selection alone. Notably, the estimate of the rate of deleterious mutations underlying the effects of background selection was unrealistically high—substantially greater than the upper limit based on estimates of the total mutation rate per site in humans (*Kong et al., 2012*; *Besenbacher et al., 2016*; Appendix 1 Section 5). In light of this finding, *McVicker et al., 2009* suggested that the model might be soaking up effects of other modes of selection, particularly those of selective sweeps (*McVicker et al., 2009*). Subsequent work indicated that selective sweeps had little effect on diversity levels in humans (*Coop et al., 2009*; *Hernandez et al., 2011*), however, with no more of a reduction in diversity around plausible targets of positive selection (nonsynonymous substitutions) than around sites assumed to be predominantly neutral (synonymous substitutions) (*Coop et al., 2009*; *Hernandez et al., 2011*). Yet, the interpretation of these findings was contested: it was suggested that on average, background selection causes more of a reduction in diversity around synonymous than nonsynonymous substitutions, and consequently that the comparison between the two types of sites may obscure the reduction due to sweeps around nonsynonymous substitutions (*Enard et al., 2014*). The map of predicted background selection effects offered little help in evaluating this hypothesis, because it provided poor quantitative fits of diversity levels around both synonymous and nonsynonymous substitutions (*Hernandez et al., 2011*). Thus, despite clear evidence for the impact of background selection, we still lack an understanding of its contribution relative to sweeps (*Stephan, 2010*), as well as maps of their respective effects on human diversity levels.

## Results and disussion
### Model and inference

Here we resolve these issues by considering the effects of background selection and selective sweeps on diversity levels jointly (*Figure 1* and Appendix 1 Section 1). We model the effects on the expected neutral heterozygosity (i.e., the probability of observing different alleles in a sample size of two) at a given autosomal position $x$, as

$$\pi(x) = \frac{2u(x)}{2u(x) + 1/(2N_e B(x)) + S(x)},$$

where $u(x)$ is the local mutation rate, $N_e$ is the effective population size without linked selection, $B(x)$ is the local (multiplicative) reduction in effective population size due to background selection, and $S(x)$ is the local coalescence rate caused by selective sweeps (*Wiehe and Stephan, 1993*; *Elyashiv et al.,*

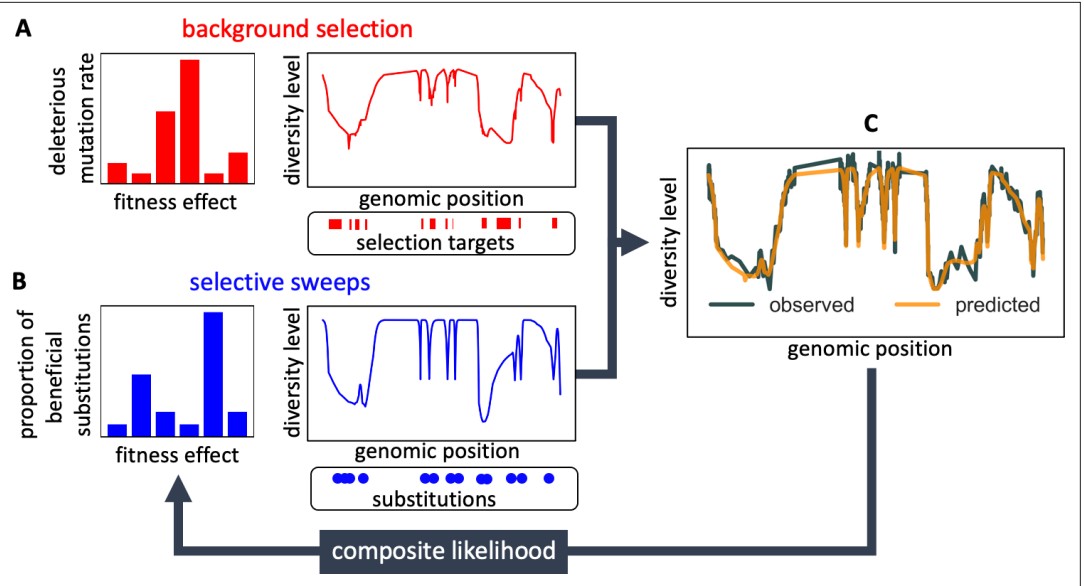

**Figure 1.** Modeling and inferring the effects of linked selection in humans. Given the putative targets of selection and corresponding selection parameters (**A and B**), we calculate the expected neutral diversity levels along the genome (**C**). We infer the selection parameters by maximizing their composite likelihood given observed diversity levels (**C**). Based on these parameter estimates, we calculate a map of the expected effects of selection on linked diversity levels.

*2016*). This model can be understood by thinking about a pair of lineages backward in time and noting that, considering mutation vs. coalescence events, $\pi(x)$ is the probability that a mutation occurs (at a rate $2u(x)$ per generation) before the pair coalesces, owing either to genetic drift (at a rate $1/(2N_eB(x))$), which includes the effect of background selection, or to selective sweeps (at a rate $S(x)$) (*Hudson, 1990*).

We model the effects of background selection, $B(x)$, as a function of genetic distance from regions that may be under purifying selection (*Figure 1A*) following the theory developed by *Hudson and Kaplan, 1995* and *Nordborg et al., 1996*. In this model, the deleterious mutation rate per site and distribution of selection effects in a given type of region (e.g. exons) are parameters to be estimated (see Appendix 1 Section 1.1 for details). In turn, we model the effects of sweeps, $S(x)$, as a function of genetic distance from substitutions on the human lineage that may have been beneficial (*Figure 1B*), following *Barton, 1998* and *Gillespie, 2000*. Here, the fraction of substitutions of a given type (e.g. nonsynonymous) that were beneficial and their distribution of selection effects are parameters to be estimated (see Appendix 1 Section 1.1 for details). Importantly, our model should capture the effects of any kind of sweeps, be they hard, partial or soft, so long as they eventually resulted in a substitution and affected diversity levels nearby (see *Coop and Ralph, 2012* and SOM Section D in *Elyashiv et al., 2016*).

Given the positions of different types of putatively selected regions and substitutions, their corresponding selection parameters, and a fine-scale genetic map, the model allows us to calculate the marginal probability that any given neutral site in the genome is polymorphic in a sample (*Figure 1C*). Provided measurements of polymorphism at neutral positions throughout the genome, we combine information across sites and samples to calculate the composite likelihood of selection parameters, and find the parameter values that maximize this likelihood (*Figure 1*). In addition to parameter estimation, this approach yields a map of the expected neutral diversity levels along the genome (*Figure 1C*). The mathematical form of the model and of the algorithms used for inference are detailed in Appendix 1 Section 1.

To infer the effects of background selection and selective sweeps on human diversity levels, we analyze autosomal polymorphism data from 26 human populations, collected in Phase III of the 1000 Genomes Project (*Auton et al., 2015*). Here, we focus on data from 108 genomes sampled from the Yoruba population (YRI), but we get similar results for the other populations (Appendix 1 Sections 7 and 9). To estimate diversity levels at neutral sites, we focus on non-genic autosomal sites that are the

least conserved in a multiple sequence alignment of 25 supra-primates (see Appendix 1 Section 3.1). To account for variation in mutation rates among neutral sites, we use estimates of the relative mutation rate for contiguous, non-overlapping blocks of 6000 putatively neutral sites, obtained from substitution rates in an eight-primate phylogeny (see Appendix 1 Section 3.3). To minimize the confounding of recombination rate estimates and diversity levels, we use a high-resolution genetic map inferred from ancestry switches in African-Americans (*Hinch et al., 2011*), which is highly correlated with other maps (*Hinch et al., 2011*) but is less dependent on diversity levels.

## Background selection

We first focus on two of our best-fitting models of the effects of background selection (see below and Appendix 1 Section 4). In both cases, we take as putative targets of purifying selection the 6% of autosomal sites estimated as most likely to be under selective constraint. In one, we choose these sites using phastCons conservation scores obtained for a 99-vertebrate phylogeny that excludes humans (*Siepel et al., 2005*). In the other, we rely on Combined Annotation-Dependent Depletion (CADD) scores, which are based primarily on phylogenetic conservation (excluding humans) but also on information from functional genomic assays (*Kircher et al., 2014*; *Rentzsch et al., 2019*); to avoid circularity, we use scores that were generated without the *McVicker et al., 2009* B-map as input (see Appendix 1 Section 2.5).

From these models, we obtain a map of predicted diversity levels (accounting for variation in mutation rates), which we can then compare to observed data (*Figure 2A* and *Appendix 1—figure 24*). We generate these maps using out-of-sample predictions in non-overlapping, contiguous 2 Mb windows (which we note is substantially greater than the scale of linkage disequilibrium in human populations; *Wall and Pritchard, 2003*). Over-fitting has a negligible effect on our results (also see Appendix 1 Section 6.1 and *Appendix 1—figure 48*), as expected given that the model has few parameters and the large amount of data (7 fitted parameters in this case and 2580 Mb blocks of ~653M putatively neutral sites spread over ~2600 LD blocks; *Berisa and Pickrell, 2016*). As a measure of the precision of our predictions, we consider the variance in diversity levels explained in non-overlapping autosomal windows (*Figure 2B*). Our predictions explain a large proportion of the variance across spatial scales: at the 1 Mb scale, the predictions based on CADD scores account for 60% of the variance in diversity levels compared to 32% explained by previous work (*McVicker et al., 2009*; see Appendix 1 Section 4.6).

## Selective sweeps

Next, we examine whether incorporating selective sweeps alongside background selection improves our predictions. Our inference should be able to tease apart the effects of selective sweeps, primarily because their effects, unlike those of background selection, should be centered around the locations

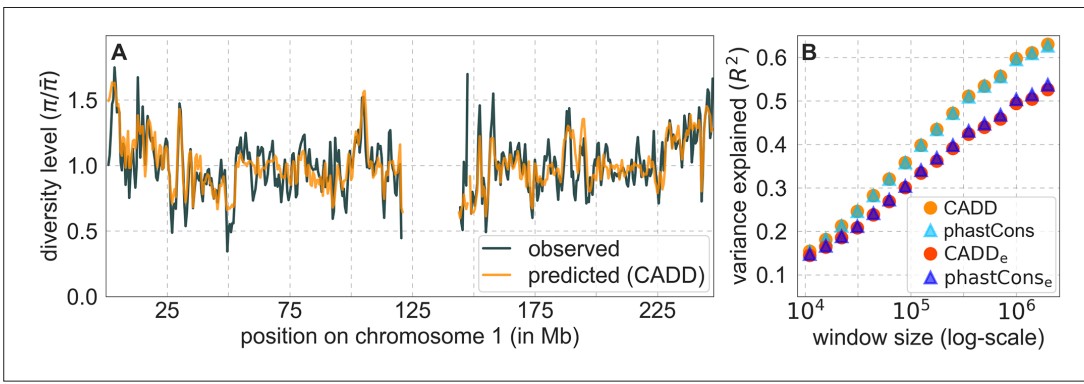

**Figure 2.** Comparison of diversity levels predicted by our best-fitting maps of background selection effects with observations. (**A**) Predicted and observed diversity levels along chromosome 1 in the YRI sample. Diversity levels are measured in 1 Mb windows, with a 0.5 Mb overlap, with the autosomal mean set to 1. (**B**) The proportion of variance in YRI diversity levels explained by background selection models at different spatial scales. Shown are the results for four choices of putative targets of selection: all sites with the highest 6% of CADD or phastCons scores (denoted CADD and phastCons, respectively) and the subset of these sites that are exonic (denoted CADD$_e$ and phastCons$_e$, respectively). The results shown for our best-fitting models (based on the 6% of sites with the highest CADD or phastCons scores) are based on out-of-sample predictions in non-overlapping, contiguous 2 Mb windows. See Appendix 1 Section 4 for similar graphs with other choices, and Appendix 1 Sections 7 and 9 for other populations.

of substitutions. Moreover, as noted, we expect to capture the effects of selective sweeps, be they hard, partial or soft (*Smith and Haigh, 1974*; *Kaplan et al., 1989*; *Hermisson and Pennings, 2005*; *Przeworski et al., 2005*; *Pennings and Hermisson, 2006a*; *Pennings and Hermisson, 2006b*; *Coop and Ralph, 2012*; *Berg and Coop, 2015*), so long as they resulted in substitutions and substantially affected diversity levels (see *Coop and Ralph, 2012* and SOM Section D in *Elyashiv et al., 2016*). Indeed, previous work that applied a similar methodology to data from *Drosophila melanogaster* was able to identify distinct effects of background selection and sweeps (*Elyashiv et al., 2016*). To examine whether we can identify such effects in humans, we consider several choices of putatively selected substitutions along the human lineage, including any nonsynonymous substitutions or any nonsynonymous and non-coding substitutions in constrained regions, allowing each type to have its own selection parameters and considering different measures of constraint (see Appendix 1 Section 4.5). Regardless of the types of substitutions considered, incorporating sweeps does not improve our fit. In fact, in all cases, our estimates of the proportion of substitutions resulting in sweeps with discernable effects on neutral diversity is approximately 0.

Moreover, in contrast to previous attempts (*McVicker et al., 2009*; *Hernandez et al., 2011*), our model of background selection alone provides good quantitative fits to the diversity levels observed around different genomic features and in particular around nonsynonymous and synonymous substitutions (*Figure 3* and *Appendix 1—figure 49*). Together, these results refute the hypothesis that reduced diversity levels around nonsynonymous substitutions in humans reflect 'masked' effects of selective sweeps (*Enard et al., 2014*); more generally, they indicate that selective sweeps resulting in substitutions had little effect on diversity levels in contemporary humans.

The lack of sweeps does not imply that adaptation was rare in recent human evolution, as instead, much of it may have been driven by selection on genetically complex traits, that is, traits with heritable variation arising from many segregating loci (*Coop et al., 2009*; *Pritchard et al., 2010*; *Pritchard and Di Rienzo, 2010*; *Hernandez et al., 2011*; *Sella and Barton, 2019*). Complex traits are often subject to ongoing stabilizing selection, that is, selection that acts to maintain traits near an optimal value (*Wright, 1935*; *Robertson, 1966*; *Walsh and Lynch, 2018*; *Sella and Barton, 2019*). Changes in selection pressures, that is, in optimal trait values, introduce transient directional selection on such complex traits. Under plausible conditions, we expect the adaptive response to directional selection to be highly polygenic, with phenotypic adaptation to new optima achieved rapidly, via tiny increases to the frequency of many alleles that change the traits in the direction favored by selection (*Hayward and Sella, 2019*). Over the long run, these tiny frequency changes cause a tiny excess of fixations of the alleles that were initially favored by selection (*Hayward and Sella, 2019*). Consequently, polygenic adaptation introduces only minor perturbations to allele trajectories compared to the case in which selection pressures on traits remain constant. In particular, the alleles that eventually fix do so extremely slowly, with trajectories that are predominated by weak selection and drift (*Hayward and Sella, 2019*), implying that their effects on linked diversity levels should be negligible (*Barton, 2000*; *Thornton, 2019*).

In contrast, ongoing stabilizing selection on complex traits could have a substantial effect on linked, neutral diversity levels (*Hayward and Sella, 2019*). Stabilizing selection induces purifying selection against minor alleles that affect complex traits (*Wright, 1931*; *Robertson, 1966*; *Simons et al., 2018*), and purifying selection on these alleles could be a

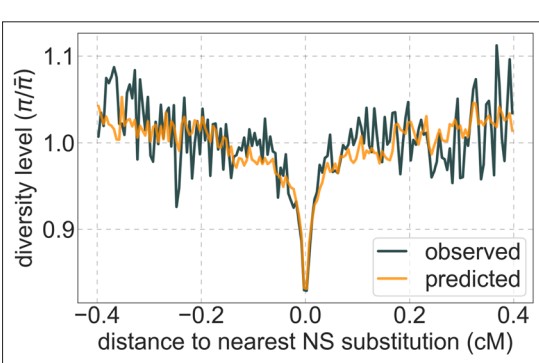

**Figure 3.** A background selection model predicts neutral diversity levels observed around human-specific nonsynonymous (NS) substitutions. Shown are the results for putatively neutral sites as a function of their genetic distance to the nearest nonsynonymous substitution (in 160 bins, each spanning 0.005 cM). For observed values, we average diversity levels within each bin. For predicted values, we average diversity levels predicted by our best-fitting CADD-based model (using the out-of-sample predictions in non-overlapping, contiguous, 2 Mb windows) and correct for relative mutation rate in each bin (using substitution data; see Appendix 1 Section 3.3). Both observed and predicted diversity levels are plotted relative to the autosomal mean. See *Appendix 1—figure 49* and *Appendix 1—figure 51* for similar graphs for other genomic features and using data from other populations.

major source of background selection (*Hayward and Sella, 2019*). In other words, if much of the selection in humans is driven by ongoing and changing selection pressures on complex traits, we may expect background selection to be the dominant mode of linked selection, as our results indicate.

## The source of background selection

Focusing then on models of background selection alone, we ask which genomic annotations appear to be the sources of purifying selection. Previous work found selection on non-exonic regions to contribute little, to the extent that removing conserved non-exonic sites from a model of background selection had little effect on predicted diversity levels (*McVicker et al., 2009*). In contrast, when we include only conserved exonic regions in our inference, our predictive ability is considerably diminished (*Figure 2B*).

Moreover, in models that include separate selection parameters for conserved exonic and non-exonic regions, purifying selection on non-exonic regions accounts for most of the reduction in linked neutral diversity (Appendix 1 Section 4.3). Our estimates suggest that ~80% of deleterious mutations affecting neutral diversity occur in non-exonic regions (e.g. in the model with the top 6% of phast-Cons scores, ~84% of selected sites and ~76% of deleterious mutations are non-exonic; with the top 6% of CADD scores, ~83% of selected sites and ~85% of deleterious mutations are non-exonic; see Appendix 1 Sections 4.3 and 4.6). Our estimates of the average strength of selection differ between exonic and non-exonic regions, but because the total reduction in diversity levels caused by background selection is fairly insensitive to the strength of selection (with the reduction being more localized for weakly selected mutations than for strong ones), the proportions of deleterious mutations that occur in these regions approximate their relative effects on neutral diversity levels (*Hudson, 1994*; see Appendix 1 Sections 4.3, 4.4, and 4.6). Thus, our estimates suggest that purifying selection on non-exonic regions accounts for ~80% of the reduction in linked neutral diversity. Moreover, including separate selection parameters for conserved exonic and non-exonic regions does not improve our predictions (Appendix 1 Section 4.3 and *Appendix 1—figure 19*).

Incorporating additional functional genomic information also does little to improve our predictions (Appendix 1 Sections 4.2 and 4.4). Notably, when we do not incorporate information on phylogenetic conservation, but include separate selection parameters for coding regions and for each of the Encyclopedia of DNA Elements (ENCODE) classes of candidate cis-regulatory elements (cCRE) (*Moore et al., 2020*), our predictive ability is considerably diminished (Appendix 1 Section 4.4). Moreover, using CADD scores (*Kircher et al., 2014*; *Rentzsch et al., 2019*), which augment information on phylogenetic conservation with functional genomic information, offers little improvement over relying on conservation alone (e.g., explaining 59.9% compared to 59.7% of the variance in diversity levels in 1 Mb windows, a difference that is not statistically significant; Appendix 1 Section 6). Thus, at present, functional annotations that do not incorporate phylogenetic conservation appear to provide poorer predictions of the effects of linked selection and those that do, offer little improvement over using conservation alone (see Appendix 1 Sections 4.1–4).

In turn, our predictions based on conservation are fairly insensitive to the phylogenetic depth of the alignments used to infer conservation levels, although we do slightly better using a 99-vertebrate alignment (excluding humans) compared to its monophyletic subsets (e.g. *Appendix 1—figure 14* and *Appendix 1—figure 33* and Appendix 1 Section 6.2). Our best-fitting models by a variety of metrics, are obtained using 5–7% of sites with the top CADD or phastCons scores as selection targets (*Appendix 1—figure 16* and *Appendix 1—figure 26*). This percentage is in good accordance with more direct estimates of the proportion of the human genome subject to functional constraint (*Ward and Kellis, 2012*; *Rands et al., 2014*).

## Estimates of the deleterious mutation rate

Reassuringly, the deleterious mutation rates that we estimate for our best-fitting models are plausible (*Figure 4*). Current estimates of the average mutation rate per site per generation in humans, including point mutations (*Kong et al., 2012*; *Besenbacher et al., 2016*), indels (*Besenbacher et al., 2016*), mobile element insertions (*Gardner et al., 2019*), and structural mutations (*Sudmant et al., 2015*; *Belyeu et al., 2021*) lie in the range of $1.29 \times 10^{-8} - 1.38 \times 10^{-8}$ per base pair per generation (Appendix 1 Section 5). Further accounting for the length of deletions (*Besenbacher et al., 2016*)—whereby a deletion that starts at a neutral site and includes selected sites should contribute to our estimate of the deleterious mutation rate, but deletions that affect one or several selected sites should have the same contribution—suggests that

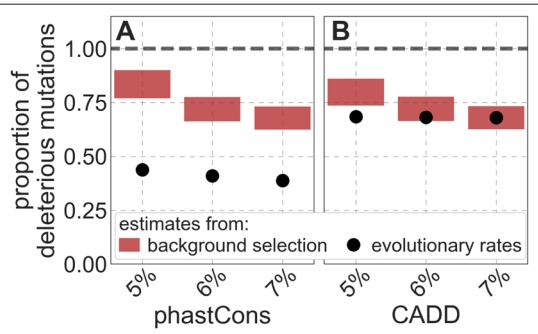

**Figure 4.** Estimates of the proportion of mutations at putatively selected sites that are deleterious. Shown are the results using 5–7% of sites with the highest phastCons scores (**A**) and CADD scores (**B**) as selection targets. For estimates based on fitting background selection models, we divide our estimates of the deleterious mutation rate per selected site by the estimate of the total mutation rate per site, where the ranges correspond to the range of estimates of the total rate, that is, $1.29 \times 10^{-8} - 1.51 \times 10^{-8}$ per base pair per generation (Appendix 1 Section 5.1). For estimates based on evolutionary rates (on the human lineage from the common ancestor of humans and chimpanzees), we take the ratio of the estimated rates at putatively selected sites and at matched sets of putatively neutral sites (see text and Appendix 1 Section 5.2 for details).

the upper bound on estimates of the deleterious mutations rate at putatively selected sites should fall in the range of $1.29 \times 10^{-8} - 1.51 \times 10^{-8}$ per base pair per generation (Appendix 1 Section 5). The estimates for all of our best-fitting models fall well below this bound (**Figure 4**). This is expected, because not every mutation at putatively selected sites will be deleterious: some sites are misclassified as constrained and some mutations at selected sites are selectively neutral.

To test whether our estimates of the proportion of mutations that are deleterious are plausible, we compare them with independent estimates based on the relative reduction in evolutionary rates at putatively selected vs. neutral sites along the human lineage (these sets of sites were identified from an alignment that excludes humans; Appendix 1 Sections 3.1, 4.1, and 4.4). The relative reduction allows us to estimate the proportion of deleterious mutations because deleterious mutations at selected sites rarely fix in the population whereas neutral mutations fix at a much higher rate, which is the same at selected and neutral sites (**Kimura and Crow, 1964**). In estimating the reduction at putatively selected sites, we matched the set of putatively neutral sites for the AT/GC ratio, and checked that our estimates were insensitive to the composition of other genomic features associated with mutation rates and with other non-selective processes that affect substitution rates (e.g., triplet context, methylated CpGs and recombination rates, which affect rates of biased gene conversion; Appendix 1 Section 5).

Our estimates based on evolutionary rates are closer to (and even overlap) those obtained from fitting models of background selection based on CADD scores compared to those based on phastCons scores (**Figure 4**). This is expected given that CADD scores are much better than phastCons scores at identifying constraint on a single site resolution (**Kircher et al., 2014**; **Rentzsch et al., 2019**), which markedly influences evolutionary rates at putatively selected sites (but not the predictions of background selection effects). We expect the two estimates to be similar but not identical, both because weak selection has a larger effect on evolutionary rates than on linked diversity levels (**McVean and Charlesworth, 2000**; **Comeron and Kreitman, 2002**; **Gordo et al., 2002**; **Charlesworth, 2013**; **Good et al., 2014**) and because estimates based on the effects of background selection may absorb the deleterious mutation rate at selected sites that were not included in our sets but are closely linked to sites in them (Appendix 1 Section 5). In summary, given the fit to data and plausible estimates of the deleterious rates, it is natural to interpret our maps as reflecting the effects of background selection, that is, as maps of $B$ (defined as the ratio of expected diversity levels with background selection, $\pi$, and in its absence, $\pi_0$; **Charlesworth et al., 1993**).

## Background selection on autosomes

Our maps are also well calibrated (**Figure 5**). When we stratify diversity levels at putatively neutral sites by our predictions, predicted and observed diversity levels are similar throughout nearly the entire range of predicted values (e.g. $R^2 = 0.96$ when sites are in predicted percentile bins). One exception is for ~5% of sites in which background selection is predicted to be the strongest (i.e. with the lowest $B$), where our predictions are imprecise. This behavior is due to a technical approximation we employ in fitting the models (see Appendix 1 Section 1.5). The other exception is for ~2% of sites in which background selection is predicted to be the weakest (i.e. with $B$ near 1), where observed diversity levels are markedly greater than expected. We observe similar behavior in all the human populations examined (**Appendix 1—figure 52**), and we cannot fully explain it by known mutational and recombination effects (e.g. of base composition and biased gene conversion; Appendix 1 Section

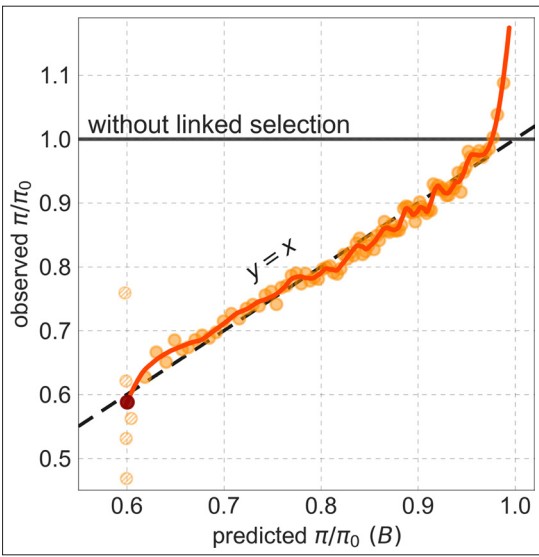

**Figure 5.** Observed vs. predicted neutral diversity levels across the autosomes. Shown are the results for the best-fitting CADD-based model, using the out-of-sample predictions in non-overlapping, contiguous 2 Mb windows. Light orange scatter plot: we divide putatively neutral sites into 100 equally sized bins based on the predicted $B$. For predicted values (x-axis), we average the predicted $B$ in each bin. For observed values (y-axis), we divide the average diversity level by the estimate of the average relative mutation rate (obtained from substitution data; see Appendix 1 Section 3.3) in each bin, and normalize by the autosomal average of $\pi_0$ (estimated from fitting the model; see Appendix 1 Section 1.1). Owing to a technical approximation (see Appendix 1 Section 1.5), our method forces the predictions for the 5 bins with the lowest predicted $B$ (open, hatched circles on the left) to be similar; we therefore also show the results for these bins grouped together (dark red circle). Dark orange curve: the LOESS fit for a similarly defined scatter plot but with 2000 rather than 100 bins (with span = 0.1). For similar graphs corresponding to other models and using data from other populations, see Appendix 1 Sections 4 and 9, respectively.

8). This behavior could reflect ancient introgression of archaic human DNA into ancestors of contemporary humans (Appendix 1 Section 8.3), indicated also in other population genetic signatures (*Wall and Hammer, 2006*; *Green et al., 2010*; *Reich et al., 2010*; *Sankararaman et al., 2014*; *Racimo et al., 2015*; *Steinrücken et al., 2018*). Such introgressed regions are expected to increase genetic diversity and persist the longest in regions with low functional density and high recombination, corresponding to weak background selection effects (*Sankararaman et al., 2014*; *Harris and Nielsen, 2016*; *Juric et al., 2016*; *Schumer et al., 2018*).

Setting these outlier regions aside, we can use the maps to characterize the distribution of background selection effects in human autosomes. We note that background selection effects that are not captured by our models would cause us to underestimate the range and extent of background selection effects (*Elyashiv et al., 2016*). We find that diversity levels throughout almost all of the autosomes are affected by background selection, with a ~37% reduction in the 10% most affected sites, a non-zero (~2.1%) reduction even in the 10% least affected (after excluding outliers in the top 2% of bins; see *Figure 5*), and a mean reduction of ~17%. These conclusions are robust across our best-fitting maps and populations (Appendix 1 Section 4 and *Appendix 1—figure 35* and *Appendix 1—figure 52*). An important implication is that our maps of the effects of background selection provide a more accurate null model than currently used for other population genetic inferences that rely on diversity levels, notably inferences about demographic history (*Schiffels and Durbin, 2014*; *Terhorst et al., 2017*; *Pouyet et al., 2018*).

## Conclusion

Our results indicate that background selection is the dominant mode of linked selection in human autosomes and the major determinant of neutral diversity levels on the Mb scale (after accounting for variation in mutation rates). They further reveal that background selection effects arise primarily from purifying selection at non-coding regions of the genome. Non-coding regions are known to exhibit substantial functional turnover on evolutionary timescales (*Ward and Kellis, 2012*; *Rands et al., 2014*), and yet we find phylogenetic conservation to be the best predictor of selected regions. Moreover, at present, augmenting measures of conservation with functional genomic information in humans offers little improvement. It therefore remains unclear how much our maps can still be improved. Even without these potential refinements, our findings demonstrate that a simple model of background selection, conceived three decades ago (*Charlesworth et al., 1993*), provides a reliable quantitative prediction of genetic diversity levels throughout human autosomes.

## Acknowledgements

We thank Molly Przeworski for helpful discussions throughout this work. We also thank Ipsita Agarwal, Eduardo Amorim, Peter Andolfatto, Iain Mathieson, Priya Moorjani, Itsik Pe'er, Joe Pickrell

and Jonathan Pritchard for helpful discussions. We thank Yun Song for sharing unpublished results, and Lusiné Nazaretyan, Philipp Rentzsch, Max Schubach and Martin Kircher from the Kircher lab for generating CADD scores that were tailored for our purposes. We thank Ipsita Agarwal, Peter Andolfatto, Jeff Ross-Ibarra, Magnus Nordborg, Jonathan Pritchard, Molly Przeworski and one anonymous reviewer for comments on the manuscript.

## Additional information

### Competing interests

Eyal Elyashiv: is affiliated with MyHeritage. The author has no financial interests to declare. Guy Amster: is affiliated with Flatiron Health Inc, The author has no financial interests to declare. The other authors declare that no competing interests exist.

### Funding

| Funder | Grant reference number | Author |
| --- | --- | --- |
| National Institutes of Health | GM115889 | Guy Sella |
| National Institutes of Health | T32GM008798 | David A Murphy |

The funders had no role in study design, data collection and interpretation, or the decision to submit the work for publication.

### Author contributions

David A Murphy, Conceptualization, Resources, Data curation, Software, Formal analysis, Funding acquisition, Validation, Investigation, Visualization, Methodology, Writing – original draft, Writing – review and editing; Eyal Elyashiv, Resources, Software; Guy Amster, Conceptualization, Formal analysis, Supervision, Investigation, Methodology, Writing – review and editing; Guy Sella, Conceptualization, Formal analysis, Supervision, Funding acquisition, Investigation, Methodology, Writing – original draft, Writing – review and editing

### Author ORCIDs

David A Murphy  http://orcid.org/0000-0002-0715-3355
Guy Amster  http://orcid.org/0000-0002-9108-5200
Guy Sella  http://orcid.org/0000-0002-5239-7930

### Decision letter and Author response

Decision letter https://doi.org/10.7554/eLife.76065.sa1
Author response https://doi.org/10.7554/eLife.76065.sa2

## Additional files

### Supplementary files

• Transparent reporting form

### Data availability

Shared data can be found at https://github.com/sellalab/HumanLinkedSelectionMaps (copy archived at swh:1:rev:c09a98ac4c82e7d1c9c5d1cc7c283b13dca76db4). This repository includes fully documented code for: downloading and processing public datasets used, running inferences, analyzing results, and generating all figures from the manuscript. This repository also includes B-maps for all "best-fitting" models described in the manuscript. Customized CADD scores with bStatistic removed are available on Data Dryad at https://doi.org/10.5061/dryad.n8pk0p2x0.

The following dataset was generated:

| Author(s) | Year | Dataset title | Dataset URL | Database and Identifier |
| --- | --- | --- | --- | --- |
| Murphy D, Elyashiv E, Amster G, Sella G | 2023 | CADD scores version 1.6 with bStatistic removed from inputs | https://doi.org/10.5061/dryad.n8pk0p2x0 | Dryad Digital Repository, 10.5061/dryad.n8pk0p2x0 |

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

## Appendix 1

# Methods and additional analyses

Table of Contents

## 1. Model and inference method

Here we detail the model and inference method used in this study. In Section 1.1, we describe our model for the effects of background selection and selective sweeps and our approach to inferring the parameters of these models. This section is adapted from Elyashiv and colleagues (*Elyashiv et al., 2016*), who applied a similar approach to data from *Drosophila melanogaster*; we reproduce it here for completeness. In Section 1.2, we describe how we calculate lookup tables for the effects of background selection and sweeps, which our inference relies upon. We introduce several changes to the methods used in previous studies (*McVicker et al., 2009*; *Elyashiv et al., 2016*), which allow us to better control the precision of maps of the effects of linked selection. In Section 1.3, we

describe how we represent neutral polymorphism data and maps of the effects of linked selection in our calculations in order to increase computational tractability. In Section 1.4, we describe the optimization algorithm that we use to find the selection parameters that maximize our models composite-likelihood, and we apply the optimization to simulated datasets in order to demonstrate its efficacy and robustness. In Section 1.5, we introduce a thresholding approach that contends with biases in our optimization that arise from model misspecification, and we investigate how this thresholding affects our inferences. Finally, in Section 1.6, we provide an overview of the software that we use for inference and for other key analyses in the paper. The software, its documentation, and maps of the effects of linked selection are available for download at (https://github.com/sellalab/HumanLinkedSelectionMaps; *Murphy, 2021*).

## 1.1 Model and inference problem

We model the effects of background selection and selective sweeps on neutral heterozygosity levels (i.e. the probability of observing different alleles in a sample size of two), $\pi$, at an autosomal position $x$. In a coalescent framework, the model takes the form

$$\pi(x) = \frac{2u(x)}{2u(x) + 1/(2N_eB(x)) + S(x)},$$ (1)

where $u(x)$ is the local mutation rate, $N_e$ is the effective population size without linked selection, $B(x)$ is the local (multiplicative) reduction in the effective population size due to background selection and $S(x)$ is the local coalescence rate caused by selective sweeps (*Wiehe and Stephan, 1993*; *Elyashiv et al., 2016*). This approximation can be derived by considering the probability that a mutation occurs (at a rate $2u(x)$ per generation) before the pair of lineages coalesces, owing either to genetic drift ($1/(2N_eB(x))$), which includes the effect of background selection, or to a selective sweep ($S(x)$). While we consider autosomes, the model can be extended to sex chromosomes with straightforward modifications.

The model for the effects of background selection, $B(x)$, follows *Hudson and Kaplan, 1995* and *Nordborg et al., 1996* (*Appendix 1—figure 1a*). We assume a set of distinct annotations $i_B = 1, \ldots I_B$ under purifying selection (e.g. conserved exonic and non-exonic regions) and positions in the genome

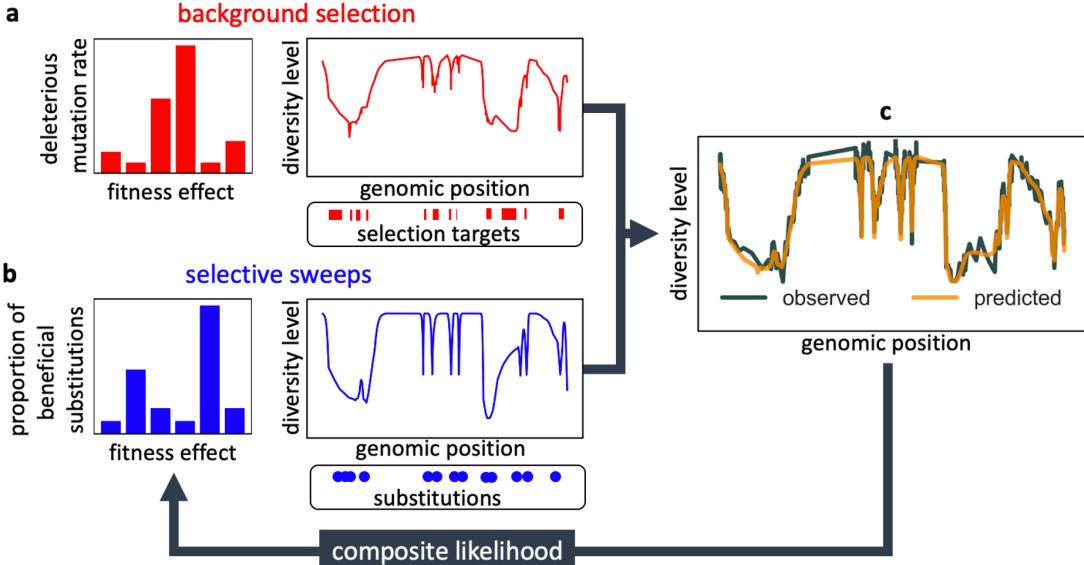

**Appendix 1—figure 1.** Modeling and inferring the effects of linked selection in humans. Given the targets of selection and corresponding selection parameters (**a and b**), we calculate the expected neutral diversity levels along the genome (**c**). We infer the selection parameters by maximizing their composite-likelihood given observed diversity levels (**c**). Based on these parameter estimates, we calculate a map of the expected effects of linked selection on diversity levels.

$A_B = \{a_B(i_B) | i_B = 1, \ldots, I_B\}$, where $a_B(i_B)$ denotes the set of genomic positions with annotation $i_B$. The selection parameters at these annotations are given by $\Theta_B = \{(u_d(i_B), f(t|i_B)) | i_B = 1, \ldots, I_B\}$, where $u_d$ is the rate of deleterious mutations and $f(t)$ is the distribution of selection coefficients in heterozygotes for a deleterious mutation. The reduction in the effective population size is then

$$B(x|A_B, \Theta_B, R) = \text{Exp}\left(-\sum_{i_B} \sum_{y \in a_B(i_B)} \int \frac{u_d(i_B)}{t\left(1 + r(x,y)(1-t)/t\right)^2} f(t|i_B) dt\right),$$

(2)

where $R$ is the genetic map and $r(x, y)$ is the genetic distance between the focal position $x$ and positions $y$ (only positions on the same chromosome are considered). The integrand reflects the effect that a site under purifying selection at position $y$ exerts on a neutral site at position $x$. This expression and its combination across sites provide a good approximation to the effect of background selection so long as selection is sufficiently strong (i.e. when $2N_e t \gg 1$).

In turn, the model for the effect of selective sweeps follows from an approximation used by **Barton, 1998** and **Gillespie, 2000**, among others (**Appendix 1—figure 1b**). Similarly to the model for background selection, we assume a set of distinct annotations $i_S = 1, \ldots, I_S$ subject to sweeps, but here the specific positions at which substitutions have occurred are known, $A_S = \{a_S(i_S) | i_S = 1, \ldots, I_S\}$ with $a_S(i_S)$ denoting the set of substitution positions with annotation $i_S$. The selection parameters at these annotations are $\Theta_S = \{(\alpha(i_S), g(s|i_S)) | i_S = 1, \ldots, I_S\}$, where $\alpha$ is the fraction of substitutions that are beneficial and $g(s)$ is the distribution of their additive selection coefficients. For autosomes, the expected rate of coalescence per generations at position $x$ due to sweeps is then approximated by

$$S(x|A_S, \Theta_S, R, \bar{N}_e, T) = \frac{1}{T} \sum_{i_S} \alpha(i_S) \sum_{y \in a(i_S)} \int \text{Exp}\left(-r(x,y)\tau(s, \bar{N}_e)\right) g(s|i_S) ds,$$

(3)

where $T$ is the length of the lineage (in generations) over which substitutions occurred, the positions of substitutions $y$ are summed over the chromosome with the focal site, $\bar{N}_e$ is the average effective population size and $\tau(s, \bar{N}_e)$ is the expected time to fixation of a beneficial substitution with selection coefficient $s$ and given an effective population size $\bar{N}_e$. We use the diffusion approximation for the fixation time

$$\tau(s, N_e) = \frac{2\left(\ln(4N_e s) + \gamma - (4N_e s)^{-1}\right)}{s},$$

(4)

where $\gamma$ is the Euler constant (**Hermisson and Pennings, 2005**). This model relies on several simplifying assumptions and approximations. In particular, the term $1/T$ relies on an assumption of one substitution per site per lineage and neglects variation in the length of lineages across loci. In combining the effects over substitutions, we further assume that the timings of beneficial substitutions are independent and uniformly distributed along the lineage, and that they are infrequent enough such that we can ignore interference among them (**Kim and Stephan, 2003**). The exponent approximates the probability of coalescence of two samples due to a classic sweep with additive selection coefficient $s$ (where $2N_e s \gg 1$) in a panmictic population of constant effective size $\bar{N}_e$. (For the relationships between these expressions and other kinds of sweeps see SOM Section D in **Elyashiv et al., 2016**). In principle, we should use the local $N_e$ incorporating the effects of background selection but given the logarithmic dependence of **Equation (4)** on $N_e$, we simply use the average $\bar{N}_e$.

To infer the selection parameters $\Theta_B$ and $\Theta_S$, we use a composite-likelihood approach across sites and samples (**Hudson, 2001**; **Appendix 1—figure 1**). We denote the positions of neutral sites by $X$ and the set of samples by $I$. We then summarize the observations by a set of indicator variables across sites and all pairs of samples $O = \{O_{i,j}(x) | x \in X, i \neq j \in I\}$, where $O_{i,j}(x) = 1$ indicates that samples $i$ and $j$ ($j \neq i$) differ at position $x$ and $O_{i,j}(x) = 0$ indicates that they are the same. In these terms, the composite log-likelihood takes the form

$$\log(L) = \sum_{x \in X} \sum_{i \neq j \in I} \log \left( \Pr \left\{ O_{i,j}(x) | \Theta_B, \Theta_S \right\} \right), \tag{5}$$

where

$$\Pr\{O_{i,j}(x) | \Theta_B, \Theta_S\} = \begin{cases} \pi(x | \Theta_B, \Theta_S) & O_{i,j}(x) = 1 \\ 1 - \pi(x | \Theta_B, \Theta_S) & O_{i,j}(x) = 0 \end{cases}$$

Using composite-likelihood circumvents the complications of considering linkage disequilibrium (LD) and of coalescent models for larger sample sizes. Importantly, maximizing this composite-likelihood should yield unbiased point estimates (*Fearnhead, 2003*; *Wiuf, 2006*). Beyond losing the information in LD patterns and in the site frequency spectrum, the main cost of this approach is the difficulty in assessing uncertainty in parameter estimates (as standard asymptotic results do not apply). We therefore use other ways to assess the reliability of our inferences.

To make the composite-likelihood calculations (i.e. the calculation of $\pi(x | \Theta_B, \Theta_S)$) feasible genome-wide, we discretize the distribution of selection coefficients on a fixed grid. Given a grid of negative and positive selection coefficients, $t_g$ and $s_k$, $g = 1, \ldots G$ and $k = 1, \ldots K$, the distribution of selection coefficients for each annotation becomes a set of weights on this grid, $w(t_g | i_B)$ and $w(s_k | i_s)$. (In principle, the grid could also be annotation-specific.) For background selection, these weights reflect the rate of deleterious mutations with a given selection coefficient and their sum should therefore be bound by the maximal deleterious mutation rate per site. For sweeps, the weights reflect the fraction of beneficial substitutions with a given selection coefficient and their sum should be bound by 1. In these terms, the effect of background selection takes the form

$$B\left(x | \Theta_B\right) = \mathrm{Exp}\left(-\sum_{i_B} \sum_{g=1}^{G} w(t_g | i_B) b(x | t_g, i_B)\right), \tag{6}$$

where $\mathrm{Exp}\left(-b(x | t_g, i_B)\right)$ is the proportional reduction in the effective population size induced by having one deleterious mutation per generation per site with selection coefficient $t_g$ at all the positions in annotation $i_B$. By the same token, the effects of sweeps take the form

$$S\left(x | \Theta_S\right) = \frac{1}{T} \sum_{i_S} \sum_{k=1}^{K} w(s_k | i_S) s(x | s_k, i_S), \tag{7}$$

where $\frac{1}{T} s(x | s_k, i_S)$ is the probability of coalescence per generation induced by sweeps in annotation $i_S$, if all the substitutions in this annotation are beneficial with selection coefficient $s_k$. By using a grid, we can calculate a lookup table of $b(x | t_g, i_B)$ and $s(x | s_k, i_S)$ once and then use it repeatedly to calculate the likelihood of different sets of weights. Moreover, the interpretation of estimated distributions on a grid is arguably simpler than that of the continuous parametric distributions commonly used (e.g. gamma and exponential), which impose rigid interdependencies between the densities associated with different selection coefficients with little justification and while the data is only informative about a subset of the domain. In the next section, we describe additional simplifications in the calculation of $b(x | t_g, i_B)$ and $s(x | s_k, i_S)$.

Other parameters are estimated as follows. Consider *Equation (1)* rewritten as

$$\pi(x) = \frac{\pi_0 \cdot \left(u(x)/\bar{u}\right)}{\pi_0 \cdot \left(u(x)/\bar{u}\right) + 1/B(x) + 2N_e S\left(x; \overline{N}_e, T\right)}, \tag{8}$$

to clearly specify all the additional parameters required for inference. $\pi_0 \equiv 4N_e \bar{u}$ is (approximately) the average neutral heterozygosity, given the effective population size in the absence of linked selection and the average mutation rate per site ($\bar{u}$); $\pi_0$ is estimated through the likelihood maximization. The local variation in mutation rate $u(x)/\bar{u}$ is estimated based on substitution rates at putatively neutral sites in an eight-primate phylogeny (excluding humans) in nonoverlapping

windows, with a window size chosen to balance true variation in mutation rates and measurement error (see Section 3.3). Finally, $\bar{N}_e$ is estimated based on the average genome-wide heterozygosity at putatively neutral sites, after dividing out by a direct estimate of the spontaneous point mutation rate of $1.2 \times 10^{-8}$ per site per generation (**Kong et al., 2012**), and $T/2\bar{N}_e$ is estimated by $(\bar{K}/2)/\pi_0$, where $\bar{K}$ is the average number of point substitutions per putatively neutral site on the human lineage (see Section 2.7).

## 1.2 Calculating lookup tables

Here we describe how we calculate the lookup tables for

$$s(x|s_k, i_S) \equiv \sum_{y \in a(i_S)} \mathrm{Exp}\left(-r(x,y)\tau(s_k, \bar{N}_e)\right) \tag{9}$$

and

$$b(x|t_g, i_B) \equiv \sum_{y \in a_B(i_B)} \frac{1}{t_g\left(1 + r(x,y)(1 - t_g)/t_g\right)^2} \tag{10}$$

at all putatively neutral autosomal positions ($x$), given annotations ($i_B$ and $i_S$) and selection coefficients ($t_g$ and $s_k$). We focus on one annotation and selection coefficient at a time and therefore simplify the notation to $b(x)$ and $s(x)$, and omit the variables in $\tau$ and the subscripts of the selection coefficients. When we refer to accuracy in this section, we assume that there is no model misspecification (e.g., that putatively neutral sites are neutral, that sets of selected sites and selection parameter values are accurate, that genetic maps are accurate, etc.); once we control the accuracy in this sense, the main sources of error in our predictions will be due to model misspecification.

*Our general approach is to calculate $b(x)$ and $s(x)$ with high accuracy at a subset of positions and to use linear interpolation between them.* The distances between these positions are chosen such that maps built using the lookup tables maintain a preset level of accuracy $\epsilon$. Specifically, we require that our approximation $\tilde{s}$ and $\tilde{b}$ at any position $x$ satisfy

$$\left|\frac{\tilde{s}(x) - s(x)}{s(x)}\right| < \epsilon \text{ and } \left|\frac{\mathrm{Exp}\left(-u_M \cdot \tilde{b}(x)\right) - \mathrm{Exp}\left(-u_M \cdot b(x)\right)}{\mathrm{Exp}\left(-u_M \cdot b(x)\right)}\right| < \epsilon,$$

where $u_M$ is an upper bound on the deleterious mutation rate per site per generation. When these conditions are met one can show (based on **Equations 6; 7**) that the relative accuracy of $S$ and $B$, and consequently of the expected neutral diversity level $\pi$ (based on **Equation 1**), are also bound by $\epsilon$.

### Sweeps

Assume that we have calculated $s$ accurately at position $x$ and consider the distance $\Delta$ at which the relative change in $s$ is bound by $\epsilon$, i.e., where

$$\left|\frac{s(x + \Delta) - s(x)}{s(x + \Delta)}\right| \leq \epsilon. \tag{11}$$

From **Equation 11**, we find that

$$\begin{aligned}
\left|s(x + \Delta) - s(x)\right| &\leq \sum_y \left|\mathrm{Exp}\left(-r(x + \Delta, y) \cdot \tau\right) - \mathrm{Exp}\left(-r(x,y) \cdot \tau\right)\right| \\
&= \sum_y \mathrm{Exp}\left(-r(x + \Delta, y) \cdot \tau\right) \cdot \left|1 - \mathrm{Exp}\left((r(x + \Delta, y) - r(x,y)) \cdot \tau\right)\right| \\
&\approx \sum_y \mathrm{Exp}\left(-r(x + \Delta, y) \cdot \tau\right) \cdot \left|1 - \mathrm{Exp}\left(r(x + \Delta, x) \cdot \tau\right)\right| \\
&\approx s(x + \Delta) \cdot \left(r(x + \Delta, x) \cdot \tau\right),
\end{aligned}$$

where the approximations assume $r(x + \Delta, x) \ll 1$. Consequently, by solving for $\Delta$ such that

$$r(x + \Delta, x) = \epsilon/\tau \tag{12}$$

we assure that the relative accuracy between $x$ and $x + \Delta$ is bound by $\epsilon$. We therefore calculate $s$ at the selected set of positions on a chromosome beginning at one end and choosing our step sizes according to *Equation 12* until we reach the other end.

## Background selection

Our calculation for background selection is based on the algorithm developed by *McVicker et al., 2009* (their calc_bkgd program) with several important modifications (*Appendix 1—figure 2*). The problems that require these modifications are most pronounced for small selection coefficients, whose background selection effects are localized at short genetic distances from selected segments where they can be quite strong. First, McVicker et al. used an additional lookup table to integrate over the effects of background selection exerted by a contiguous selected segment (SI of *McVicker et al., 2009*). This lookup table had poor resolution for small selection coefficients at short genetic distances from selected segments, and we have increased the resolution accordingly to fix the problem. Second, the algorithm for choosing the step size $\Delta$ is designed to control the absolute error, such that

$$\left| \text{Exp}\left(-u_M \cdot \widetilde{b}(x)\right) - \text{Exp}\left(-u_M \cdot b(x)\right) \right| < \epsilon,$$

rather than the relative error (*Equation 13*), which results in large relative errors when background selection effects are the strongest (which is with small selection coefficients). Third, the choice of step size $\Delta$ is based on the local behavior of background selection at the previous position, and consequently it sometimes skips over selected segments largely ignoring their highly localized effects (which are due to small selection coefficients). We describe how we resolve the last two problems in turn.

Assume that we have calculated $b$ accurately at position $x$ and consider the distance $\Delta$ at which the relative change in $\text{Exp}(-u_M \cdot b)$ is bound by $\epsilon$ (see *Equation 13*), that is, where

$$\left| \frac{\text{Exp}\left(-u_M \cdot b(x+\Delta)\right) - \text{Exp}\left(-u_M \cdot b(x)\right)}{\text{Exp}\left(-u_M \cdot b(x+\Delta)\right)} \right| \leq \epsilon. \tag{13}$$

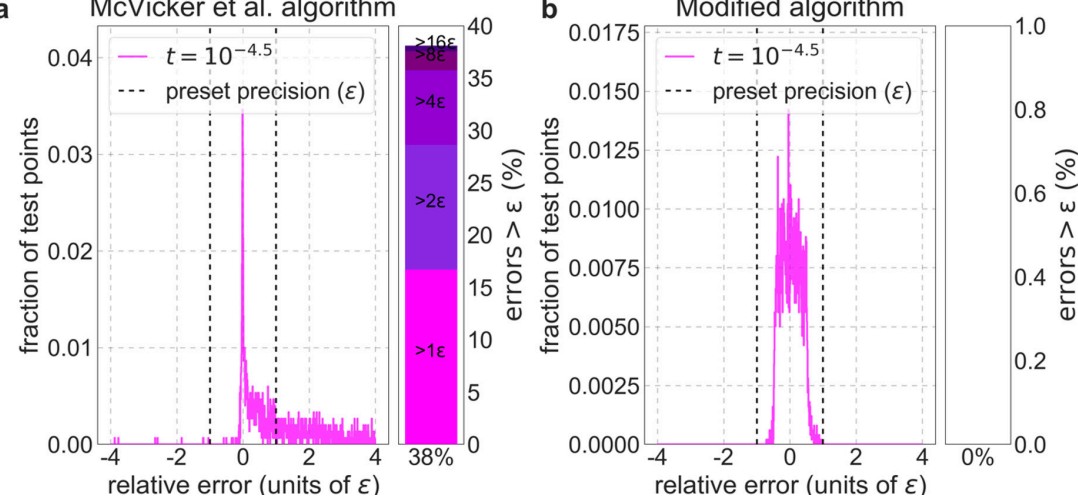

**Appendix 1—figure 2.** Distribution of relative errors in predictions before and after modifying calc_bkgd. We consider the model in which autosomal sites with the top 6% of CADD scores are chosen as selection targets, the deleterious mutation rate is $u_d = 7.4 \cdot 10^{-8}$ per bp per generation and the selection coefficient is the lowest in our grid ($t = 10^{-4.5}$), because this is the case most prone to errors (see text). We calculate $B$-values accurately (using *Equation 10*) at a million positions picked randomly from the 22 autosomes and use these values to calculate the relative errors based on the McVicker et al. algorithm (**a**) and on our modified algorithm (**b**). The side panel shows the proportion of sites in which the error exceeds $\epsilon$ (below), as well as its breakdown in multiples of $\epsilon$.

Rearranging the left-hand side, we find that

$$\left|1 - \mathrm{Exp}\left(-u_M \cdot \left(b(x) - b(x + \Delta)\right)\right)\right| \leq \epsilon,$$

and assuming that $\left|u_M \cdot \left(b(x) - b(x + \Delta)\right)\right| \ll 1$ we find that this requirement is well approximated by

$$\left|b(x + \Delta) - b(x)\right| \approx \left|b'(x) \cdot \Delta + b''(x) \cdot \Delta^2/2\right| \leq \epsilon/u_M.$$

As our putative step size, we therefore take the (smallest) solution of the quadratic

$$\left|b'(x) \cdot \Delta + b''(x) \cdot \Delta^2/2\right| = \epsilon/u_M. \tag{14}$$

As in the case of sweeps, we calculate $b$ at a selected set of positions on a chromosome, beginning on one end and choosing our step sizes in a way that maintains the preset relative accuracy $\epsilon$ until we reach the other end. Assuming that we have calculated $b$ accurately at position $x$, our algorithm for choosing the step size consists of the following steps:

1. If $x$ is at the end of the chromosome, stop.
2. Calculate a candidate step size $\Delta^*$ by solving **Equation 14**.
3. If $\Delta^*$ is greater than a preset maximal step size $\Delta_{max}$ then set $\Delta^* = \Delta_{max}$.
4. If there is a selected segment between positions $x$ and $x + \Delta^*$ then set $\Delta^*$ such that $x + \Delta^*$ is the midpoint between $x$ and the beginning of the (closest) selected segment. This step assures that we do not 'skip' selected segments.
5. Convert $\Delta^*$ from Morgans to base-pairs, rounding downwards. But if the step $\leq 1$ bp then set it to 1 bp, calculate $b(x + \Delta^*)$, set $x$ to $x + \Delta^*$, and return to step 1.
6. Calculate $b(x + \Delta^*)$ . If $|b(x + \Delta^*) - b(x)| > \epsilon/u_M$ then set the step size in Morgans to $\Delta^*/2$ and return to step 4. Otherwise, set $x$ to $x + \Delta^*$ and return to step 1.

### Interpolation and representation of lookup tables

We calculate $b(x)$ or $s(x)$ at every autosomal position $x$ (for a given selection coefficient and selected annotation) by linear interpolation between adjacent positions at which we calculated $s$ and $b$ accurately. We then discretize the values of $b(x)$ or $s(x)$ on a linear grid of values corresponding to the preset accuracy $\epsilon$, and group together contiguous autosomal segments with the same discrete value. We intersect these segments with our list of putatively neutral sites (Section 3.1) to obtain lookup tables consisting of contiguous segments of putatively neutral sites with the same coarse-grained $s$ and $b$ values for our sets of selected annotations and selection coefficients.

### 1.3 Binning neutral sites

A direct calculation of the composite log-likelihood function for given sets of selected annotations and selection coefficients and parameters (**Equation 5**) requires that we store and access lookup tables and calculate the log-likelihood function at $\sim 6.5 \times 10^8$ putatively neutral autosomal sites (see Section 2.1). Doing so would entail high computation and memory demands in the search for selection parameters that maximize the composite-likelihood. For example, our best-fitting models of background selection (see Main Text) with a grid of 6 selection coefficients would require storing and repeatedly accessing lookup tables that amount to $6.5 \times 10^8 \times 8 \times 6 \approx 32$ GB (given a precision of $\epsilon = 0.01$), and models involving multiple annotations for background selection and sweeps push the memory requirement to hundreds of GBs.

We reduce the computational and memory demands by dividing the set of putatively neutral sites into bins in which all the effects of background selection and sweeps predicted by the lookup tables and our estimates of the local (relative) mutation rate ($u(x)/\bar{u}$ in **Equation 8**; Section 3.3) are identical. The composite

log-likelihood function can then be calculated by summing over log-likelihood functions corresponding to bins, where the calculation per bin requires only the bin-specific parameters and bin-specific summaries of polymorphism. The number and identity of bins varies with the sets of selected annotations and selection coefficients and parameters and with the precision ($\epsilon$). For our best-fitting models, the average number of sites per bin is ~100, implying a ~100 fold reduction in demands on memory and in the number of log-likelihood calculations. For our most complex selection models (Section 4), the binning reduces memory and computational demands tenfold.

## 1.4 Optimization

Here we describe how we developed and tested the algorithm we use in order to find the selection parameters that maximize the composite-likelihood of our different models. The high dimensional parameter space (including up to 55 parameters in the most complex model in Section 4) potentially makes this optimization problem non-trivial.

### One step optimization

First, we tested the performance of standard optimization algorithms from the SciPy minimization toolkit (*Virtanen et al., 2020*). To this end, we generated polymorphism datasets based on our best-fitting model of background selection based on phastCons conservation scores, as follows:

1. We fixed the total deleterious mutation rate to $u_d = 10^{-8}$ per base pair per generation, and randomly divided it among the 6 selection coefficients of the model by sampling from a Dirichlet distribution (with $\alpha = 1$). We set the expected neutral diversity level in the absence of background selection to $\pi_0 = \pi_{YRI}$, where $\pi_{YRI}$ is a value of $\pi_0$ from an iteration of our best-fitting phastCons-based model using polymorphism data from the Yoruba (YRI) population (Section 2.1).
2. We generated the map of expected neutral diversity levels in autosomes given the chosen parameters. The map was represented in terms of the expected levels at each bin of putatively neutral sites (see Section 1.3).
3. We generated a polymorphism dataset corresponding to a sample size $n = 108$ pairs of (haploid) autosomes by picking the number of pairwise differences in each bin such that the average diversity level in it most closely matched the level predicted by the map. The discretization step introduces small differences between average and expected diversity levels in bins.

We tested each algorithm by applying it to *10 simulated datasets*, with *3 sets of initial conditions* for each dataset, corresponding to weak, intermediate and strong background selection (with $u_d = 5 \times 10^{-10}$, $5 \times 10^{-9}$, and $5 \times 10^{-8}$ per base pair per generation, respectively), and *5 randomly chosen initial conditions* in each set (with the total rate divided among the 6 selection coefficients by sampling from a Dirichlet distribution with $\alpha = 1$) amounting to *150 runs*. The initial value of $\pi_0$ was always set to the average diversity level in the dataset $\bar{\pi}$.

None of the algorithms closely converged to the ground truth parameters in all cases. Nelder-Mead downhill simplex minimization (*Nelder and Mead, 1965*) (NM) and Constrained Trust Region minimization (*Conn et al., 2000*) (CTR) performed the best overall, closely recovering the true parameters in ~2/3 of cases. While CTR was slightly more reliable, it was also up to ten times slower than NM. We therefore decide to combine them in order to leverage the relative strengths.

### Two-step minimization algorithm

After some experimentation we converged on the following two-step algorithm (*Appendix 1—figure 3*):

1. We apply NM with multiple initial conditions. For models of background selection with a single selected annotation we generate 3 sets of initial conditions, with 5 randomly chosen initial conditions per set, as we described above. For models of sweeps with a single annotation we generate the initial conditions analogously. Namely, we generate 3 sets of initial conditions corresponding to a low, intermediate and high proportion of beneficial substitutions (with

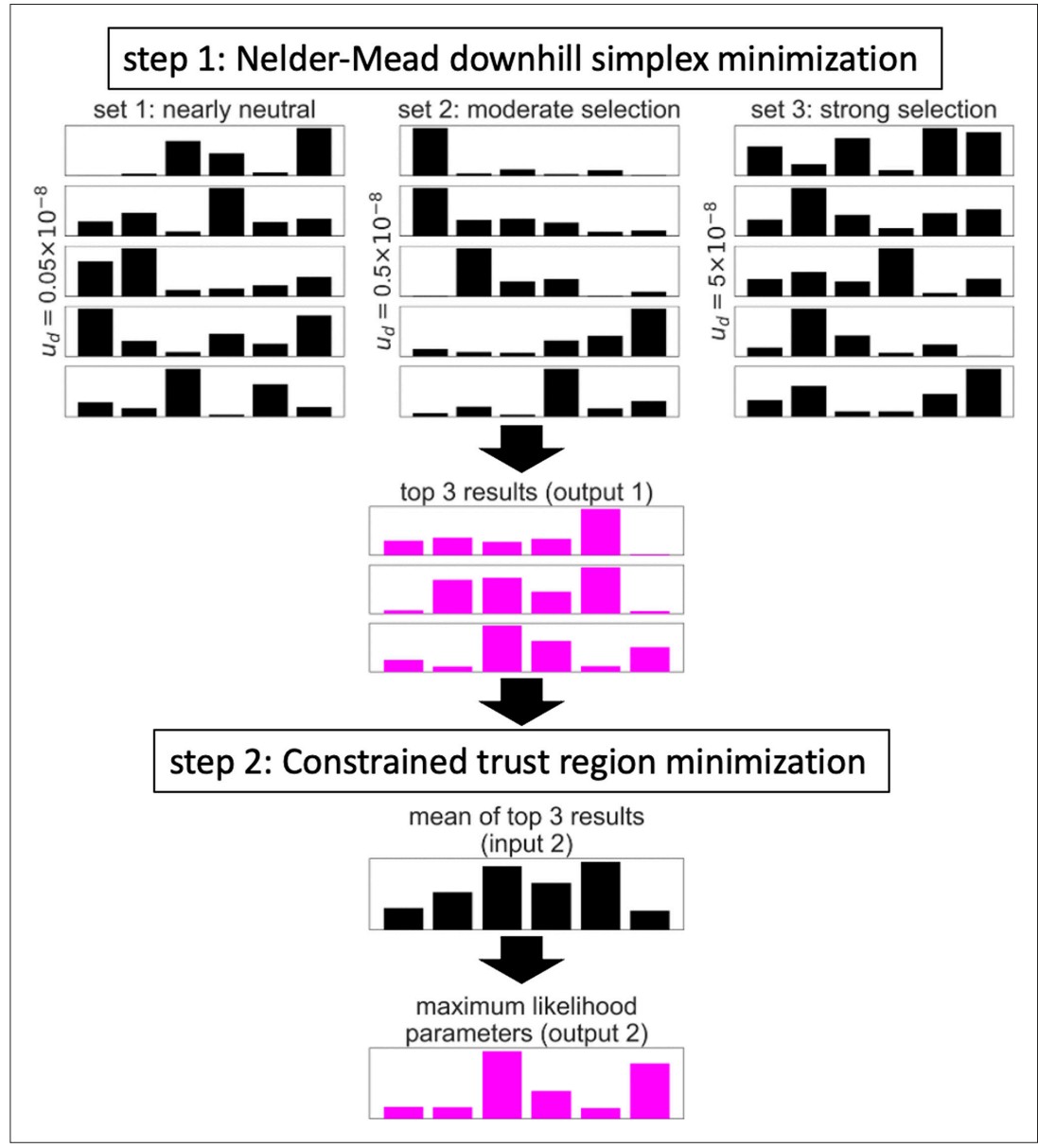

**Appendix 1—figure 3.** Illustration of the two-step algorithm. In this example, the optimization is applied to a model of background selection with a single selected annotation and a grid of 6 selection coefficients. See text for details.

$\alpha = 0.0125, 0.125$ and $1$, respectively) with 5 randomly chosen initial conditions per set (with the total proportion divided among selection coefficients by sampling from a Dirichlet distribution with $\alpha = 1$). For models with background selection and sweeps and/or multiple annotations, we generate 3 sets of initial conditions, corresponding to the weak/low, intermediate, and strong/high categories, with 5 random initial conditions per set that are chosen similarly for each mode and annotation. In all cases, the initial value of $\pi_0$ is set to the average diversity level in the dataset $\bar{\pi}$.

2. We apply the CTR algorithm with a single initial condition that is chosen based on the output of the previous step. Specifically, we focus on the sets of selection parameters inferred in the 3 out

of 15 initial runs that yielded the highest composite-likelihood, and use their average as our initial condition.

We tested the two-step algorithm under a variety of scenarios. When we applied it to the aforementioned 'deterministically' simulated datasets corresponding to the best-fitting model of background selection, it always closely recovered the ground truth parameters (*Appendix 1— figure 4*). The tiny differences between predicted and simulated diversity levels introduced by discretizing sometimes caused tiny differences between the inferred and ground-truth parameter values (see e.g. *Appendix 1—figure 4c*), but the composite log-likelihood of the inferred parameters was always higher, indicating that the algorithm is working well. Moreover, the runtime of the CTR algorithm in step 2 was typically short, presumably because its initial conditions were close to the true maximum.

We also tested the algorithm on simulated datasets that include substantial noise in diversity levels. We generated the datasets for a sample size $n = 2$ by sampling the number of pairwise differences in a bin of neutral sites from a Binomial distribution with a probability of success that equals the predicted diversity level (replacing step 3 in the simulations described above). The parameters inferred by our optimization algorithm were always similar to those used in the corresponding simulations, but with noticeable differences (*Appendix 1—figure 5*). In all cases, however, the composite-likelihood of the inferred parameters was greater than that of the ground-truth parameters indicating that the differences were due to overfitting (which is expected given the noise we introduced in the simulations) rather than a problem in the optimization.

Lastly, we tested the optimization algorithm on datasets simulated under a joint model of background selection and selective sweeps. We modeled the effects of sweeps driven by nonsynonymous substitutions, assuming that they made up $\alpha = 0.25$ of the nonsynonymous substitutions on the human lineage since divergence from the common ancestor with chimpanzees (see Section 2.7), and randomly dividing this proportion among 6 selection coefficients of by sampling from a Dirichlet distribution (with $\alpha = 1$). We modeled background selection as we detailed above, and generated the dataset using the 'noisy' simulation scheme corresponding to a sample size of $n = 2$. The parameters inferred by our optimization algorithm were always similar to those used in the simulations, with greater composite-likelihood of inferred than of ground-truth parameters indicative of overfitting (*Appendix 1—figure 6*) as we observed in the case with background selection alone.

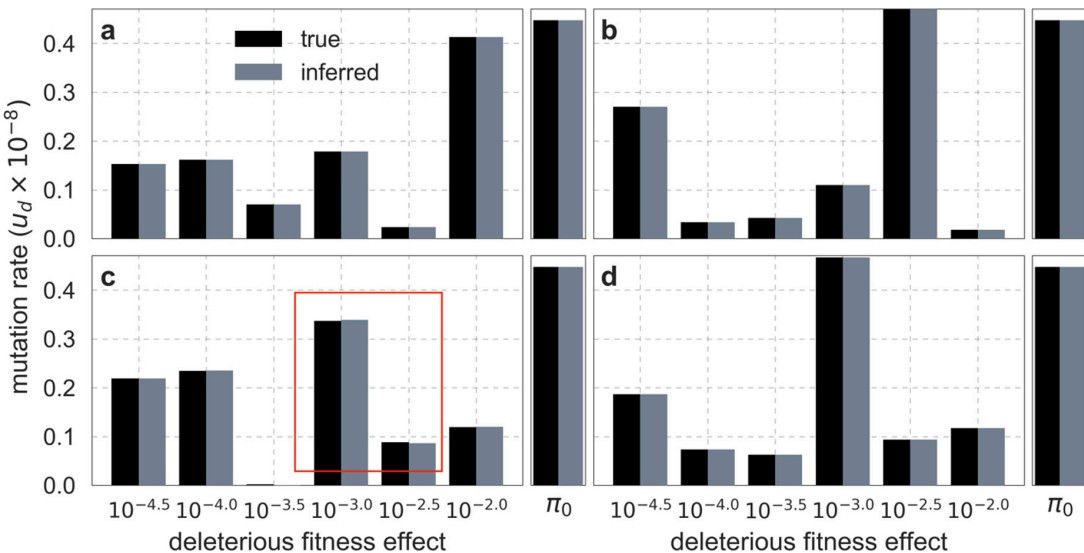

**Appendix 1—figure 4.** Comparison of inferred and ground-truth parameters for datasets simulated 'deterministically' under the best-fitting background selection model. Panels **a-d** correspond to different simulated datasets. Boxed region in (**c**) highlights the small differences between inferred and ground truth parameters introduced by discretization.

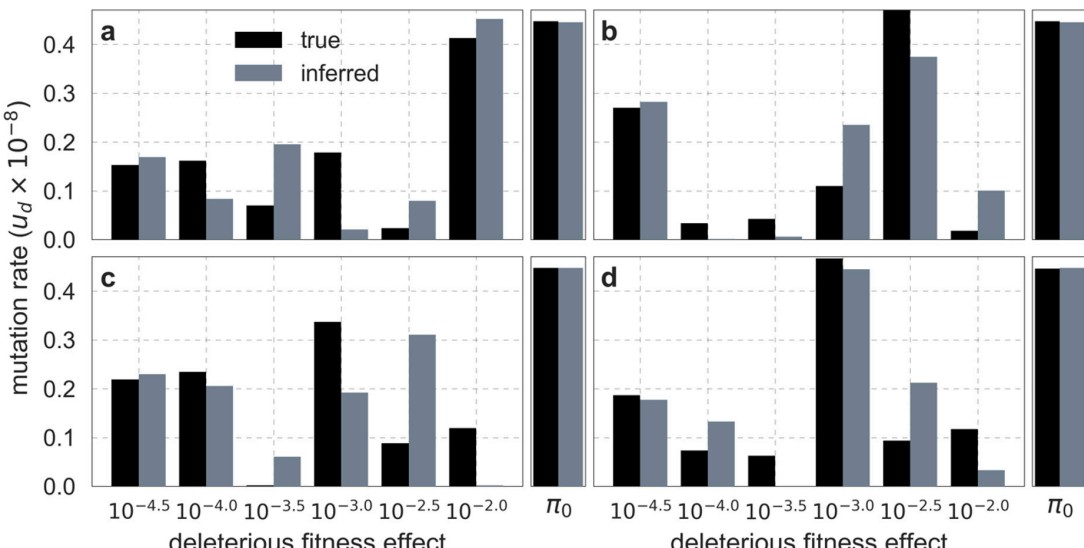

**Appendix 1—figure 5.** Comparison of inferred and ground-truth parameters for datasets simulated with noise under the best-fitting background selection model. Panels **a-d** correspond to different simulated datasets.

We obtained similar results when we simulated datasets under a variety of scenarios corresponding to the combinations weak, intermediate and strong background selection ($u_d = 5 \times 10^{-10}$, $5 \times 10^{-9}$ and $5 \times 10^{-8}$ per base pair per generation, respectively) with low, intermediate, and high proportions of beneficial substitutions ($\alpha = 0.0125, 0.125$ and $1$, respectively).

## 1.5 Thresholding

Our inference is strongly affected by forms of model misspecification that cause erroneous predictions of strong background selection effects (i.e., low values of $B$) and thus of low diversity levels at a relatively small proportion of neutral sites in our dataset. (We refer to neutral rather than putatively neutral sites for brevity and because low error in the identification of neutral sites is irrelevant to the problem at hand). These kinds of erroneous predictions can occur, for example,

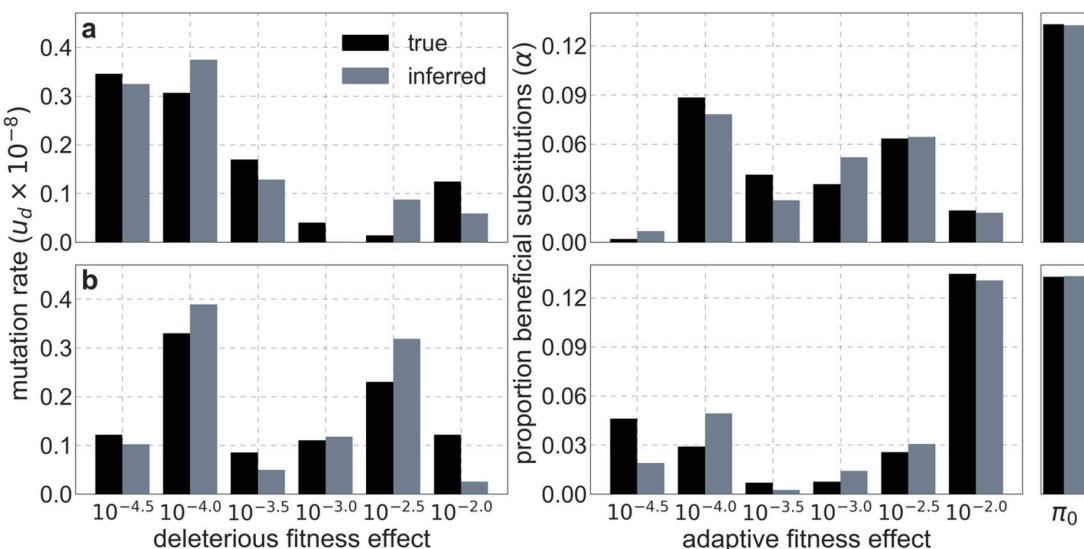

**Appendix 1—figure 6.** Comparison of inferred and ground-truth parameters for datasets simulated with noise under a joint model of background selection and selective sweeps. Panels **a and b** correspond to different simulated datasets.

at neutral sites near regions that are incorrectly annotated as conserved or that are truly conserved but have proportionally fewer weakly deleterious mutations than most similarly annotated regions (because weakly deleterious mutations have strong localized effects on diversity levels). Even when neutral sites near such regions make up a small proportion of the dataset, having more of them be polymorphic than predicted can substantially reduce the composite-likelihood of models that may otherwise fit the data well (see *Equation 5*), potentially biasing our inference. Here, we present evidence for this problem, show how we modify our inference to solve it – by imposing a lower threshold for the value of $B$ in the lookup tables or in the optimization, and address the consequences of this modification.

In *Appendix 1—figures 7–9*, we compare the results of our inference with and without thresholding for our best-fitting CADD-based model (the results for other models are qualitatively similar). Under the aforementioned forms of model misspecification, we might expect excess neutral diversity in regions where background selection is predicted to be strongest. Accordingly, when we apply the inference with little or no thresholding and focus on 1% of neutral sites where background selection is predicted to be the strongest, we find that observed diversity levels are up to twofold higher than our predictions (*Appendix 1—figure 7a*). Additionally, we expect this form of model misspecification to bias the inferred distribution of selection effects toward larger selection coefficients, because smaller selection effects cause a more localized reduction in diversity levels and are therefore expected to be heavily penalized by having even relatively few misspecified regions. Accordingly, we find that the inferred distribution without thresholding is shifted toward greater selection coefficients ($t \geq 10^{-2.5}$) compared to the distributions with thresholding (*Appendix 1—figure 7c(i)*).

Importantly, the map of background selection effects generated without thresholding fits the data more poorly than the maps with thresholding. Notably, when we compare observed and predicted diversity levels around nonsynonymous substitutions, we find that the predictions generated without thresholding underestimate the reduction in diversity levels near nonsynonymous substitutions (inset in *Appendix 1—figure 7b*). This can be explained by the bias toward larger selection coefficients, which causes the inference without thresholding to underestimate the reduction in diversity levels near conserved regions that are specified correctly (in order to avoid the reduction in diversity levels near misspecified regions). Additionally, when we compare the fit of maps with and without thresholding, we find that without thresholding the composite-likelihood is lower (*Appendix 1—figure 7c(iv)*), the variance in diversity levels explained throughout the range of window sizes is lower (*Appendix 1—figure 7d* and *Appendix 1—figure 8*) and the calibration of our predictions is poorer (*Appendix 1—figure 7a*; this remains the case when we exclude the top and bottom 5% of our predicted values, such that the predictions with and without thresholding span the same ranges of values; for example, Pearson $R^2$ of 0.99 and 0.97 with a threshold of $B = 0.6$ and without thresholding, respectively).

We considered two ways of thresholding, where in both we set any value of $B$ that is below the threshold to the threshold value: (1) applying the threshold in the lookup tables, that is, *before* the composite-likelihood maximization step, and (2) applying the threshold *at each step* of the maximization, when $B$ values are calculated for a given distribution of selection effects (see *Equation 6*). The two approaches yield similar improvements in fit at equivalent threshold levels, and even applying a relatively low threshold improves fits markedly compared to *B*-maps without thresholding (*Appendix 1—figure 8*). Based on our metrics of fit, we find that applying a threshold of $B = 0.6$ in the lookup tables yields the best fits (*Appendix 1—figures 7–9*), although thresholds within the range $0.45 \leq B \leq 0.65$ yield comparable results. Nonetheless, lower thresholds yield better fits to data in regions of the genome where selection is particularly strong (e.g. *Appendix 1—figure 7a–b*, red box in *Appendix 1—figure 9*). It may therefore be useful to use a lower threshold when considering regions of the genome that are subject to especially strong background selection. We provide *B*-maps for a range of thresholds that can be downloaded at https://github.com/sellalab/HumanLinkedSelectionMaps (in addition to the 'best-fitting *B*-maps' presented in the Main Text).

While thresholding largely resolves the aforementioned problem of model misspecification, it also introduces some problems. First, as we already noted, it leads to an underestimation of background

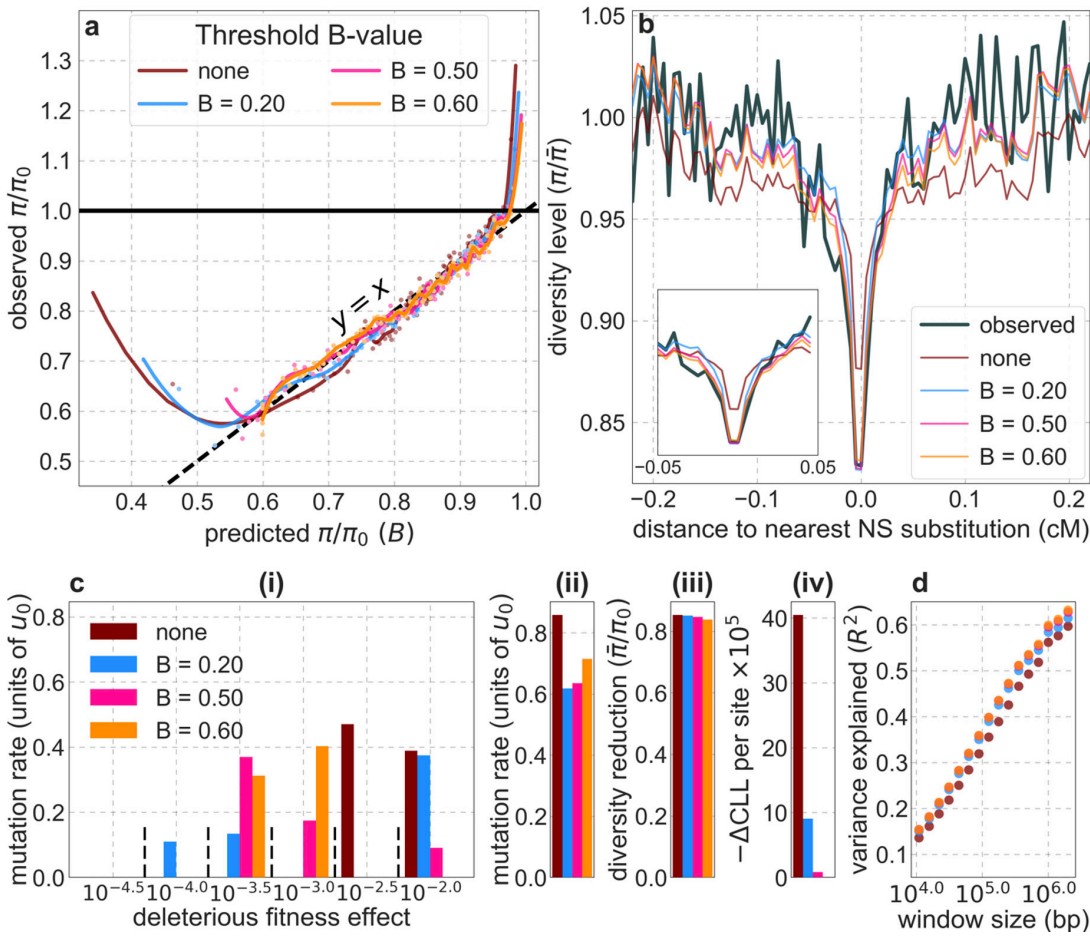

**Appendix 1—figure 7.** Comparison of inference results with and without thresholding. The results shown correspond to our best-fitting CADD-based model (see Main Text), with threshold values of $B = 0$ (without threshold, labeled 'none'), $0.2$, $0.5$, and $0.6$ applied in the lookup tables. (**a**) Observed vs. predicted neutral diversity levels across the autosomes. The graph was generated as detailed in *Figure 5*. Note that the division of neutral sites among bins varies with the choices of thresholds because it is based on corresponding maps. (**b**) Observed vs. predicted neutral diversity levels as a function of genetic distance from human-specific nonsynonymous (NS) substitutions. The graph was generated as detailed in *Figure 3*, using a narrower range of genetic distances to NS substitutions to highlight differences among thresholds. (**c**) Parameter estimates and summaries of the inferences. From left to right: (i) The estimated distribution of fitness effects, described in terms of the rate of mutation per generation with a given selection coefficient. Mutation rates (throughout) are measured relative to the estimate of the total mutation rate in humans, $u_0 = 1.4 \cdot 10^{-8}$ per bp per generation (see Section 5). (ii) The total deleterious mutation rate ($u_d$) measured in units of $u_0$. (iii) Our prediction of the mean reduction in neutral diversity level due to background selection, measured as the ratio of the average predicted level across the genome, $\bar{\pi}$, to the predicted level in the absence of selection at linked sites, $\pi_0$. (iv) The reduction in composite log-likelihood (CLL) per site relative to the model with the highest CLL. Differences in CLL should be interpreted with caution, as this measure does not account for linkage disequilibrium. (**d**) The proportion of variance in diversity levels explained ($R^2$) on different spatial scales (measured in non-overlapping windows).

selection effects at ~5% of the genome in which background selection effects is predicted to be the strongest. Second, thresholding potentially biases our estimates of the distribution of selection effects. While this bias is probably smaller than the bias without thresholding, its form and magnitude are not obvious. This is why we decided not to report the inferred distributions of selection effects in the Main Text. We are working on more principled ways of resolving the problems introduced by model misspecification, but these fall beyond the scope of the current paper.

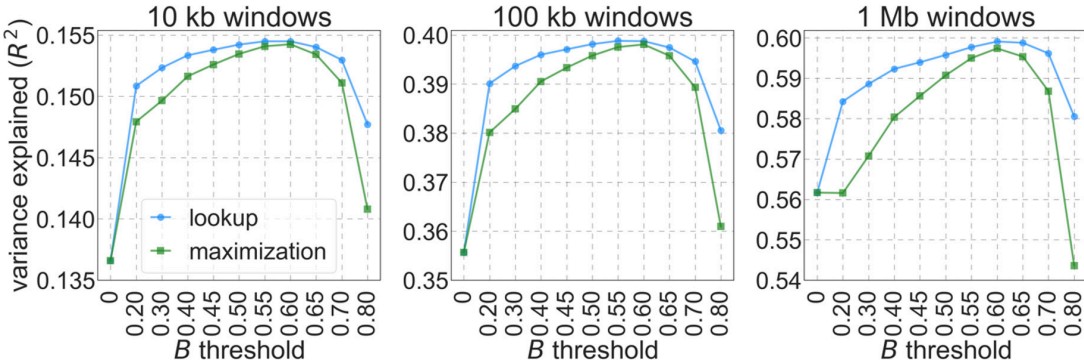

**Appendix 1—figure 8.** The proportion of variance explained in 10 kb, 100 kb, and 1 Mb windows using a range of $B$ thresholds applied to lookup tables ('lookup') or during maximization ('maximization').

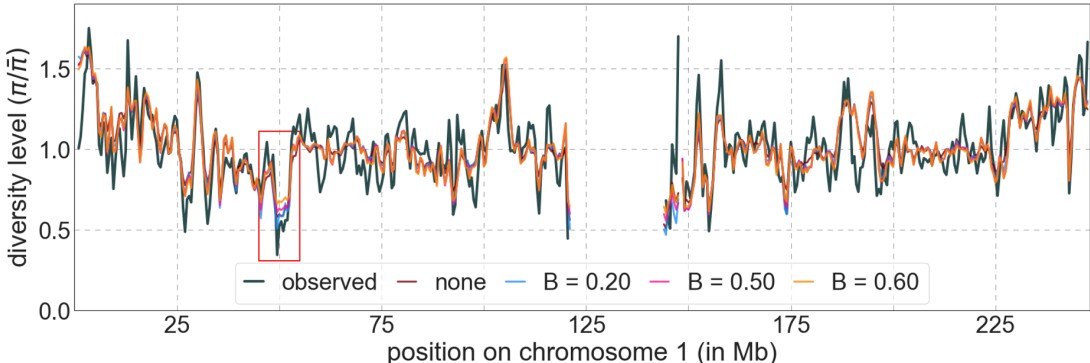

**Appendix 1—figure 9.** Predicted and observed diversity levels along chromosome 1 in the YRI sample. Diversity levels are measured in 1 Mb windows, with a 0.5 Mb overlap, with the autosomal mean set to 1. Thresholds were applied in the lookup tables. Lower thresholds yield better predictions in regions with low diversity levels, for example, near 50 Mb (red box).

## 1.6 Software

We provide a set of Python programs to download and format the genomic data that we use (see Section 2), infer maps of the effects of linked selection and reproduce all of the analyses and figures described in this study (https://github.com/sellalab/HumanLinkedSelectionMaps). We rely on publicly available software for some steps, including the PHAST package (*Siepel and Haussler, 2004*; *Siepel et al., 2005*), which we use to identify conserved regions and to estimate substitution rates (see Sections 3 and 5), and a modified version of the calc_bkgd program from *McVicker et al., 2009*, which we use to generate lookup tables of the effects of background selection (see Section 1.2).

### Running the inference pipeline

The inference pipeline is controlled by a data structure called RunStruct, which is initialized with information about input/output file paths used, model parameters and other control variables, such as the precision $\epsilon$ of lookup tables (see Section 1.2) and the $B$ threshold (see Section 1.5). Once RunStruct has been initialized, the pipeline proceeds through the following steps:

1. Download and organize input files (annotations, genetic maps, etc.).
2. Create lookup tables of the effects of background selection and/or selective sweeps (Section 1.2) for the given set of selected annotations and grid of selection coefficients.
3. Organize polymorphism dataset that includes polymorphism data at putatively neutral sites (Section 2.1), corresponding estimates of substitution rates (Section 3.3) and corresponding values of lookup table into our compressed bins format (Section 1.3).
4. Run the two-step optimization algorithm to obtain estimates of model parameters, a map of the predicted effects of linked selection, and summary statistics including, for example, the estimated deleterious mutation rate ($u_d$) and proportion of beneficial substitutions ($\alpha$) associated with different annotations and the average reduction in diversity levels ($\bar{\pi}/\pi_0$).

## Parallelization and runtimes

The composite-likelihood calculations during optimization can be partitioned into sums over subsets of bins of neutral sites, which in turn allows us to parallelize the optimization. The number of processing cores used in optimization is controlled by RunStruct. For our best-fitting models of background selection, loading lookup tables and neutral polymorphism data and running the two-step optimization requires ~1 GB of memory for each of the 15 processes in step 1 and the single process in step 2. Running each process on a single core takes ~12–24 hr or ~200–400 CPU × GB hours. The computing cluster we used allows up to 12 cores per process and thus using parallelization we were able to run the optimization for the best-fitting models in 1–2 hr. Our most complex models (see Section 4) required up to 10 GB of memory per process and took up to 60 hr with using 12 cores (i.e. ~ $10^4$ CPU × GB hours).

## 2. Data sources and filters

### 2.1 Polymorphism data

We download 1000 Genomes Project phase 3 VCF files for all 26 populations from across the world (*Auton et al., 2015*). Unless otherwise noted, results in the Main Text and Appendix 1 are based on autosomal data from Yoruba (YRI); the results for other populations are reported in Sections 7 and 9 of this Appendix 1.

We apply several filters to these data. First, we restrict our analysis to bases that pass all filters, denoted 'P' in the 1000 Genomes Project strictMask accessibility mask (*Abecasis et al., 2012*; *Auton et al., 2015*). In addition, we remove low-complexity, simple repeats, duplications, and hg19 build gaps using repeatMasker files downloaded from UCSC (*Karolchik et al., 2004*). For each population, we restrict polymorphic sites to that population's subset of biallelic SNPs from VCF files, excluding indels and other variants using VCFTools (*Danecek et al., 2011*). Remaining sites are treated as monomorphic.

We apply additional filters to restrict our analyses to putatively neutral sites. First, we remove the union of genic regions, as detailed in section 2.4. Second, we remove all remaining sites with phastCons conservation scores greater than 0.001 as described in section 3.1. Third, we remove putatively neutral sites at the telomeric ends of autosomes, near the edges of our genetic maps (Section 2.3), as detailed in Section 3.2. Accessibility and repeat masks remove ~33.3% of all autosomal sites; excluding genic regions removes an additional ~3.3%; filtering based on phastCons scores removes another ~40.5%; and filtering sites at the telomeric ends removes ~1.2% more. We are left with a set of ~653 M putatively neutral sites, which correspond to ~23% of autosomal sites (based on hg19 build).

### 2.2 Multiple species alignment data

We rely on multiple sequence alignments to identify phylogenetically conserved and non-conserved regions of the genome, as well as for estimating local variation in neutral substitution rates (see Sections 3 and 4). To this end, we download mutation annotation format (MAF) files containing 99 vertebrate genomes aligned to the human genome (build hg19), using the Multiz software from UCSC (*Blanchette et al., 2004*).

### 2.3 Genetic map

We use the (*Hinch et al., 2011*) genetic map, which was inferred from ancestry switches in African-Americans. At the >10 kb scale, it is highly correlated to other fine-scale maps (e.g. *Frazer et al., 2007*; *Halldorsson et al., 2019*). Among high-resolution genetic maps in humans, however, this one is likely the least confounded by diversity levels along the genome.

### 2.4 Human gene annotations

We use genic annotations from the UCSC knownGene database (*Hsu et al., 2006*) to identify putative targets of selection as well as regions that should be removed from our set of putative neutral sites. To this end, we rely on exon coordinates from knownGene transcripts to identify four kinds of annotations: (1) upstream/downstream regions, defined as 1 kb upstream of a transcript start and 1 kb downstream of a transcript end; (2) untranslated region (UTR), both 5' and 3'; (3) protein coding sequences (CDSs); and (4) splice regions, defined as 200 bp from the start and end of each intron.

For the purpose of identifying putative targets of selection, we rely on a non-overlapping subset of these four annotations. For genes with multiple splice variants, we keep only the set of exons

within the longest isoform. In rare cases of two overlapping gene predictions, we retain the gene with the longer exonic sequence. For the purpose of removing putatively functional regions from our set of putative neutral sites, however, we remove the union of all four annotations for all gene transcripts.

## 2.5 CADD scores

We use CADD scores (*Kircher et al., 2014*; *Rentzsch et al., 2019*) in order to annotate putative targets of selection in a couple of models (Sections 4.4 and 4.5). The standard CADD scores rely on the map of background selection effects generated by *McVicker et al., 2009* as one of their inputs. While this input has minor effects on CADD scores (i.e. the top 1–10% of scores; see Table S3 in *Kircher et al., 2014*), in order to avoid any measure of circularity we approached the Kircher Lab (Martin Kircher, Lusiné Nazaretyan, Philip Rentzsch and Max Schubach), who manage the development of CADD scores, and who kindly agreed to generate and share a version of CADD score without the background selection map as input (this set of CADD scores is available on request from either the Kircher or Sella labs). For each site in the genome, we retain the highest of the three CADD scores (corresponding to the three possible point mutations). We use the distribution of scores across the autosomes to determine cutoffs for our annotations (e.g. sites within the top 6% of scores) and use sites with scores that exceed these cutoffs as putative targets of selection (sometimes in conjunction with another annotation, e.g. exons).

## 2.6 ENCODE cCRE annotations

In two of our models (Section 4.4), we consider regulatory elements identified by the ENCODE project as putative targets of selection (*Moore et al., 2020*). To this end, we download ENCODE candidate cis-regulatory elements (cCREs) from the Tier 1 a group of biosamples, which include experimental support from all relevant assays used to define elements: high DNase signal and high H3K4me3, H3K27ac or CTCF signal (*Moore et al., 2020*). The resulting cCREs are categorized as (1) enhancer-like signatures (ELS), (2) promoter-like signatures (PLS), (3) CTCF-bound (CTCF) and (4) poised elements marked by DNase and H3K4me3 (H3K4me3). cCRE annotations were downloaded for each individual Tier 1 a biosample using the SCREEN tool (*Moore et al., 2020*) and lifted over from hg38 to hg19 coordinates.

## 2.7 Substitutions in the human lineage

We rely on an estimate of the human-chimpanzee ancestor inferred using the Enredo-Pecan-Ortheus (EPO) 6-species alignment pipeline (*Paten et al., 2008*) to identify likely substitutions on the human lineage. We use subsets of these substitutions that arose in putative targets of positive selection as candidate substitutions resulting in classic sweeps (Section 4.5). We derive sets of likely substitutions in a couple of different ways. First, we compare the reconstructed ancestral genome with the human hg19 reference, taking the differences as putative substitutions. In this case and others, we do not differentiate between low and high confidence calls (lower and upper case, respectively) in the estimated ancestor. Because the hg19 reference genome is a composite of genomes with different ancestries (*Church et al., 2011*), we also consider population-specific inferences of substitutions for YRI and CEU. To this end, we compare the reconstructed ancestral genome with the polymorphism data collected in the 1000 Genome Project for a given population. If a site is monomorphic in the population and differs from the HC ancestor, we include the site in our set of substitutions. For biallelic sites where one of the two alleles is ancestral, we randomly choose one of the alleles with probabilities that are weighted by allele frequency; if the chosen allele is the derived one, the site is considered a substitution. We generate two such samples for a given population to see whether different choices of substitutions affect our results. In practice, each of these sets differs from the set based on the hg19 reference at fewer than 1% of sites, the differences between the two samples for a given population are even smaller, and the results of our inference end up being insensitive to these differences (Section 4.6).

## 2.8 Covariates of $B$

In Section 8, we ask whether genomic features that covary with $B$ could account for the divergence between observed and predicted diversity levels in the ~2% of sites in which background selection is predicted to be the weakest. In addition to annotations of features whose sources were already mentioned, we also use the following datasets: (1) BED files of CpG islands downloaded from the UCSC Table Browser *Karolchik et al., 2004*; (2) BED files of testis CpG methylation levels in human males downloaded from the GEO database (GEO accession: GSM1127119; *Barrett et al., 2013*); (3)

coordinates of C>G hypermutable regions, given at 1 Mb resolution, taken from the Supplemental Information of *Jónsson et al., 2017*; (4) coordinates of centromeres and telomeres taken from the hg19 gaps track in the UCSC Table Browser *Karolchik et al., 2004*; (5) inferred proportions of archaic ancestry in European (CEU) and East-Asian (CHB/CHS) populations based on estimates from *Steinrücken et al., 2018*.

## 3. Choice of exogenous parameters

Fitting our model to data requires several choices beyond those of datasets and filters. Here, we describe how we chose our set of putatively neutral sites and estimate the substitution rate at these sites. In Section 4, we describe how our results depend on the choice of targets of selection.

### 3.1 Choosing putatively neutral sites based on phylogenetic conservation

Our main source of information for choosing the set of putatively neutral sites is the degree of conservation in multiple species alignments. To this end, we rely on running phastCons (*Siepel et al., 2005*) on subsets of the 99-vertebrate alignment (from which we exclude the human genome). PhastCons fits a phylogenetic hidden Markov Model (phylo-HMM) with two states, neutral and conserved, to multiple species alignments of contiguous sites along the genome using the relative substitution rates in the alignment columns to infer conservation. The phastCons score is the posterior probability that any given site is conserved. In principle, including more species in the alignment increases the power to distinguish between conserved and neutral sites (*Appendix 1— figure 10a*). However, as the phylogenetic distance from humans increases, sequence conservation might become less informative about conservation in humans because of functional turnover (*Ward and Kellis, 2012*; *Rands et al., 2014*). In practice, the latter effect is ameliorated by the fact that phastCons only uses information at aligned sites and the proportion of the genome that aligns to the human reference decreases with phylogenetic distance (*Appendix 1—figure 10b*), especially in regions with considerable turnover.

In relying on phastCons scores to identify a set of putatively neutral sites, we need to choose two parameters: the phylogenetic depth of species included in the alignment and the cutoff conservation score below which a site will be considered neutral. In both cases, we pick the parameter values that maximize the variance in diversity levels explained by our best-fitting models (*Appendix 1— figure 11*). Given these criteria, we chose to base our set of neutral sites on the alignment of supra-primates (*Appendix 1—figure 11a*), and use the 35% of sites (in the set remaining after filters and removing genic regions; see Sections 2.1 and 2.4) with the lowest phastCons scores in this alignment, which includes sites with scores ≤ 0.001 (*Appendix 1—figure 11b*). These choices are robust to the phylogenetic depth used to specify the selection targets (see Section 4) and to the window size in which we measure the variance explained by our model (we show the results for windows of 1 Mb in *Appendix 1—figure 11*).

### 3.2 Removing sites at the telomeric ends of chromosomes

The Hinch et al. genetic map (*Hinch et al., 2011*) does not include recombination rate estimates for ~0.5–1 Mb at the 5′ and 3′ ends of autosomes. Consequently, we are unable to describe background selection effects of putatively selected regions that lie in these telomeric regions, and our inferences and predictions at putatively neutral sites near the telomeres are less accurate. We therefore exclude putatively neutral sites in telomeric regions not covered by the genetic map. Similar to our approach in the previous section, we choose the map size of the region to remove based on how the choice affects the model fit to diversity levels across autosomes (*Appendix 1— figure 12a*). We find that filtering putatively neutral sites in 0.1 cM from the edge of the genetic map, which amounts to ~0.8% of neutral sites, largely removes this 'edge effect'. This genetic distance makes sense, as it is roughly one at which background selection effects of deleterious mutations with $s = 10^{-3}$ – the strongest selection effects inferred to contribute substantially (*Appendix 1—figure 12b*) – become negligible. Moreover, our estimates of model parameters are fairly insensitive to the removal of larger regions (*Appendix 1—figure 12b*).

### 3.3 Estimating local variation in mutation rates

We rely on estimates of substitution rates at putatively neutral sites along the genome to control for the effect of variation in mutation rates on neutral diversity levels (see *Equation 1* in Section 1.1). To this end, we use phyloFit (*Siepel and Haussler, 2004*) to estimate the substitution rate in a phylogeny, in windows of putatively neutral sites across the genome. We choose the species to

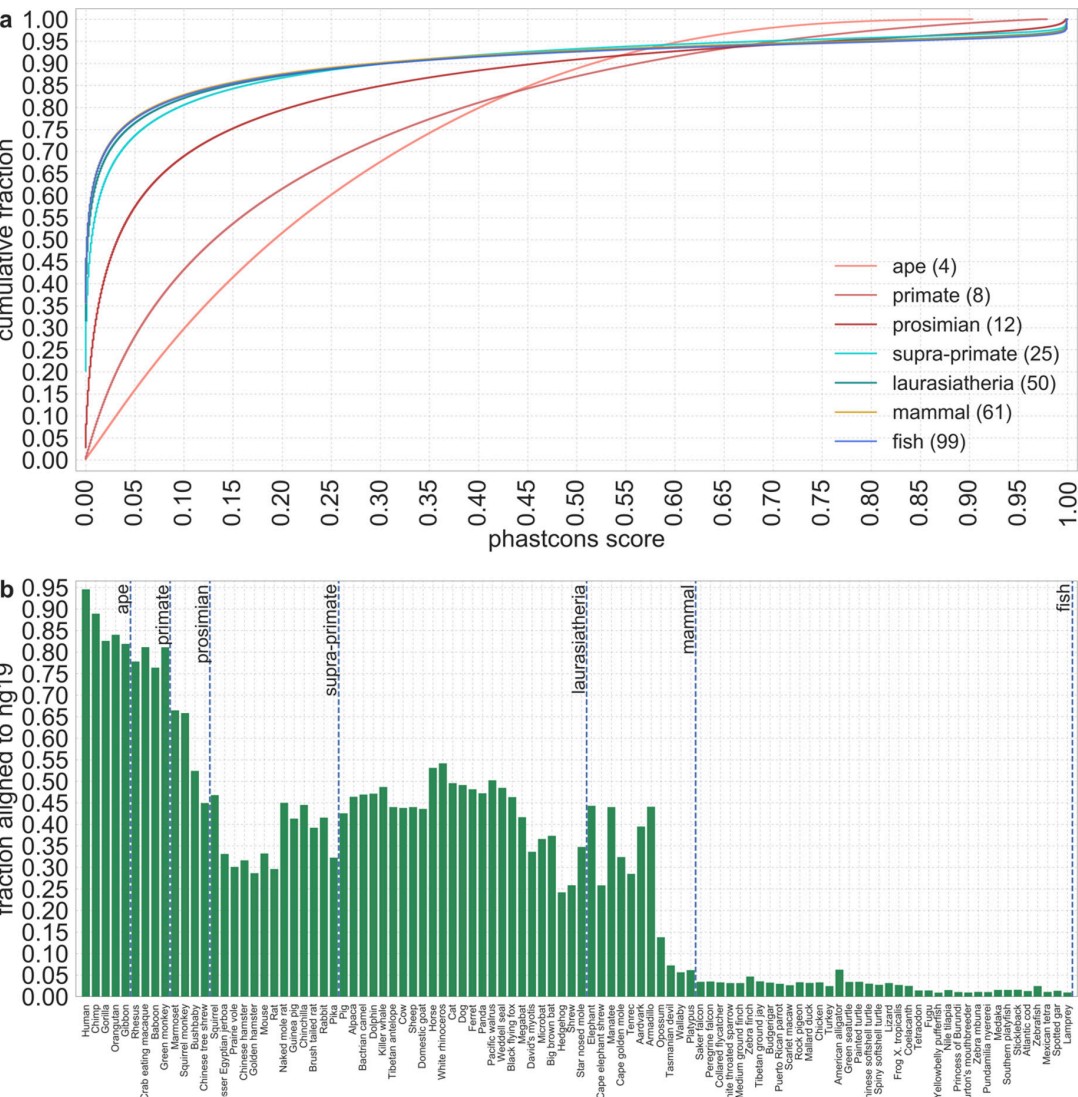

**Appendix 1—figure 10.** The distribution of phastCons scores across autosomes for varying phylogenetic distances from humans. (**a**) The number of species included at each phylogenetic depth is noted in the caption. As the number of species in the alignment increases, the ability to distinguish between conserved and neutral sites increases. (**b**) The proportion of a species' sequenced genome that aligns to the human reference (hg19) decreases with their phylogenetic distance from humans. The decrease is not monotonic because of other factors, for example, the quality of the sequencing. The proportion is not 1 for humans because of missing information in the reference genome.

include in the phylogeny based on the following considerations. The number of substitutions in a given window can be approximated by a Poisson random variable with expectation $\lambda$, which is proportional to the total branch length of the phylogeny, $T$, and the number of putatively neutral sites in the window, $n$. Consequently, the precision of our estimates of the relative mutation rate increase with $\lambda \propto n \cdot T$. Including more species in the phylogeny increase $T$ but reduces $n$, because it reduces the fraction of putatively neutral sites that align to the human reference in all the species included. *Appendix 1—figure 13a* shows the trade-off between the two factors, for all subsets of 9 primate species included in the 99-vertebrate alignment (see Section 2.2). We chose the subset that maximizes $n \cdot T$, which includes 8 of the 9 species (gibbon is removed) with an average of ~0.135 substitutions per putatively neutral site.

We estimate relative mutation rates along the genome based on the estimated substitution rates in the 8-primate phylogeny in windows with a fixed number of contiguous putatively neutral sites. Using windows with a greater number of sites decreases the sampling error but reduces the spatial resolution of our estimates. We use the variance in diversity levels explained by our best-fitting models

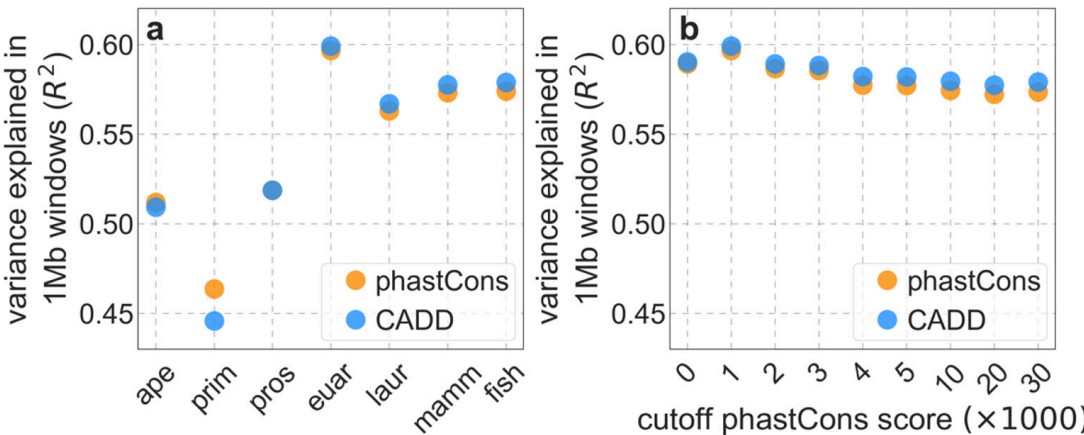

**Appendix 1—figure 11.** The variance in diversity levels explained by our two best-fitting models using different choices of putatively neutral sites. In (**a**) we vary the phylogenetic depth of the multi-species alignment (i.e. the maximal phylogenetic distance from humans to any/all of the other species) and in (**b**) we vary the cutoff phastCons score for the least conserved sites included in our set. The best fit corresponds to the least conserved 35% of sites (phastCons scores ≤ 0.001) in the supra-primate alignment (euar).

as a criterion for choosing the window size, finding that a window with 6000 putatively neutral sites performs best among the options we examined (*Appendix 1—figure 13b*). This choice corresponds to mean physical window sizes of 26,454 bp (with a S.D. of 18,455 bp) and to a mean relative error of ~3.3% in our estimates of the relative mutation rate per window. We also examined other ways of estimating the relative mutation rate, including using windows of fixed physical length and sliding windows with varying degrees of overlap, but none of these approaches yielded better results.

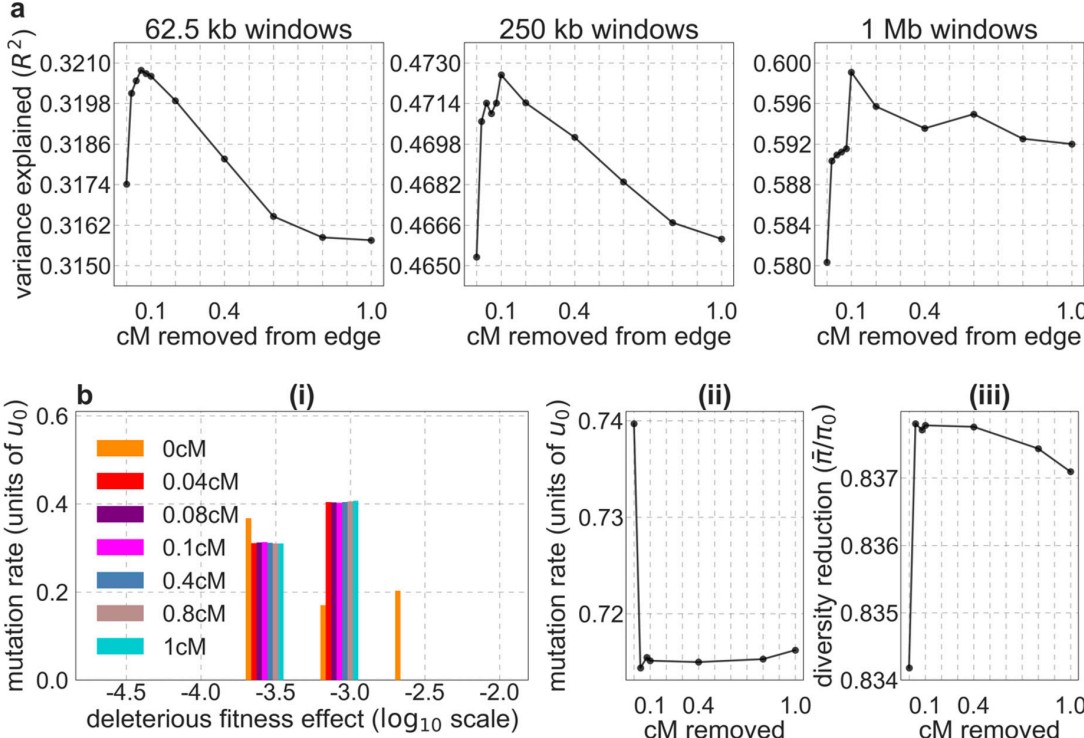

**Appendix 1—figure 12.** The effect of removing putatively neutral sites near telomeres on model fit and parameter estimates. We show the result for our best-fitting CADD-based model; results for phastCons scores are highly similar (not shown). (**a**) The proportion of variance in diversity levels explained for different window sizes, as a function of the size of the removed region (in cM). (**b**) (**i-iii**) Estimates of model parameters as a function of the size of the removed region (in cM).

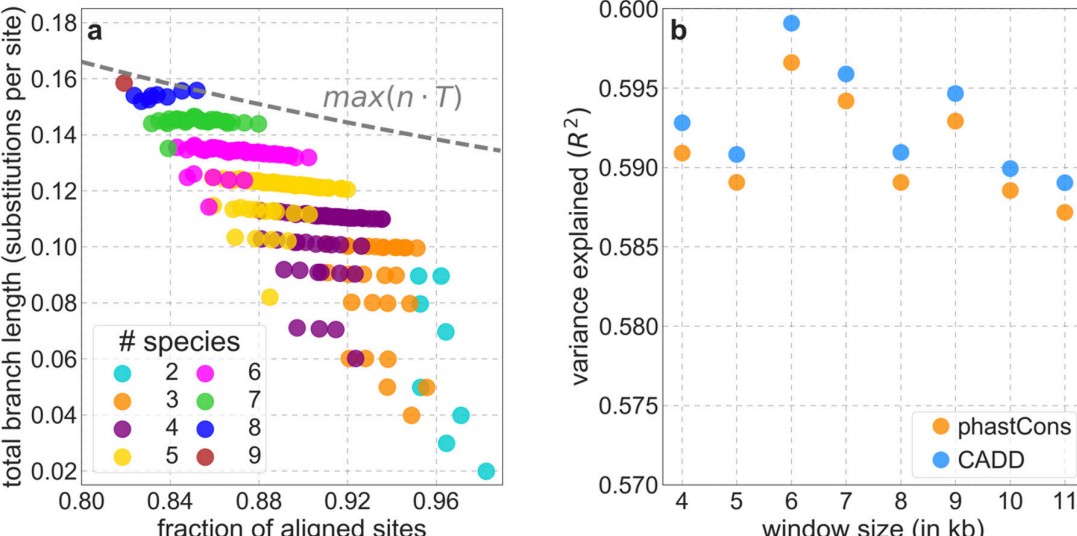

**Appendix 1—figure 13.** Choosing the parameters used in estimating the relative mutation rate at putatively neutral sites. (**a**) The trade-off between the fraction of aligned sites and total branch length for subsets of the primate phylogeny. The fraction of aligned sites is estimated for our set of putatively neutral sites, and the total branch length is measured in terms of the average number of substitutions per site on the phylogeny, estimated by phyloFit. The maximum product of the fraction and branch length is attained by including all primates included in the 99-vertebrate alignment other than gibbon. (**b**) The variance in diversity levels explained by our best-fitting models across 1 Mb windows, for different choices of window sizes (i.e. the number of putatively neutral sites) used to control for variation in mutation rates at putatively neutral sites.

In the analyses in which we bin neutral sites, either by their distance to genomic elements (e.g. *Figure 3*) or by predicted $B$ (e.g. *Figure 5*), we estimate the relative mutation rate in each bin. To this end, we use phyloFit (*Siepel and Haussler, 2004*) to estimate the substitution rate in the 8-primate phylogeny on all sites in that bin jointly and then normalize this estimate by the average across bins.

## 4. Fitting models with different targets of selection

Our framework allows us to fit models of background selection, selective sweeps, or both, based on different choices of putative targets of negative and/or positive selection. Here we detail the analysis of the models and choices that are described in the Main Text. We use several criteria to evaluate how well the models fit the data; these indicate that models of background selection alone in which the targets of selection are chosen based on constrained elements annotated by either phastCons or CADD scores are best supported by the data. We also compare the predictions of these models with those of *McVicker et al., 2009*.

### 4.1 Background selection model based on phylogenetic conservation

We first consider a model of background selection in which targets of selection are chosen based on phylogenetic conservation. We identify conserved genomic elements using phastCons scores (*Siepel et al., 2005*) calculated on monophyletic subsets of the 99-vertebrate alignment to the human genome (*Blanchette et al., 2004*), all of which exclude the human genome itself (see Section 2.2). We vary the phylogenetic depth of the subset of species considered (i.e. the maximal distance from humans). For a given depth, we obtain targets of selection by specifying a proportion of selected sites (i.e. of the total autosomal length in hg19) and choosing those sites that have the highest phastCons scores in the alignment (after excluding some sites, e.g. from up to 5% of the four-ape alignment to less than 0.1% of the 99-vertebrate alignment, that are in our putatively neutral set). As we have done for previous choices (e.g. Section 3.1), we examine how our choices of phylogenetic depth and of proportion of selected sites affect the models' fit to autosomal diversity levels.

We find the fit to be largely insensitive to the choice of phylogenetic depth, with models based on conservation in the full 99-vertebrate alignment fitting slightly better than other choices of depth (*Appendix 1—figure 14*). Notably, the explained variance in diversity levels (in windows of different

sizes) is similar across depths, increasing slightly with the number of species included, other than for the four-ape phylogeny (*Appendix 1—figure 14b and c*). The fits of predicted diversity levels along the genome (e.g. *Appendix 1—figure 14d*) and around genomic features (e.g. *Appendix 1—figure 14e*) are similar, with none of the choices of depth clearly outperforming others. Moreover, for all choices, the predicted diversity levels are well calibrated (*Appendix 1—figure 14f*), with the exception of regions in which background selection is predicted to be very weak, that is, $B \approx 1$ (see Section 8). When we restrict each annotation to the top 6% of scores in sites for which all phylogenetic depths include phastCons scores (~98% of sites satisfy this criterion), our results are unchanged.

Distantly related species, such as those added when we move from supra-primates ($n = 25$) to vertebrates out to lamprey ($n = 99$), have little effect on phastCons scores and thus on our models, because only a small proportion of their genomes align with humans (*Appendix 1—figure 10b*). This can be seen in the high correlations between the number of conserved sites based on different depths across windows of different sizes (*Appendix 1—figure 15a*). The spatial distribution of conserved sites is even fairly insensitive to varying the species included from four apes to 99 vertebrates (*Appendix 1—figure 15a*). Interestingly, we later show that the improvement in fit across 1 Mb windows of the model based on conservation in 99 vertebrates compared with models based on conservation in shallower phylogenies is statistically significant, except for the model based on four-apes (*Appendix 1—figure 33*), whereas the spatial distributions of conserved sites in the 99-vertebrate and four-ape models are the least correlated (*Appendix 1—figure 15*). The v-shaped dependence on phylogenetic depth may reflect a tradeoff in which phastCons scores based on deeper alignments have greater power to identify long-lived selected regions (see, e.g. *Appendix 1—figure 10a*), whereas those based on apes are better at identifying regions that are selected in humans but exhibited functional turnover in the deeper phylogeny (*Rands et al., 2014*; see also Section 6.2).

The model fit is also fairly insensitive to the cutoff conservation score used in choosing selection targets, although choosing 5–7% of autosomal sites as targets does appear to yield slightly better fits than other choices (*Appendix 1—figure 16*). Notably, the variance explained for different window sizes is maximized between 5–7% (*Appendix 1—figure 16b and c*); at the higher end of the range of cutoffs from 2% to 9%, the fits of diversity levels along the genome (e.g. *Appendix 1—figure 16d*) and around genomic features (e.g., *Appendix 1—figure 16e*) appear to be slightly worse, and the stratification of observed values by predicted ones spans a smaller range (*Appendix 1—figure 16f*). Among comparisons between models based on 6% and all other cutoffs in the range of 2–9%, only 8 and 9% lead to a statistically significant reduction of fit in windows of 1 Mb (*Appendix 1—figure 33*). Based on these analyses, we use the model with the 6% of autosomal sites with the highest phastCons scores based on the 99-vertebrate alignment in many of our analyses, and refer to this as our *best-fitting phastCons-based model* in both the Main Text and throughout Appendix 1.

The insensitivity of our fits to varying the conservation cutoff can be understood as follows. phastCons estimates the probability that runs of sites belong to conserved segments (*Siepel et al., 2005*). When we reduce the conservation cutoff, shorter segments with high scores tend to expand to include adjacent, lower scoring sites. This results in a high spatial correlation between the conserved sites corresponding to different cutoffs (*Appendix 1—figure 15b*). Given a lower conservation cutoff and longer 'selected' segments, we infer a lower deleterious mutation rate per site (*Appendix 1—figure 16a(ii)*) but a similar deleterious mutation rate per segment (see, e.g. *Appendix 1—figure 15c*), thereby producing similar troughs in diversity around such segments and similar fits overall.

## 4.2 Background selection model based on genic annotations

Next, we consider a model of background selection in which selection targets are chosen based on simple genic annotations, i.e., the exons divided into UTRs and protein coding sequences (CDSs), as well as regions in the immediate vicinity of these sequences controlling transcript regulation: regions 1 kb up- and downstream of transcript start/end, and splice regions 200 bp at the start and end of introns (*Black, 2003*; *Kim et al., 2005*; see Section 2.4 for details). We allow selection parameters to vary among annotations, but find that in the best-fitting model only protein coding and splice regions have non-negligible deleterious mutation rates (for other annotations, $u_d/u_0 < 10^{-6}$).

We also find that this model fits much worse than our best-fitting phastCons-based model (*Appendix 1—figure 17*): the variance in diversity levels it explains is substantially lower across

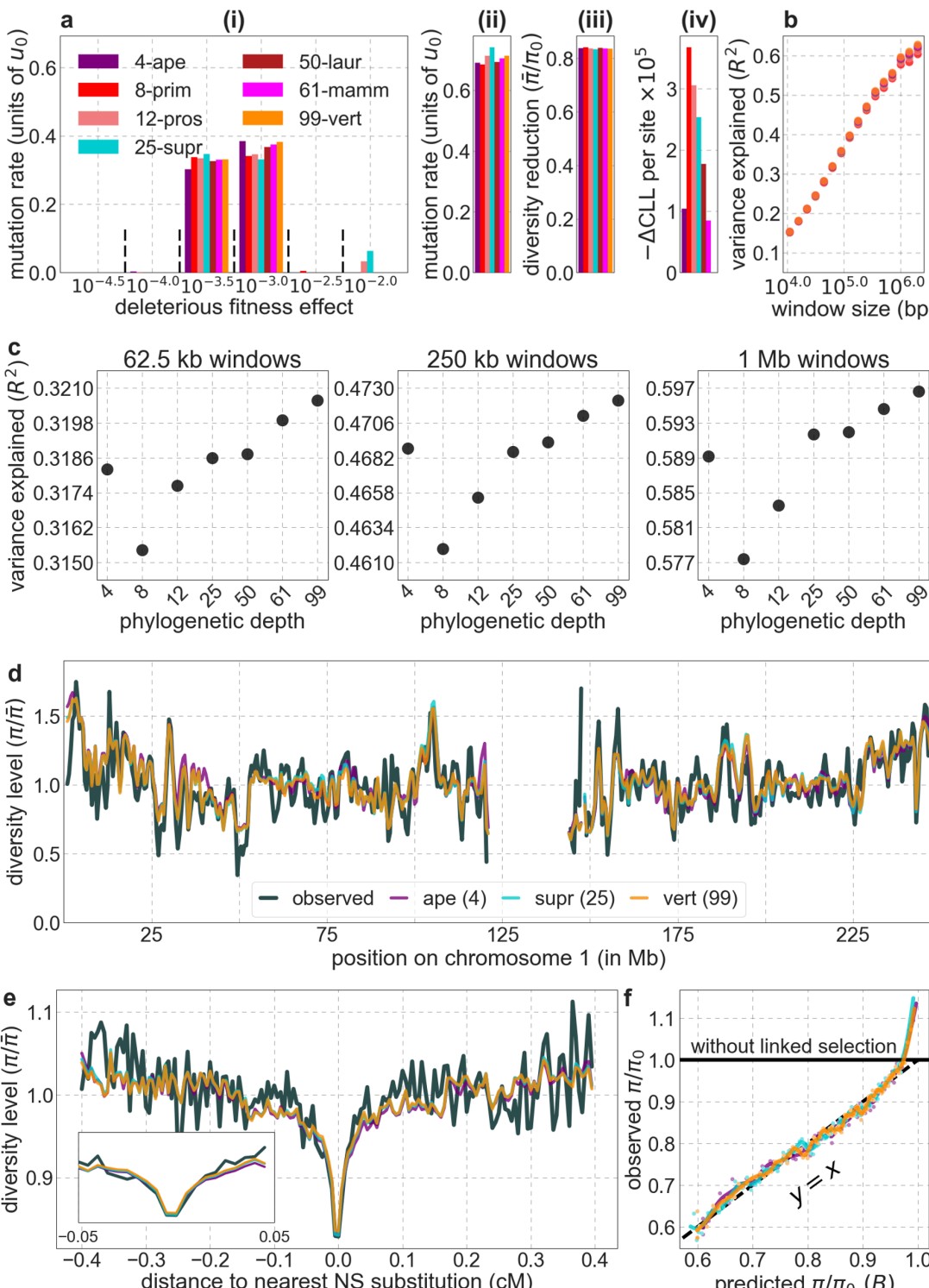

**Appendix 1—figure 14.** Comparison of background selection models based on phastCons conservation scores in phylogenies of difference depths. Shown are results of models based on conservation in four apes, eight primates, 12 prosimians, 25 supra-primates, 50 laurasiatherians, 61 mammals, and 99 vertebrates extending out to lamprey. In all cases, we take the 6% of autosomal sites with the highest phastCons scores (excluding putatively neutral sites) as our targets of selection. Throughout Appendix 1, with the exception of Section 7, we show results using data from YRI. *The panels describe*: (**a**) Parameters and summaries of models (from left to right): (**i**) Estimated distribution of fitness effects, described in terms of the rate of mutations with given selection coefficients. Mutation rates throughout are measured relative to the estimate of the estimated average mutation rate per bp

*Appendix 1—figure 14 continued on next page*

*Appendix 1—figure 14 continued*

per generation in humans, $u_0 = 1.4 \cdot 10^{-8}$ (see Section 5). As detailed in Section 1.5, the inferred distribution of selection coefficients should be interpreted with caution. (**ii**) Estimated total deleterious mutation rate per selected site ($u_d$) measured in units of $u_0$. (**iii**) Estimated autosomal average fold-reduction in neutral diversity levels due to selection at linked sites, i.e., the ratio of average predicted heterozygosity, $\bar{\pi}$, to average predicted heterozygosity in the absence of selection at linked sites, $\pi_0$. (**iv**) The reduction in composite log-likelihood (CLL) per site relative to the model with the highest CLL. Differences in CLL should be interpreted with caution, as diversity levels at putatively neutral sites are not independent. (**b**) The proportion of variance in diversity levels explained ($R^2$) on different spatial scales (measured in non-overlapping contiguous windows). (**c**) Close-up on the variance explained for several window sizes. (**d**) Predicted and observed diversity levels along chromosome 1. Diversity levels are measured in 1 Mb windows, with 0.5 Mb overlap, and are normalized by the mean level (as detailed in *Figure 2*). The results here and in subsequent panels are shown for a subset of depths, including four apes, 25 supra-primates and 99 vertebrates. (**e**) Predicted and observed diversity levels as a function of genetic distance to the nearest human-specific nonsynonymous (NS) substitutions. The plot was generated as detailed in *Figure 3*. Inset shows closeup between –0.05 and 0.05 cM. (**f**) Observed vs. predicted neutral diversity levels across the autosomes. The plot was generated as detailed in *Figure 5*.

different window sizes (*Appendix 1—figure 17b*), its fit to diversity levels along the genome is discernably worse (e.g. *Appendix 1—figure 17c*), and when observed diversity levels are stratified by the model's predictions, they are less calibrated (*Appendix 1—figure 17e*). The genic model does do reasonably well at predicting how diversity levels drop with genetic distance around nonsynonymous substitutions (e.g. *Appendix 1—figure 17d*). The generally poorer fit as well as the reasonably good fit around nonsynonymous substitutions can be understood in terms of the overlap between our simple genic annotations and direct measures of constraint (*Appendix 1—figure 18*). Namely, the genic annotations miss most constrained sites, which are intronic or intergenic (*Appendix 1—figure 18b*), but most protein coding regions (CDSs) are constrained (*Appendix 1—figure 18a*) explaining why models including them as an annotation perform well near them.

## 4.3 Background selection models separating conserved exonic and non-exonic sites

While background selection models based on simple genic annotations do worse than those based on phylogenetic conservation, using such annotations in conjunction with conservation could allow for improved fits. Notably, it is often argued that purifying selection in protein coding regions is

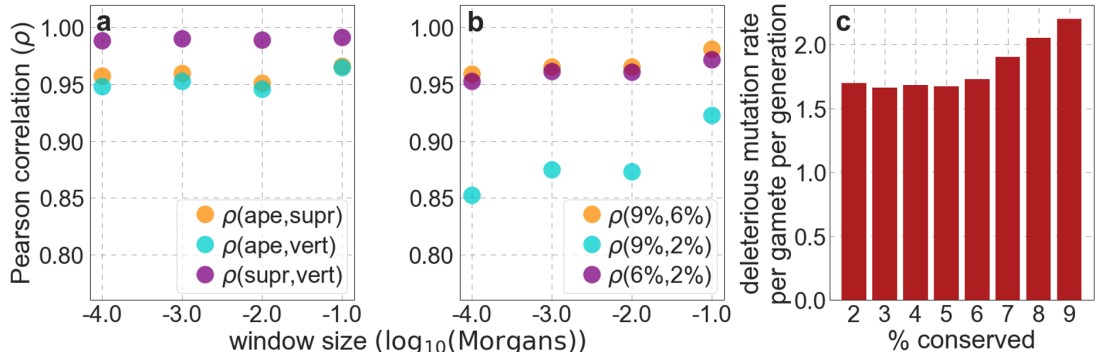

**Appendix 1—figure 15.** The spatial distribution of putatively selected sites remains similar when we vary the phylogenetic depth of the alignment used to infer conservation (shown in **a**), and the proportion of sites with the highest conservation scores included (in **b**). We compare two choices of selection targets at a time, and show the Pearson correlations ($\rho$) between the numbers of putatively selected sites among windows of different genetic lengths (measured in Morgans). The range of window sizes roughly corresponds to the spatial scales over which selection affects linked neutral diversity for the estimated range of selection effects. When we vary the phylogenetic depth, we use the 6% of autosomal sites with the highest phastCons scores, and when we vary the conservation cutoff, we use phastCons scores based on the 99-vertebrate alignment. (**c**) The deleterious mutation rate per gamete per generation inferred as a function of assumed proportion of selected sites in autosomes.

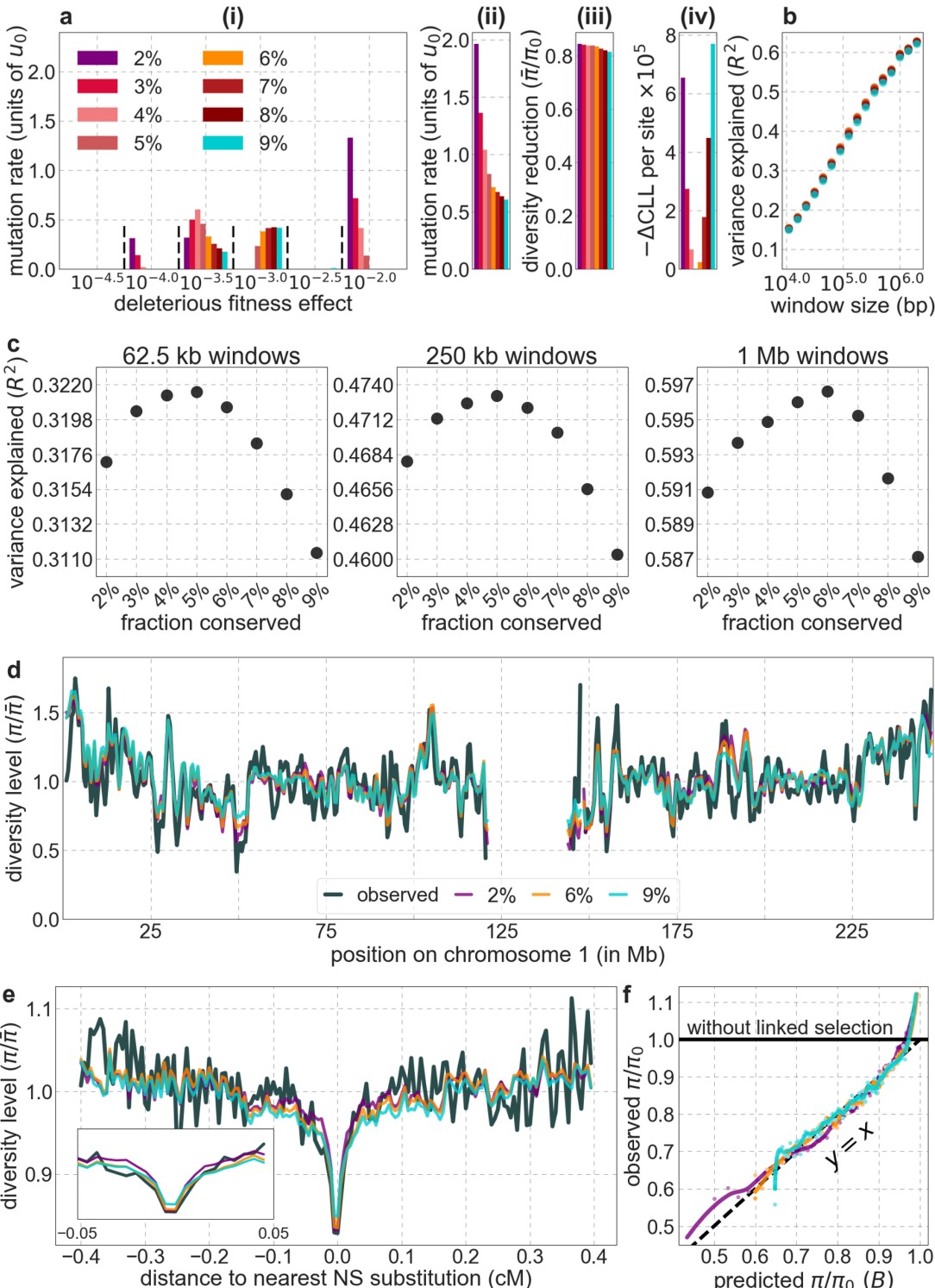

**Appendix 1—figure 16.** Comparison of background selection models based on phastCons scores using different proportions of autosomal sites as selection targets. In all cases considered, we rely on conservation in 99 vertebrates. Otherwise, all panels are as described in *Appendix 1—figure 14*.

stronger than in functional non-coding regions (*Kellis et al., 2014*; *Rands et al., 2014*); if this were true, then allowing them to have different selection parameters could result in better fits. To examine this possibility, we fit a model with two types of selection target: exonic (i.e. segments combining CDSs and UTRs) and non-exonic conserved sites (see details in Section 2.4).

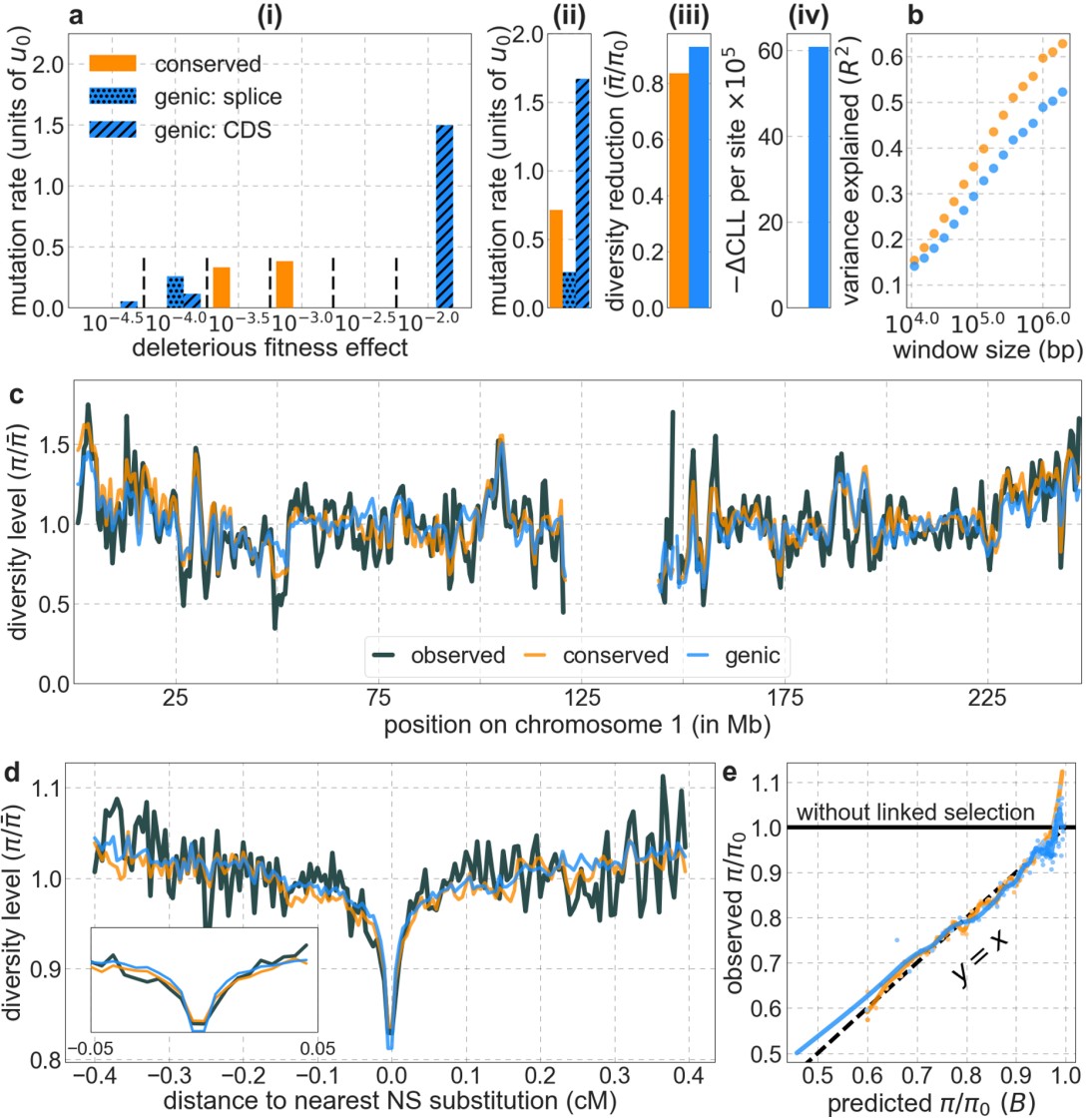

**Appendix 1—figure 17.** The background selection model based on simple genic annotations fits worse than our best-fitting phastCons-based model. All the panels are as described in *Appendix 1—figure 14* (but with the hatch-marked blue bars in **a** (**i**) and (**ii**) corresponding to different annotations of the genic model).

We infer a higher deleterious mutation rate and stronger selection in exonic compared to non-exonic sites (*Appendix 1—figure 19a*), although we note that our estimates of selection parameters could be affected by thresholding (see Section 1.5). The total deleterious mutation rate per gamete is similar in models with and without the exonic/non-exonic division ($U = 1.6$ and $U = 1.73$ per gamete per generation, respectively), but the (weighted) average selection effect is greater in the model with the division ($\bar{s} = 1.71 \times 10^{-3}$ vs. $\bar{s} = 6.8 \times 10^{-4}$ for the models with and without division, respectively), primarily due to stronger selection in conserved exonic sites. Overall, despite affording additional parameters, dividing conserved sites into exonic and non-exonic has little effect on our fits (*Appendix 1—figure 19b–e*).

Regardless of whether we separate exonic and non-exonic conserved sites, most of the reduction in diversity levels is caused by selection in non-exonic regions. Weakly selected mutations cause a large reduction in neutral diversity levels over short genetic distances, whereas strongly selected mutations cause a weak reduction over long genetic distances; but the integral reduction in diversity levels due to weak and strong selection on a given set of deleterious mutations end up roughly equivalent (*Hudson, 1994*). This property allows us to use estimates of the total deleterious

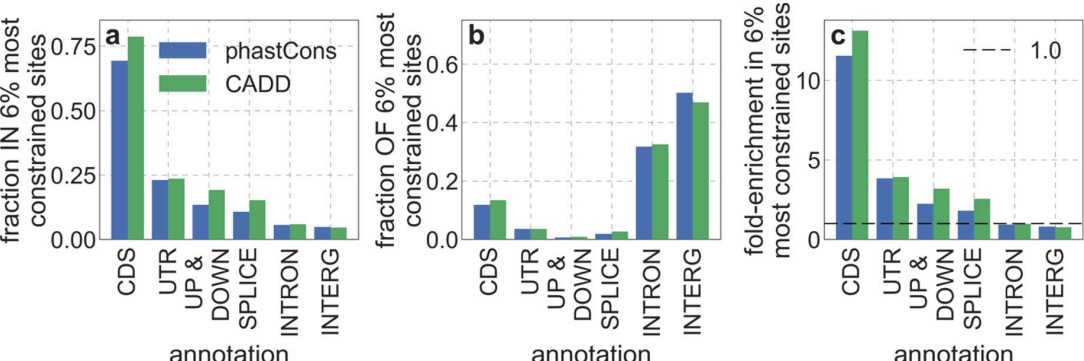

**Appendix 1—figure 18.** The relationship between simple genic annotations and our main measures of constraint. Specifically, we examine the overlap of the 6% of autosomal sites with the highest phastCons or CADD scores with the genic annotation detailed in the text; we added intronic (INTRON) and intergenic (INTERG) annotations for completeness. (**a**) The fraction of each genic annotation within the 6% most constrained sites. (**b**) The fraction of the 6% most constrained within each genic annotation. (**c**) Enrichment of genic annotations in the 6% most constrained sites, i.e., the ratio of their proportion among constrained and all autosomal sites.

mutation rates in conserved exonic and non-exonic regions as a rough measure of their proportional effects on neutral diversity levels, despite differences in selection effects in these regions. These estimates suggest that ~80% of deleterious mutations occur in non-exonic regions, indicating that they account for most of the reduction in linked neutral diversity (e.g. in the model with the top 6% of phastCons scores, ~84% of selected sites and ~76% of deleterious mutations are non-exonic; with the top 6% of CADD scores, ~83% of selected sites and ~85% of deleterious mutations are non-exonic; also see discussion in Section 4.6).

Given that the bulk of deleterious mutations exerting background selection occur in non-exonic regions, it is not surprising that a model including only conserved non-exonic sites fits the data only slightly worse than a model including all conserved sites as targets of selection (*Appendix 1—figure 20*). By the same token, it is not surprising that a model including only conserved exonic sites fits the data substantially worse than models with either conserved non-exonic or all conserved sites as targets of selection (*Appendix 1—figure 20*). Moreover, the estimate of the deleterious mutation rate per site in the exonic model is much higher than in the other two (*Appendix 1—figure 20a(ii)*).

It is somewhat surprising that the model based on conserved exonic sites alone fits the data as well as it does (*Appendix 1—figure 20b and c*). This can be understood by noting that the spatial distribution of conserved exonic sites and of all conserved sites are fairly highly correlated (*Appendix 1—figure 21*). Given similar spatial distributions of selected sites, the distribution of background selection effects in the model with all conserved sites can be approximated by having a higher deleterious mutation rate per site at the fewer selected sites in the exonic model. These considerations explain why we infer a similar (albeit lower) average reduction in diversity levels but a substantially higher deleterious mutation rate in the exonic model (*Appendix 1—figure 20a(ii) and (iii)*). They also help to explain differences between our inferences and those of *McVicker et al., 2009*, notably their implausibly high estimate of the deleterious mutation rate given that their main model assumes selection only at conserved exonic sites (see Main Text and Section 4.6).

## 4.4 Background selection models based on other annotations

We consider two additional widely-used functional annotations as putative background selection targets. First, we rely on the expanded encyclopedias of DNA elements (ENCODE) annotations of candidate cis-regulatory elements (cCREs), including enhancer-like signatures (ELS), promoter-like signatures (PLS), CTCF-bound (CTCF) and poised/DNAse-hypersensitive (H3K4me3) assayed in 25 Tier 1 a biosamples (*Moore et al., 2020*), alongside protein coding sequences (CDSs) (see Sections 2.4 and 2.6 for data sources and definition of elements). ENCODE cCREs attempt to capture the diverse repertoire of regulatory elements across cell types that control gene expression in different cellular and biological contexts. They are based on a large set of epigenomic assays, including ChIP-seq measuring the occupancy of histone marks associated with both activation and repression of gene expression, pulldown of DNA-bound transcription factors, and DNA accessibility measured in

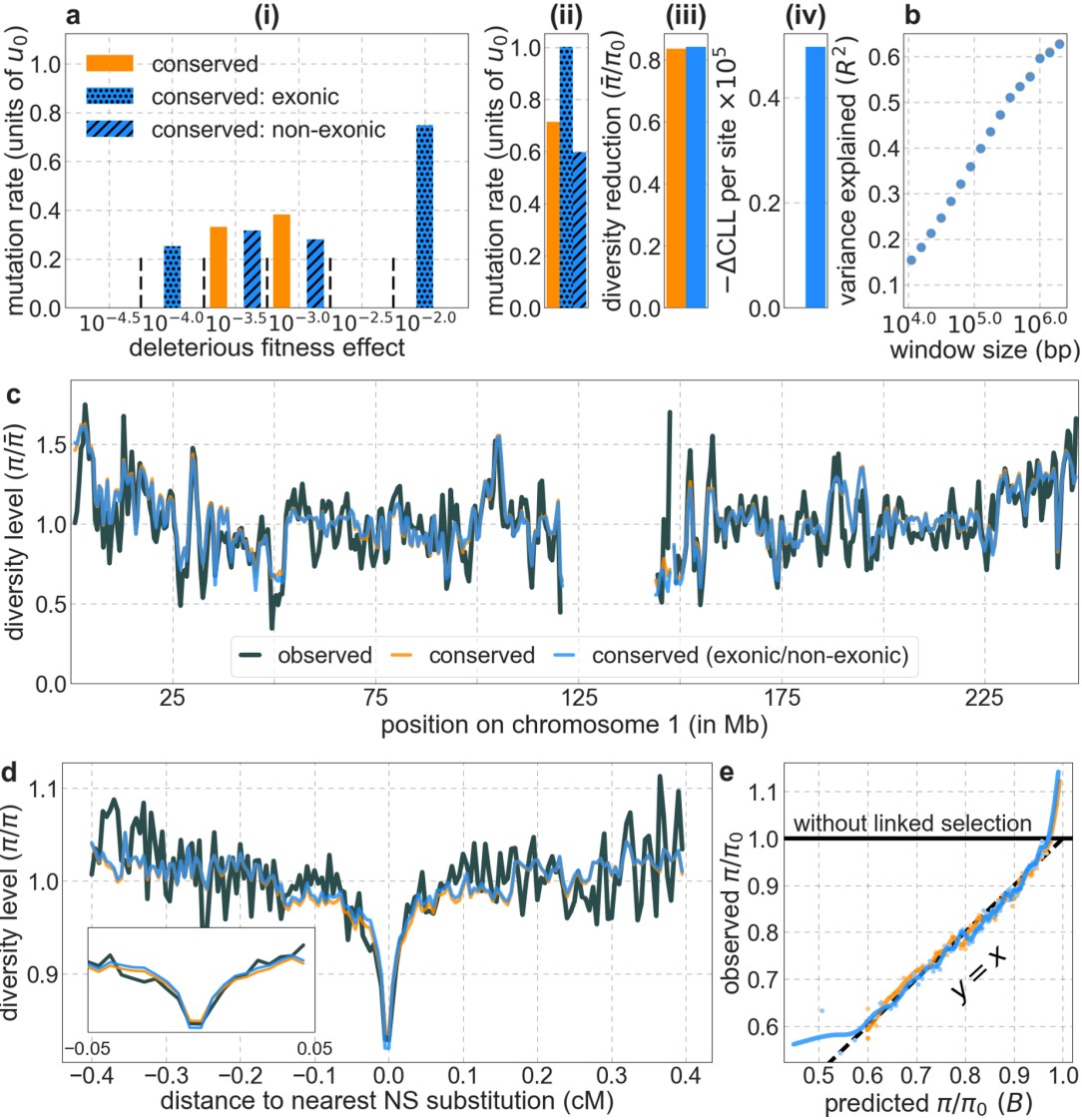

**Appendix 1—figure 19.** Dividing conserved sites into exonic and non-exonic sets leads to different estimates of selection parameters in each, but to little improvement in fit compared to the model based on conservation alone. Our set of conserved sites consists of the 6% of sites with the highest phastCons scores in the 99-vertebrate alignment (see Section 4.1). All panels are as described in *Appendix 1—figure 14*. Because of thresholding (Section 1.5), the model based on conservation alone is not formally nested in the one with the division into exonic and non-exonic sets, explaining how its maximum composite-likelihood can be slightly greater.

terms of DNAse sensitivity. Since we infer the majority of autosomal sites under purifying selection to be non-exonic (see Section 4.3), we reason that some combination of cCREs may substantially overlap these sites. Importantly, cCRE annotations may allow us to better partition non-exonic regions into sub-classes of sites experiencing different selection strengths. We define our choices of selection targets (other than CDSs) by grouping cCRE in two alternative ways. In one, we take the union of cCREs of a given type over all 25 biosamples. In the other, we divide cCREs of a given type into those identified in few (≤ median number) or in many (> median number) biosamples (in practice, most cCREs included in the first set are cell-type specific whereas most of those in the second are found in a few to all cell-types). The model in which cCREs of a given type are split performs slightly better, presumably because of the additional degrees of freedom. Both models, however, fit the data substantially worse than either of our best-fitting models (*Appendix 1—figure 22*). The poor fit accords with the modest overlap between cCREs and our estimates of constraint sites (*Appendix 1—figure 23*). Moreover, PLSs, the cCREs that are most highly enriched in constrained sites (*Appendix 1—figure 23a and c*) are inferred to have a negligible deleterious mutation rate.

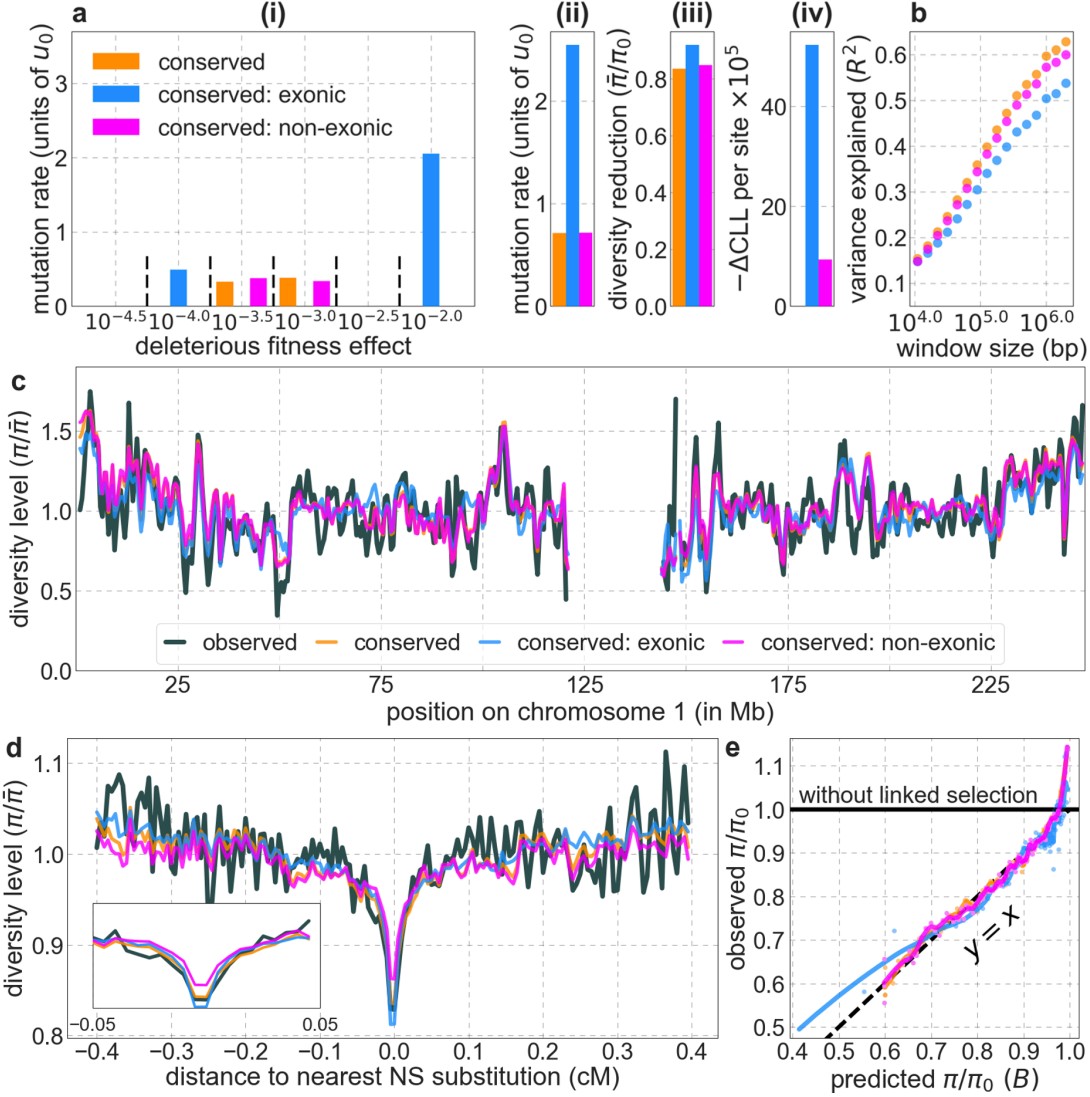

**Appendix 1—figure 20.** Comparison of background models using exonic, non-exonic and all conserved sites as targets of selection. Our set of conserved sites consists of the 6% of sites with the highest phastCons scores in the 99-vertebrate alignment (see Section 4.1). All panels are as described in *Appendix 1—figure 14*.

Next, we consider Combined Annotation-Dependent Depletion (CADD) scores (*Kircher et al., 2014*; *Rentzsch et al., 2019*). CADD scores predict the 'deleteriousness' of every point mutation in the genome. They are generated by using machine learning to integrate information from a diverse

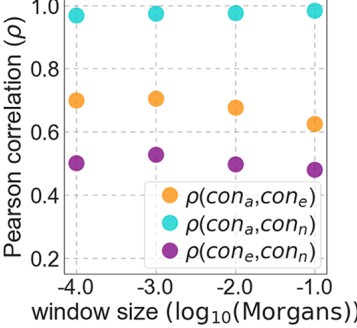

**Appendix 1—figure 21.** The spatial correlations of exonic, non-exonic and all conserved sites for varying window sizes ('con$_e$', 'con$_n$' and 'con$_a$', respectively).

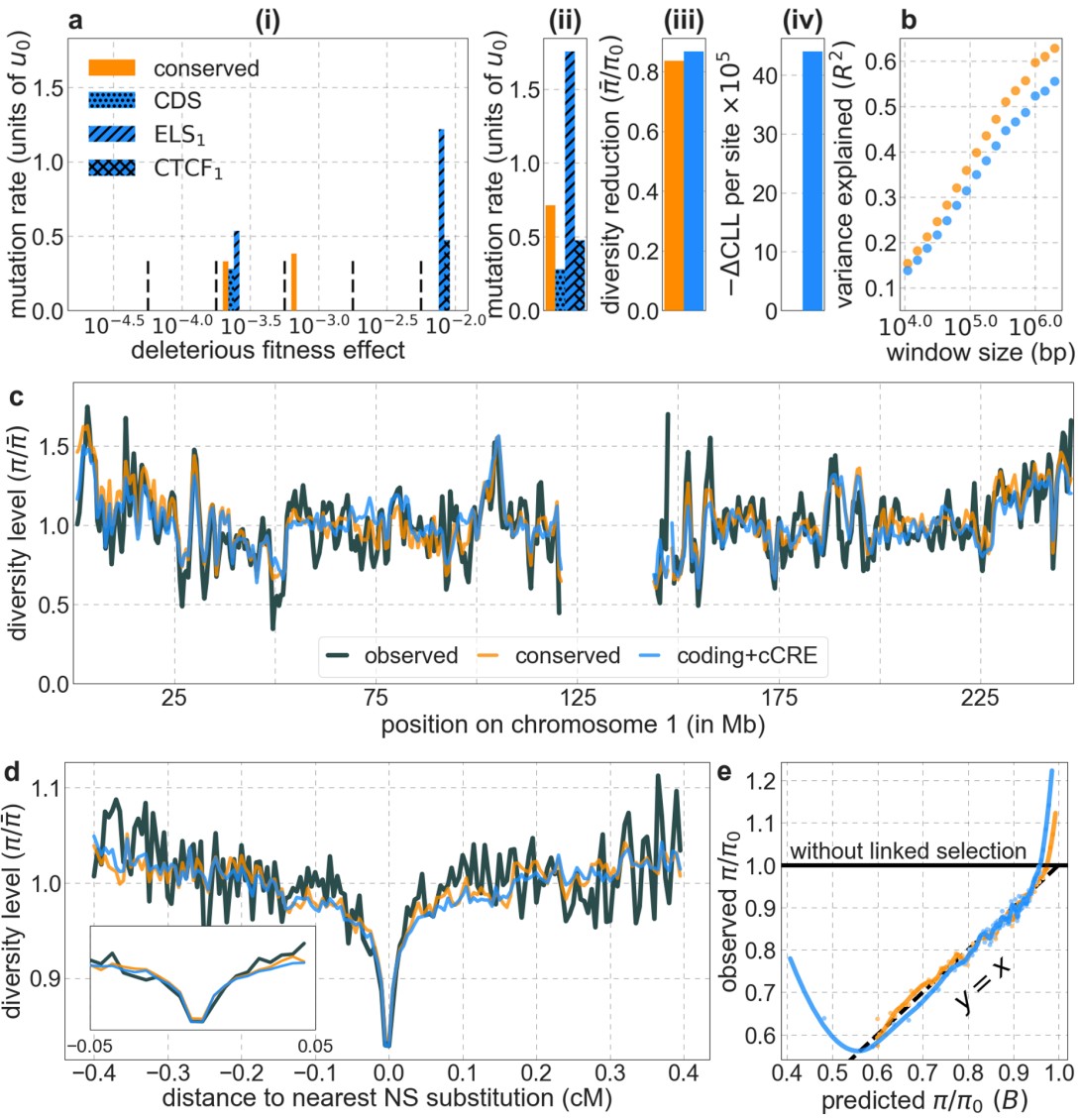

**Appendix 1—figure 22.** The model based on the ENCODE annotations of cCRE fit the data substantially worse than our best-fitting phastCons-based model using conservation in the 99-vertebrate alignment (conserved). The results shown correspond to the model in which we split each type of cCREs into those that occur in few (subscript 1) and many biosamples (subscript 2). We infer a non-negligible deleterious mutation rate (i.e. $u_d/u_0 > 0.01$) in 2 of the 8 cCRE-based putative selection targets: enhancer like sequences and CTCF binding sites identified in few biosamples, $ELS_1$ and $CTCF_1$ respectively, as well as in protein coding regions (CDS). All the panels are as described in **Appendix 1—figure 14**.

set of annotations (122 annotations in version 1.6), such as measures of phylogenetic conservation (including phastCons scores based on the 99-vertebrate alignment), predictions of regulatory elements (including many of the assays used for constructing the ENCODE cCREs), genic annotations (including those described in Sections 2.7 and 4.2) and predicted functional consequences of variants in protein coding sequences. The algorithm is trained using the depletion of 14.7 million high-frequency (>95%) derived alleles (based on 1000 Genomes Data) relative to 14.7 million simulated variants with the same genomic distribution as the criterion for 'deleteriousness'. While the standard CADD scores (version 1.6) incorporate the *McVicker et al., 2009* map of background selection effects as one of the annotations, we use a version in which this annotation was excluded in order to avoid circularity (see Section 2.5). We use the maximal score at each site (corresponding to the most deleterious of three possible point mutations), and, for comparison with our best-fitting phastCons-based model (Section 4.1), we begin by considering the 6% of autosomal sites with the highest CADD scores (excluding putatively neutral sites) as targets of selection.

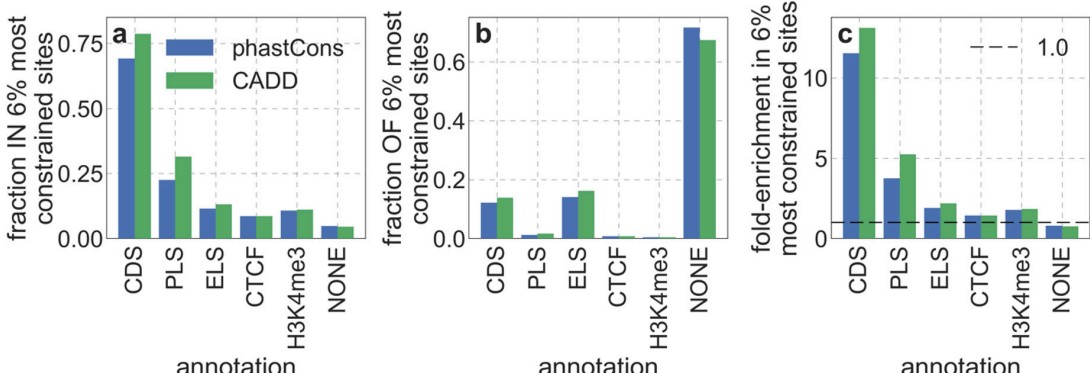

**Appendix 1—figure 23.** The relationship between ENCODE cCRE annotations and our main measures of constraint. Specifically, we examine the overlap of the 6% of autosomal sites with the highest phastCons or CADD scores with promoter like sequences (PLS), enhancer like sequences (ELS), CTCF-bound (CTCF), poised/DNAse-hypersensitive (H3K4me3), as well as sites that are not in any of these annotations (NONE). (**a**) The fraction of each cCRE annotation within the 6% most constrained sites. (**b**) The fraction of the 6% most constrained within each cCRE annotation. (**c**) Enrichment of cCRE annotations in the 6% most constrained sites, that is, the ratio of their proportion among constrained and all autosomal sites.

Despite incorporating many sources of information beyond phylogenetic conservation, and doing better than phastCons scores at predicting functional consequences of variants at a single site resolution (*Kircher et al., 2014*), the model based on CADD scores offers only a minor improvement over our best-fitting phastCons-based model (*Appendix 1—figure 24*). For example, the model based on CADD scores explains 59.9% of the variance in diversity levels in 1 Mb windows compared to 59.7% for the model based on phastCons scores, although this difference and differences in other window sizes are not statistically significant (see *Appendix 1—figure 32* and Section 6.2). The little improvement is not that surprising, given that phylogenetic conservation is the annotation most correlated with CADD scores genome-wide (*Kircher et al., 2014*), and that the spatial distributions of sites with top CADD and phastCons scores are highly correlated on the spatial scales that impact background selection effects (*Appendix 1—figure 25*).

The fit of models based on CADD scores is fairly insensitive to the proportion of sites included as selection targets, with proportions of 5–7% yielding slightly better fits than other choices (*Appendix 1—figure 26*). This insensitivity and the increase in estimates of the deleterious mutation rate per site with decreasing proportion of sites used as selection targets (*Appendix 1—figure 26a(ii)*) can be explained in the same way that we explained similar observations for models based on phastCons scores (Section 4.1).

Based on the analyses in *Appendix 1—figure 24* and *Appendix 1—figure 26*, we refer to the model with the 6% of autosomal sites with the highest CADD scores as our *best-fitting CADD-based model*, and use it in most of our analyses here and in the Main Text. While the differences in fit of our best-fitting CADD-based and phastCons-based models are minor, the improved predictions of CADD compared to phastCons scores at the single site resolution substantially affects our estimates of the deleterious mutation rate based on evolutionary rates and thus their agreement with estimates based on the effects of background selection (see Main Text and Section 5).

## 4.5 Models with selective sweeps

Next, we examine whether models that include both background selection and selective sweeps fit the data better than models with background selection alone. Our inference should be able to tease apart the effects of sweeps, primarily because these effects, unlike those of background selection, are centered around the locations of substitutions. This feature should hold true for hard, partial or soft sweeps (*Hermisson and Pennings, 2005*; *Przeworski et al., 2005*; *Pennings and Hermisson, 2006a*; *Pennings and Hermisson, 2006b*; *Coop and Ralph, 2012*; *Berg and Coop, 2015*), so long as they result in substitutions and have a substantial effect on diversity levels (SOM Section D in *Elyashiv et al., 2016*). Indeed, previous work that applied a similar methodology to data from *Drosophila melanogaster* was able to identify and quantify distinct signatures of background selection and sweeps alongside one another (*Elyashiv et al., 2016*).

**Appendix 1—figure 24.** The model based on CADD scores offer little improvement over the model based on phastCons scores (based on the 99-vertebrate alignment). In both cases, we take the 6% of sites with the highest scores. All the panels are as described in *Appendix 1—figure 14*.

We consider a variety of models characterized by different sets of putatively selected sites. For background selection, we consider the two sets used in our best-fitting models based on phastCons and CADD scores. We also consider several choices for targets of *positive selection*, i.e., for sweeps,

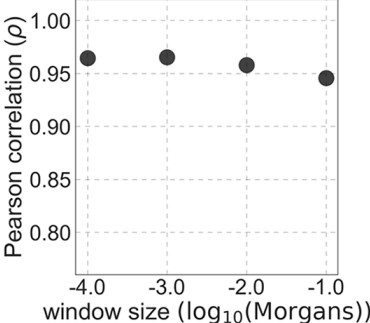

**Appendix 1—figure 25.** The spatial correlation between the 6% of sites with the highest CADD and phastCons scores.

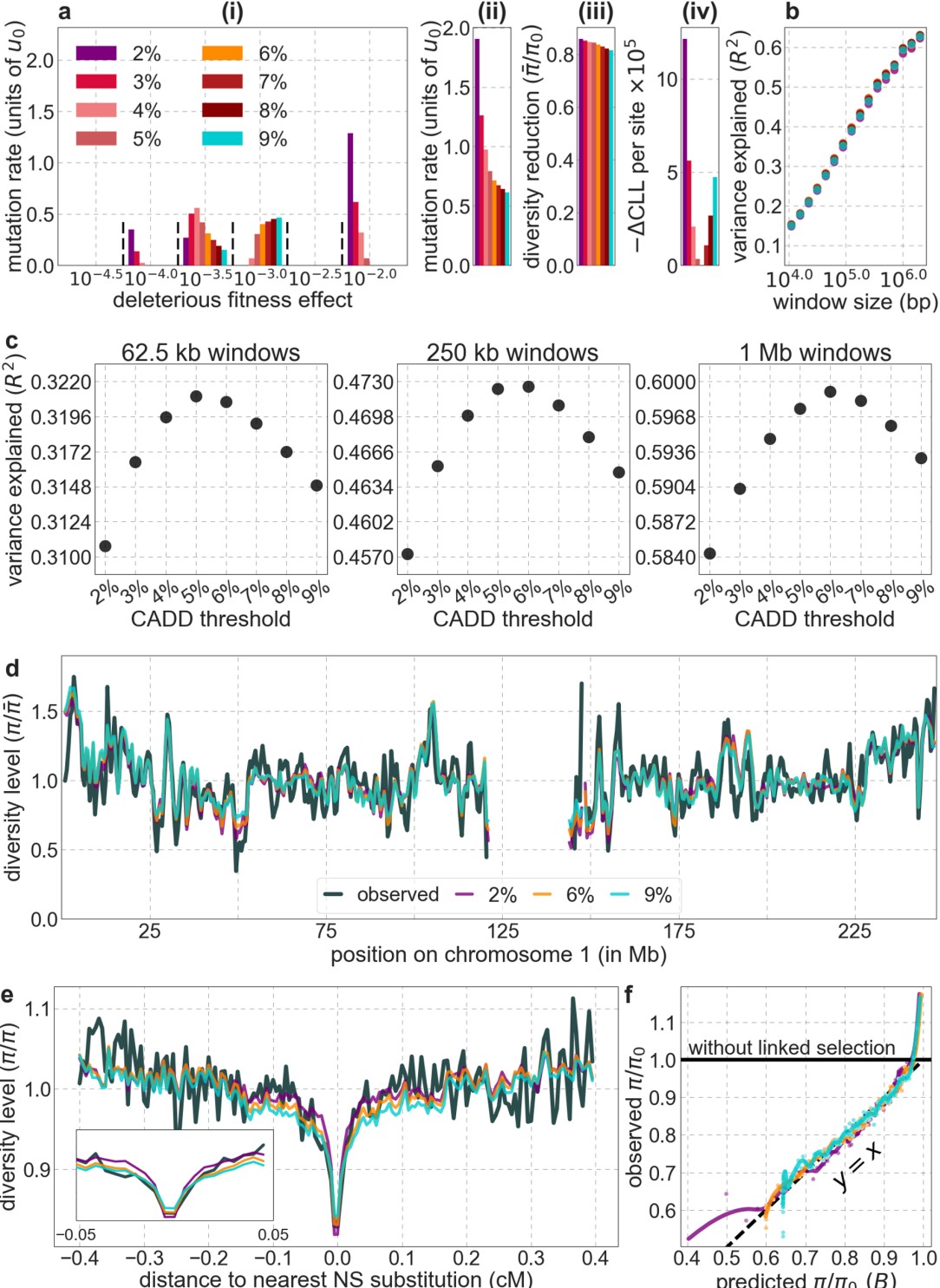

**Appendix 1—figure 26.** Comparison of background selection models based on CADD scores using different proportions of autosomal sites as selection targets. All panels are as described in *Appendix 1—figure 14*.

corresponding to different kinds of substitutions that we infer to have occurred on the human lineage from the common ancestor with chimpanzees (see Section 2.7). Notably, we consider models that include the set of all nonsynonymous substitutions paired with either of the two sets for background selection. We also consider models with the substitutions that have occurred at sites with the top 2%, 3%, …, 9% of phastCons or CADD scores, where in each case we separate substitutions into sets of nonsynonymous and other, and pair that choice with the corresponding set for background selection

(i.e. based on phastCons or CADD scores). For each of these choices, we infer the set of substitutions on the human lineage in two ways, either comparing the estimated human-chimpanzee ancestral genome (*Paten et al., 2008*) with the human reference genome (hg19) or with a population (YRI or CEU) sample of human genomes (see Section 2.7). We perform the inference for all of these models (18 in total) using the same grid of selection coefficients for each of the sets of selected sites, and data from either YRI or CEU. *In all cases, our estimate of the fraction of beneficial substitutions, α, is essentially 0* (< 10⁻⁹). We do not show the results because they are indistinguishable from those for the corresponding models with background selection alone (i.e., see *Appendix 1—figures 16 and 26*).

We also consider models with sweeps alone. *Appendix 1—figure 27* shows the results for a subset of these models, including the best-fitting one (e.g. based on variance explained). These

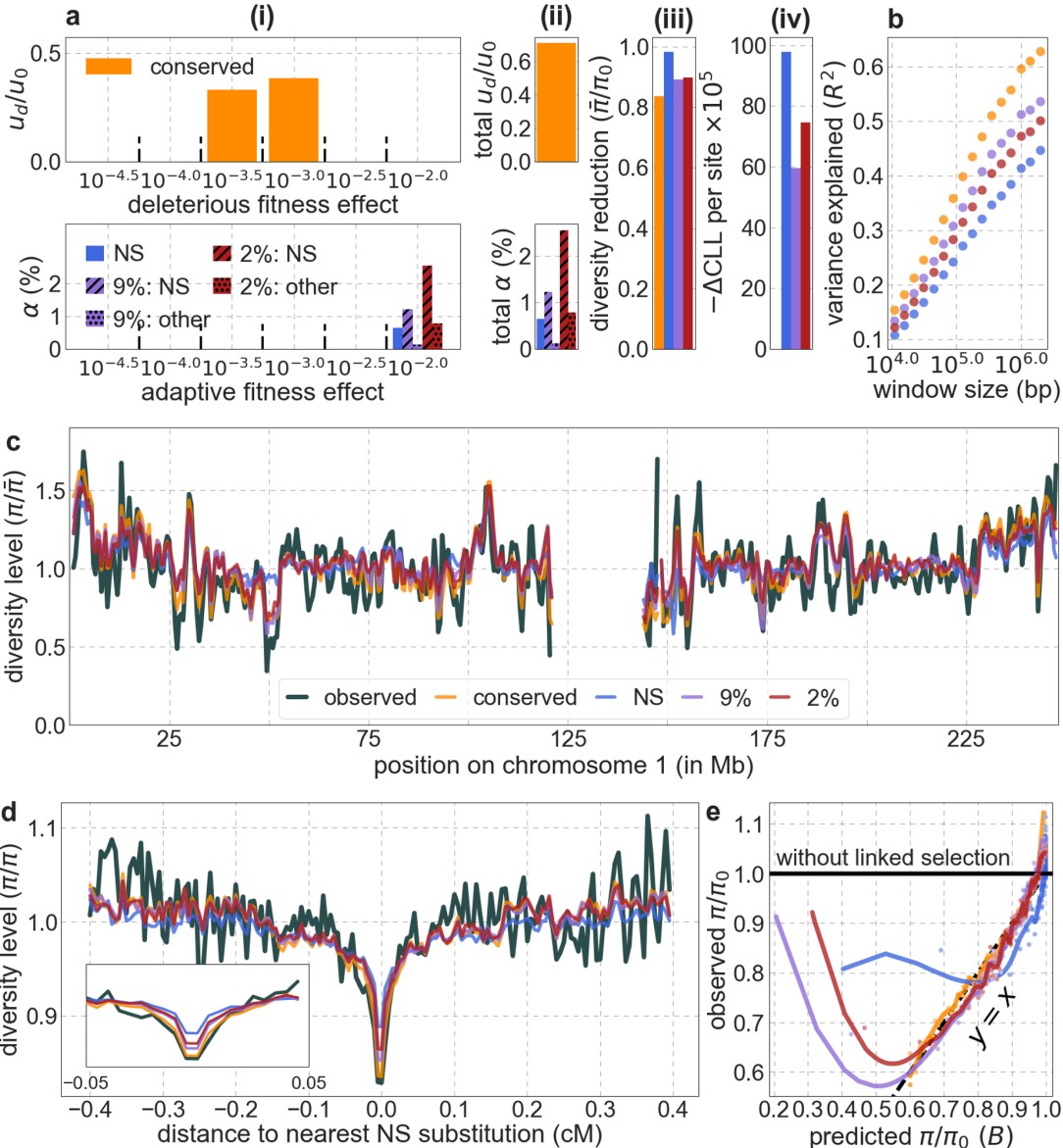

**Appendix 1—figure 27.** Models with sweeps alone fit substantially worse than models with background selection alone. Shown are the results for sweep models based on either: all nonsynonymous substitutions (NS); nonsynonymous and other substitutions at sites within the top 9% of phastCons scores (9%: NS/other); or nonsynonymous and other substitutions at sites within the top 2% of phastCons scores (2%: NS/other). For comparison, we also show the results of our best-fitting phastCons-based background selection model (conserved). The panels are as described in *Appendix 1—figure 14*, with the exception of the bottom halves of panels **a** (**i**) and (**ii**), which show the proportions of substitutions that are estimated to be adaptive for a given selection coefficient (**i**) or in total (**ii**), for different annotations and sweeps models.

models fit the data substantially worse than those with background selection alone, as seen by each of our measures (interestingly, even when considering the reduction in diversity levels around nonsynonymous substitutions; *Appendix 1—figure 27d*). Sweep models do account for substantial variance in diversity levels, but given that they add nothing to a model of background selection alone yet fit much worse, this is plausibly because they approximate some of the effects of background selection. Notably, both background selection and sweeps cause reductions in diversity levels near selected sites, and the densities of sites that give rise to background selection and sweeps in the corresponding models are spatially correlated along the genome (*Appendix 1—figure 28*). Moreover, the sweep models that fit the data best are those that rely on substitutions whose spatial distributions are the most highly correlated with the distributions of selection targets in our best-fitting background selection models (e.g. compare the fits and correlations for the models based on substitutions in the most conserved 2% and 9% of sites in *Appendix 1—figures 27 and 28*). Taken together, the evidence presented here supports previous studies (*Coop et al., 2009*; *Hernandez et al., 2011*) indicating that sweeps had little effect on current diversity levels and that background selection is the dominant mode of linked selection in humans.

## 4.6 Comparison with previous work by McVicker et al

For completeness, we conclude by comparing our inferences about the effects of background selection with those of *McVicker et al., 2009*. The McVicker et al. study was done more than a decade ago, before genome-wide resequencing polymorphism data were available. Instead, they ingeniously used a five-primate alignment of ~4.7 million putatively neutral sites, relying on incomplete lineage sorting between human, chimpanzee and gorilla in order to learn about variation in the effective population size along the genome of the common ancestor of humans and chimpanzees. We rely on diversity levels in samples of 108 individuals at ~653 million putatively neutral sites (Section 2.1). Similar to this study, they relied on conservation scores and estimates of neutral substitution rates based on multiple sequence alignments, but they based themselves on the genomes of 15 placental mammals when we have 99 aligned vertebrate genomes at our disposal (Section 2.2). Lastly, they used a genetic map based on LD patterns (*Myers et al., 2005*), whereas we rely on genetic maps based on ancestry switches in African Americans (*Hinch et al., 2011*). The McVicker study also differed in several aspects of the methodology. Notably, McVicker et al. did not incorporate selective sweeps into their models, and were therefore unable to exclude the possibility that sweeps had made a substantial contribution to their inferred effects of background selection (*McVicker et al., 2009*). Also, McVicker et al. assumed that selection coefficients are distributed exponentially, whereas we assumed a more flexible (non-parametric) distribution on a grid. Despite limitations, the McVicker et al. maps of the effects of background selection capture substantial variation in diversity levels along the human genome (*Appendix 1—figure 29a* and Figure 7 in *McVicker et al., 2009*).

Nonetheless, our maps of the effects of background selection fit the data substantially better than the map from McVicker et al., both quantitatively and qualitatively (*Appendix 1—figure 29*).

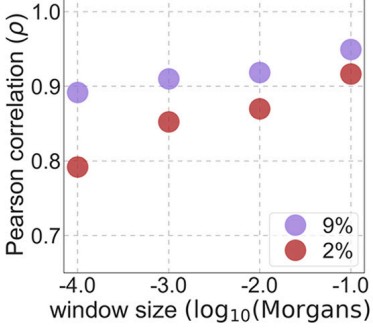

**Appendix 1—figure 28.** The spatial correlation between targets of selection in sweep models and in our best-fitting phastCons-based background selection model. Results shown for sweep models based on human-specific substitutions at sites within the top 2% and 9% of phastCons scores (see text for details).

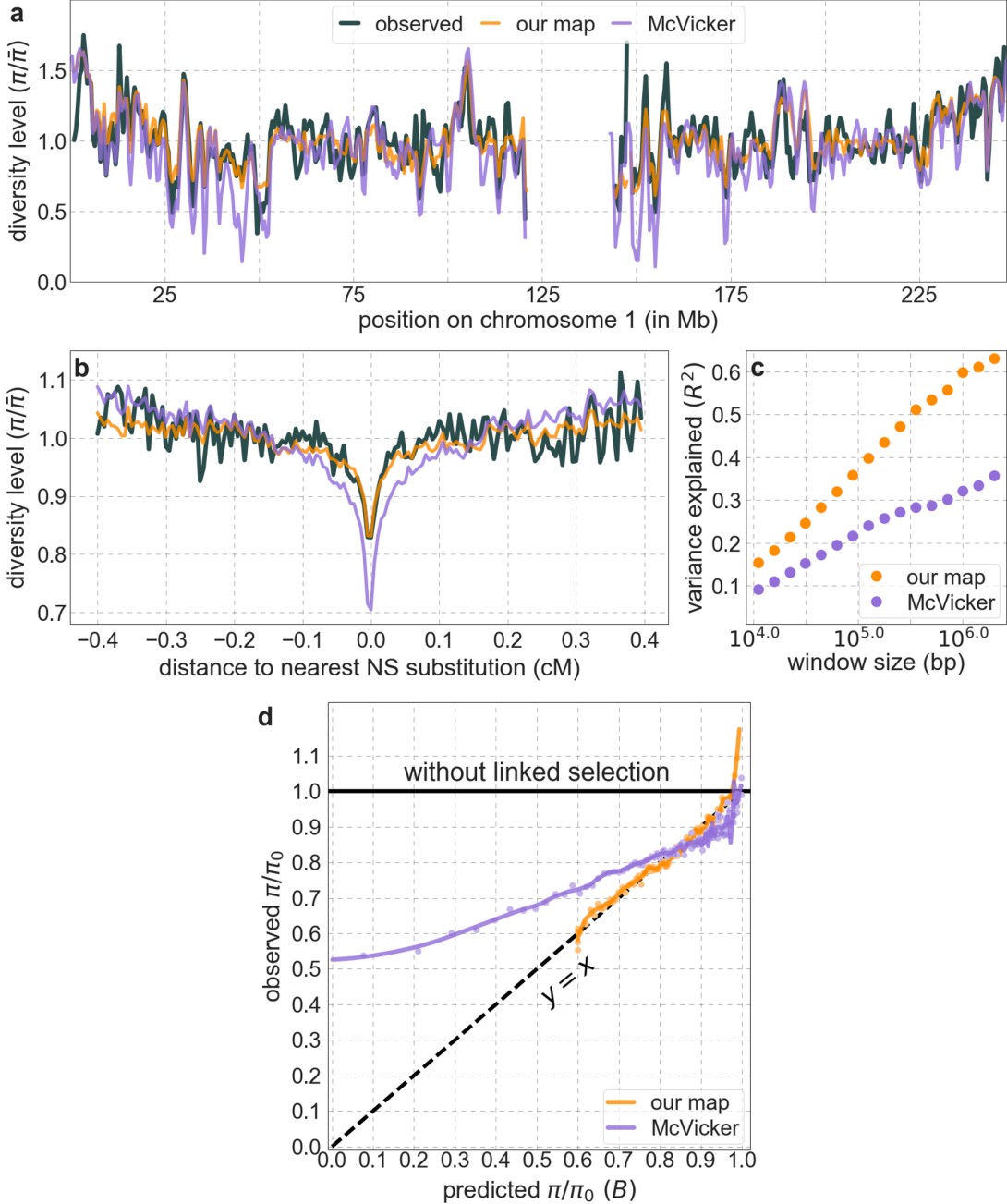

**Appendix 1—figure 29.** Our maps of the effects of background selection fit the data much better than the maps from *McVicker et al., 2009*. Shown are the results for our best-fitting CADD-based model. All panels are as described for the corresponding ones in *Appendix 1—figure 14*.

They explain considerably greater proportions of the variance in diversity levels across window sizes (*Appendix 1—figure 29c*); for example, they explain ~60% compared to ~32% of the variance on the 1 Mb scale. Our predictions are well calibrated, whereas those of McVicker et al. are not (*Appendix 1—figure 29d*). Our predictions also do substantially better at capturing diversity patterns near specific genomic features, as illustrated by the fit to diversity levels around nonsynonymous substitutions (*Appendix 1—figure 29b*). The relatively poor quantitative fit of the McVicker et al. predictions around synonymous and nonsynonymous substitutions (*Hernandez et al., 2011*) was used to argue that the effects of background selection could be more pronounced around synonymous

than nonsynonymous substitutions, thereby masking the effects of selective sweeps (***Enard et al., 2014***). In this regard, the close fit of our predictions helps to refute one of two arguments for a residual, important role of selective sweeps.

We turn to the second argument, regarding estimates of the deleterious mutation rate, next. Our work and that of McVicker et al. differ markedly in our inferences about the rate and genomic distribution of deleterious mutations causing background selection in humans. In fact, the main problem in interpreting the McVicker et al. findings in terms of background selection alone is that they are based on an estimated deleterious mutation rate of $7.4 \times 10^{-8}$ per generation at their 'conserved exonic' sites (defined as sites within the top 5.3% of conservation scores in segments that overlap exons, accounting for ~1.1% of euchromatic autosomal sites) – more than fivefold higher than current estimates of the total mutation rate per site (see next Section). In contrast, as we detail in the next section, our estimates of the deleterious mutation rate per selected site are quite plausible ($1.00 \times 10^{-8}$ per generation for both of our best-fitting models based on phastCons and CADD scores; ***Figure 4*** in Main Text). The results of McVicker et al. further suggest that background selection arises predominantly from deleterious mutations in the 'conserved exonic' regions covering ~1.1% of euchromatic autosomal sites (i.e. they estimate ~2.3 mutations per gamete per generation in such regions in exons compared to ~0.1 elsewhere). In contrast, our results suggest that background selection arises mostly from deleterious mutations at non-exonic sites (i.e. from ~1.22 and~1.27 mutations per gamete per generation in non-exonic compared to ~0.38 and~0.23 mutations in exonic sites in the models based on phastCons and CADD scores, respectively). Notably, in our best-fitting models, these deleterious mutations occur in 6% of autosomal sites as opposed to only ~1% in the McVicker et al. model. Having the effects of background selection arise from deleterious mutations in a substantially greater fraction of the genome largely explains why our estimates of the deleterious mutation rate are much lower and much more plausible (***Figure 4*** and ***Appendix 1—figure 30***).

## 5. Assessing estimates of the deleterious mutation rate

Here, we consider the plausibility of the deleterious mutation rate that we estimated by fitting models of background selection. First, we consider the total mutation rate per site in humans, which provides an upper bound on the deleterious mutation rate. Second, we rely on the reduction in substitution rates at our selection targets relative to putative neutral sites to obtain estimates of the proportion of mutations at selected sites that are deleterious. These estimates should be largely independent of those that we obtained by fitting background selection models, and can therefore be used to evaluate the plausibility of the latter. Lastly, we briefly consider to what extent we should expect the two kinds of estimates to line up.

### 5.1 Estimates of the total mutation rate per site

The total mutation rate per site includes contributions from point mutations, indels, mobile element insertion (MEIs) and copy variants such as inversions. Current estimates of mutation rates per site per generation in humans are $1.2 \times 10^{-8} - 1.29 \times 10^{-8}$ for point mutations (***Kong et al., 2012***; ***Besenbacher et al., 2016***), $8.79 \times 10^{-10} - 9.82 \times 10^{-10}$ for indels (***Besenbacher et al., 2016***), whereas the rate for MEIs and other structural variants (including inversions and duplications) are more than two orders of magnitude lower than the point mutation rate (***Sudmant et al., 2015***; ***Gardner et al., 2019***; ***Belyeu et al., 2021***), making their contribution to our calculations below negligible. Adding up point mutation and indel rates results in a per site per generation estimate of $1.29 \times 10^{-8} - 1.38 \times 10^{-8}$. In estimating an upper bound on the rate of deleterious mutations at selected sites, we may consider weighting deletions by their length. For instance, we would like to count a deletion that begins at a neutral site but includes a selected site yet avoid counting one that includes multiple selected sites more than once. Counting deletions, which account for ~0.725 of indels, between once and up to their mean size of ~2.88 bp (***Besenbacher et al., 2016***), yields estimates of the total mutation rate in the range of $1.29 \times 10^{-8} - 1.51 \times 10^{-8}$ per site per generation. Throughout the paper, we use the middle of this range, that is, $1.4 \times 10^{-8}$ per site per generation, as our estimate for the total mutation rate ($u_0$). The estimates of the deleterious rate per putatively selected site for our best-fitting models fall well below the estimated total mutation rate (***Figure 4*** and Sections 4.1 and 4.4), as one would hope.

## 5.2 Estimating the proportion of deleterious mutations at putatively selected sites

Next, we estimate the proportional reduction of the substitution rate at selection targets relative to that at putatively neutral sites. We apply phyloFit (*Siepel and Haussler, 2004*) to the human-chimp-gorilla-orangutan (HCGO) alignment (based on the HCGO sequences from the 99-vertebrate alignment described in Section 2.2) in order to estimate the substitution rate per site on the human lineage from the ancestor with chimpanzee, for sets of selected and neutral sites (Section 3.1). To control for differences in base composition between the two sets, we estimate the reduction in substitution rates separately for each type of ancestral nucleotide (e.g. substitutions from G>X), and weight the proportional reductions by the proportions of each nucleotide in the set of selected sites. Controlling for the composition of triplets rather than single nucleotides produces similar estimates. Note that in choosing our sets of neutral and selected sites based on phylogenetic conservation (Sections 3.1 and 4.1), we excluded the human genome from the alignments and therefore our estimates of the reduction in substitution rates on the human branch should be minimally confounded with the choice of sites. Similarly, the conservation scores that serve as input for calculating CADD scores are based on the same 99-vertebrate alignment excluding the human reference genome (see Supplementary Table 1 in *Kircher et al., 2014*).

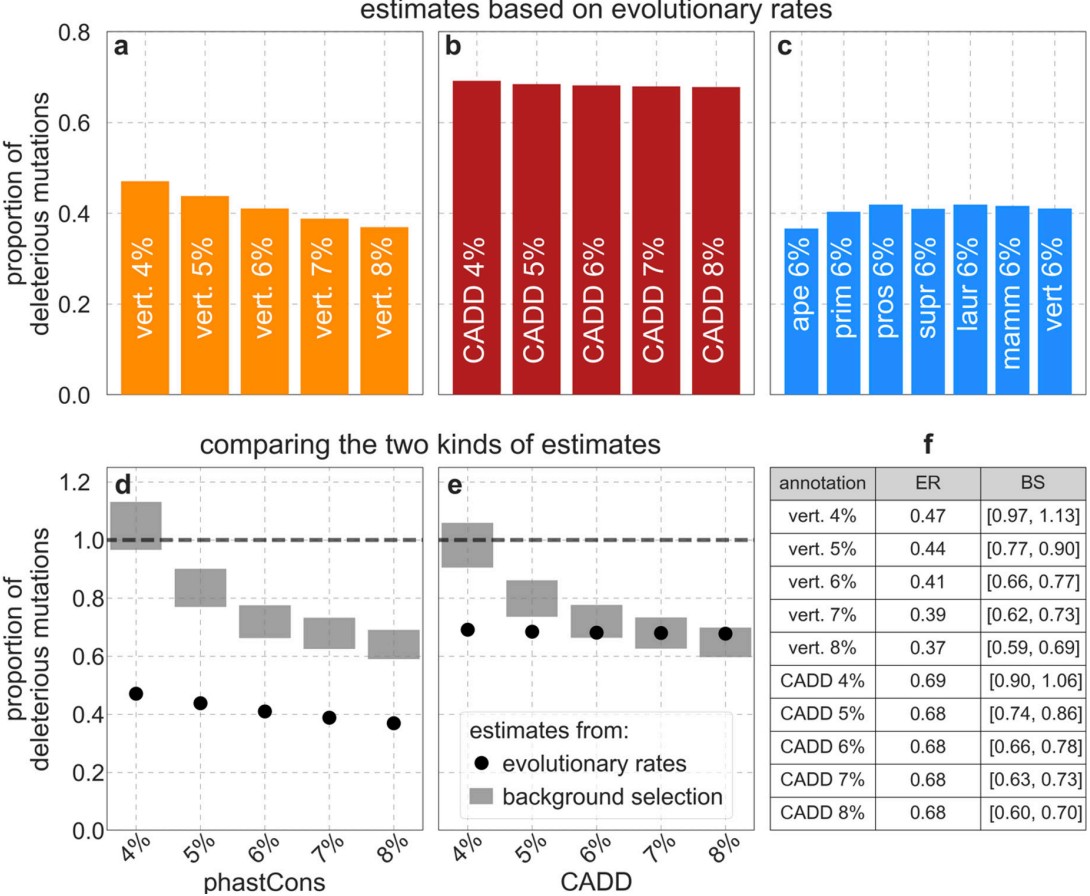

**Appendix 1—figure 30.** Different estimates of the deleterious mutation rate at putatively selected sites, measured relative to total mutation rates per site ($u_0$). Estimates based on evolutionary rates are shown for sets of selected sites chosen based on either: (**a**) the top 4–8% of phastCons scores for the 99-vertebrate alignment, (**b**) the top 4–8% of CADD scores, or (**c**) the top 6% of phastCons scores for alignments of varying phylogenetic depths. As expected, (see Section 4.1), the estimates are fairly insensitive to the phylogenetic depth (**c**). (**d and e**) Estimates based on evolutionary rates vs. those based on the effects of background selection, for sets of putatively selected sites based on phastCons scores for the 99-vertebrate alignment (**d**) and on CADD scores (**e**). (**f**) Estimates for different sets of putatively selected sites based on evolutionary rates (ER) and background selection effects (BS). The range of estimates based on background selection effects (in **d-f**) is due to the uncertainty about the total mutation rate per site (Section 5.1).

The estimates of the proportion of mutations that are deleterious are shown in *Appendix 1—figure 30* (and *Figure 4*), along with their comparison to estimates from background selection models. Expectedly, estimates based on substitution rates decline slightly as the cutoff phastCons or CADD score decreases (i.e. as the percentage of sites included in the selected set increases; *Appendix 1—figure 30a and b*). Importantly, estimates based on substitution rates are substantially greater for the sets chosen based on CADD than on phastCons scores (*Appendix 1—figure 30a and b*), whereas estimates based on background selection effects are similar in both cases (*Appendix 1—figure 30d and e*). We interpret this finding as reflecting the greater ability of CADD scores to identify selection on a single site resolution (*Kircher et al., 2014*), plausibly because CADD scores incorporate measures of phylogenetic conservation based on one site at a time (e.g. phyloP, GERP; *Cooper et al., 2005*; *Apostolico et al., 2006*) in addition to measures that rely on runs of sites, such as phastCons scores. Consequently, the two estimates of the deleterious mutation rate are within a factor of 2 for our best-fitting phastCons-based model whereas they overlap for our best-fitting CADD-based model, while the estimates based on background selection effects are similar in both cases (*Appendix 1—figure 30d and e*).

## 5.3 Interpreting the relationship between the two estimates

We would expect the two estimators of the deleterious mutation rate to yield similar but not identical answers. For one, the range of selection coefficients that cause a substantial reduction in evolutionary rates, e.g., $4N_e s \gtrsim 3$ (*Kimura and Crow, 1964*), is greater than the range of effects that cause a substantial reduction in diversity levels via classic background selection, e.g., $4N_e s \gtrsim 10$ (*Charlesworth et al., 1993*; *McVean and Charlesworth, 2000*; *Gordo and Charlesworth, 2001*). This consideration suggests that estimates based on evolutionary rates should be greater than those based on the effects of background selection (although non-equilibrium demographic history, notably changes in population size, might complicate quantitative expectations). On the other hand, we cannot expect to identify all selected sites and only those by our criteria. Estimates based on the effects of background selection plausibly soak up much of the contribution of missing selected sites, because their spatial distribution is likely to be highly correlated with sites that are included in our sets (see Section 4.1). In contrast, estimates based on evolutionary rates are affected only by the sites in our sets and would be biased downwards by the accidental inclusion of effectively neutral sites. For these reasons, we do not expect the two estimates of the deleterious mutation rate to align perfectly. Nonetheless, it is encouraging that when we rely on selected sites that amount to current estimates of the proportion the human genome under selection, that is, ~5–9% (*Kellis et al., 2014*; *Rands et al., 2014*), our two estimates of the deleterious mutation rate are quite similar. Moreover, the similarity is highest when we use CADD scores, which are better than phastCons scores at identifying selection on a single site resolution (*Kircher et al., 2014*). Thus, our results resolve the issues raised by the substantial overestimation of the deleterious mutation rate in past work (*McVicker et al., 2009*).

## 6. Statistics

### 6.1 Estimates of explained variance

Our main quantitative measure for the fit of our models is the variance in diversity levels explained by our predictions, $R^2$, for different window sizes. A concern in using $R^2$ as a measure of fit is that it not be inflated by overfitting. To avoid this problem, we exclude the data in a given window from the inference used to predict diversity levels in that window. Specifically, we divide the autosomal polymorphism data into contiguous non-overlapping blocks of 2 Mb, and repeat the inference using the data excluding one block at a time. As the autosomes cover just over 2.88 Gb, this amounts to repeating our inference 1440 times. When we calculate the contribution of a given window (of size $\leq$ 2 Mb) to $R^2$, we use the prediction based on excluding the 2 Mb block containing that window. In Section 6.3, we use the same datasets and inferences to calculate jackknife estimates for the sampling error of our estimates of model parameters.

*Appendix 1—figure 31* shows the relative difference between our $R^2$ estimates using all the data and this exclusion approach for our two best-fitting models. Additionally, in *Appendix 1—figure 48* we compare the results of our main analyses of the best-fitting CADD-based model, using all the data, out-of-sample predictions in contiguous, non-overlapping 2 MB blocks, and out-of-sample

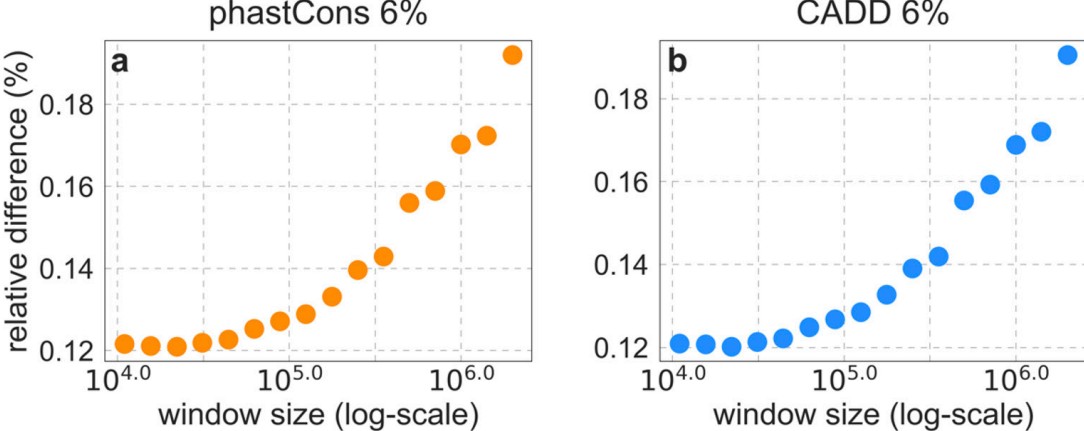

**Appendix 1—figure 31.** The relative difference between $R^2$ estimates using all the data and the exclusion approach described in the text, for our two best-fitting models.

predictions for each autosome. These results suggest that overfitting has a tiny effect, which is not surprising given the large amounts of data used in our inferences. Given the negligible effect and computational burden of these analyses, we do not repeat it for each of the models we examine, and use the $R^2$ estimates based on predictions using all the data instead.

## 6.2 Comparing the fit of different maps

We use permutations of paired maps to test whether differences in $R^2$ between two maps are statistically significant. Assume without loss of generality that $R^2$ for a given window size is greater for map I than for map II. We divide autosomes into 2881 contiguous non-overlapping blocks of 1 Mb and generate a new map (map A) by picking each 1 Mb block from map I or II at random; we generate the complementary map (map B) by picking the alternative 1 Mb blocks throughout. This way, we generate $n$ paired maps and calculate the difference in $R^2$ between each pair, $\Delta R^2$, to obtain a distribution for the expected differences in $R^2$ between maps I and II under the null hypothesis that their fit to polymorphism data is roughly equivalent. Having $r$ denote the number of permutations with $\Delta R^2$ greater than or equal to the observed difference $\Delta R_O^2$, we estimate the p-value for $\Delta R_O^2$ under the null by $p = (r + 1)/(n + 1)$.

We illustrate this procedure by comparing our two best-fitting models (*Appendix 1—figure 32*). The fit of these models is very similar, with, for example, $R^2 = 0.599$ and $0.597$ at the 1 Mb scale for the models based on CADD and phastCons scores, respectively. We find that the difference between the fits is not statistically significant, supporting our claim that the functional annotations incorporated in CADD offer little or no improvement in predictive power (see Main Text and Section 4.4). Using

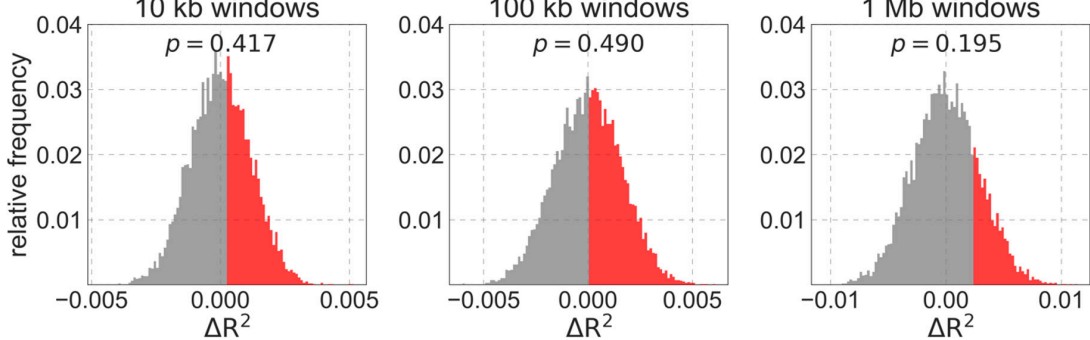

**Appendix 1—figure 32.** Assessing the differences in fit between our best-fitting models on three spatial scales. We show the distribution of differences in explained variance ($\Delta R^2$) for 10,000 paired permutations of the best-fitting CADD and phastCons based maps. The part of the distributions with $\Delta R^2 \geq \Delta R_O^2$ is in red, and the corresponding p-value is shown above.

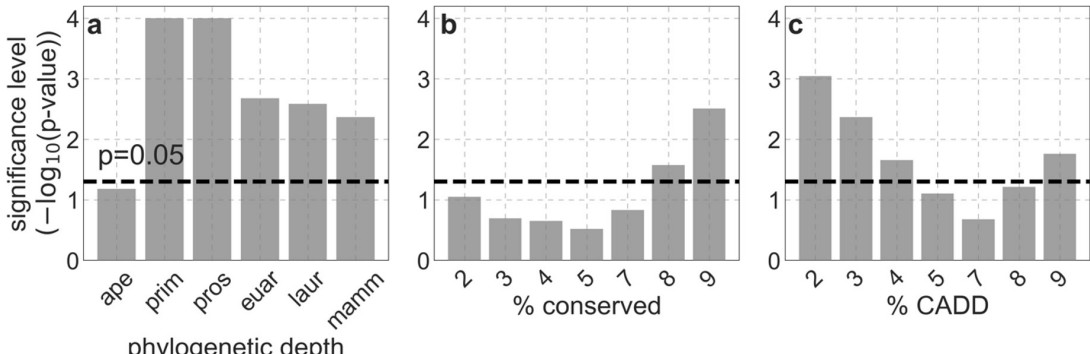

**Appendix 1—figure 33.** Significance level of differences in fit between our best-fitting models and variations on these models, on the 1 Mb spatial scale. (**a and b**) Comparison of our best-fitting phastCons-based model with phastCons-based models with alternative phylogenetic depths (**a**) and conservation thresholds (**b**). (**c**) Comparisons of our best-fitting CADD-based model with CADD-based models with alternative thresholds.

this procedure to compare our best-fitting phastCons-based model with phastCons-based models with alternative phylogenetic depths or conservation thresholds, we only find significant differences at the 1 Mb scale in a small subset of cases (*Appendix 1—figure 33a and b*). The same is true for comparisons of our best-fitting CADD-based model with CADD-based models with alternative thresholds (*Appendix 1—figure 33c*). We note that even when the fits are significantly worse, they are still far closer to the fits of our best-fitting models than any of the models based on other choices of selection targets discussed in Section 4.

Interestingly, the difference in fit between the model based on conservation in 99-vertebrates and four-apes is the only non-significant comparison across phylogenetic depths (*Appendix 1—figure 33a*), despite the fact that other phylogenetic depths have $R^2$ values closer to the 99-vertebrate result across various window sizes (*Appendix 1—figure 14c*). We believe this is due to the fact that the four-ape alignment may actually better capture some recent targets of selection, but with greater noise, reflecting a tradeoff in the power to detect conservation vs. functional turnover. As a result, a non-negligible subset of 1 Mb windows in the four-ape yield better fits to the data than the 99-vertebrate map. In contrast, maps from deeper in the phylogeny are essentially highly correlated to the 99-vertebrate map, and the small differences in fit are uniformly biased in favor of the 99-vertebrate map across 1 Mb windows due to its greater power to resolve the boundaries of conserved elements.

## 6.3 Sampling error in parameter estimates

We use a jackknife resampling approach to estimate the sampling errors of our parameter estimates (see, e.g. *Patterson et al., 2012*). To this end, we perform the inference on datasets excluding 2 Mb blocks as described in Section 6.1. Specifically, denoting the parameter of interest by $\theta$, and the estimate based on the data set excluding block $i = 1, \ldots, n$ by $\hat{\theta}_i$, our jackknife estimates of the

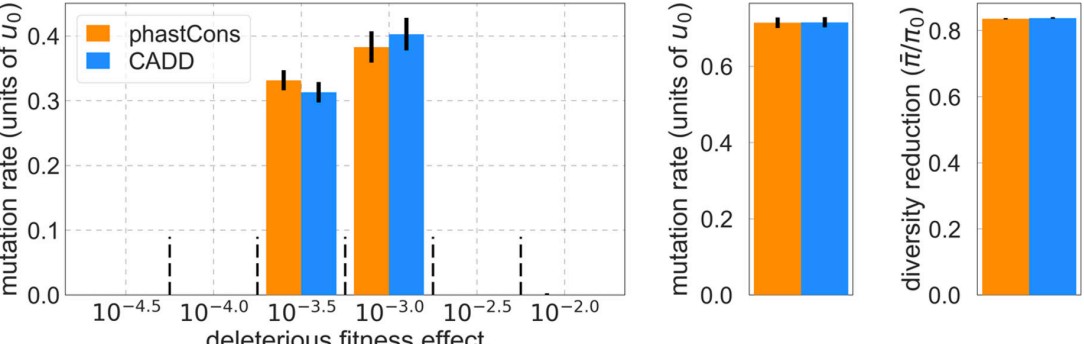

**Appendix 1—figure 34.** Estimates of the sampling errors of parameter estimates for our two best-fitting models. The bars denote $\pm SE$ estimated using jackknife as described in the text.

mean and variance are $\bar{\theta} = \frac{1}{n}\sum\limits_{i=1}^{n}\hat{\theta}_i$ and $V(\bar{\theta}) = \frac{n-1}{n}\sum_{i=1}^{n}\left(\hat{\theta}_i - \bar{\theta}\right)^2$, respectively. We use the standard deviation $\sqrt{V(\bar{\theta})}$ as our measure of sampling error (SE). *Appendix 1—figure 34* shows the SEs for the parameter estimates of our two best-fitting models. As these examples illustrate, these errors are quite small and do not affect the conclusions of our analyses. Consequently, and given the computational cost of obtaining them, we do not calculate these SEs for most models.

## 7. Results for other human populations

Here, we examine whether the maps of the effects of background selection that we infer and evaluate using polymorphism data from YRI provide a good fit to data from other populations. To this end, we use data from each of the other 25 populations collected in Phase III of the 1000 genomes project (*Auton et al., 2015*), which span a wide geographic range and have had different demographic histories (*Auton et al., 2015*), to infer the maps corresponding to our best-fitting models. The population-specific maps can be found at https://github.com/sellalab/HumanLinkedSelectionMaps.

Overall, we find that the maps and main parameters inferred in different populations are remarkably similar (*Appendix 1—figures 35, 50–52*). When we compare the predictions of relative diversity levels along autosomes (i.e. relative to the mean in each population) we find nearly perfect correlations across window sizes (*Appendix 1—figure 35a and b*). The distributions of selection effects of deleterious mutations, estimates of the total deleterious mutation rate per selected site, and the mean reduction in diversity levels, are all quite close among populations (*Appendix 1—figure 35c*). The similarity among maps implies that we can use the maps of relative diversity levels inferred in YRI to predict diversity levels in other populations without loss of accuracy. Specifically, we multiply predictions based on YRI by a constant, chosen such that the predicted and observed mean diversity level in the focal population match. *Appendix 1—figure 36* illustrates that the adjusted YRI maps predict diversity levels as well as the population specific ones.

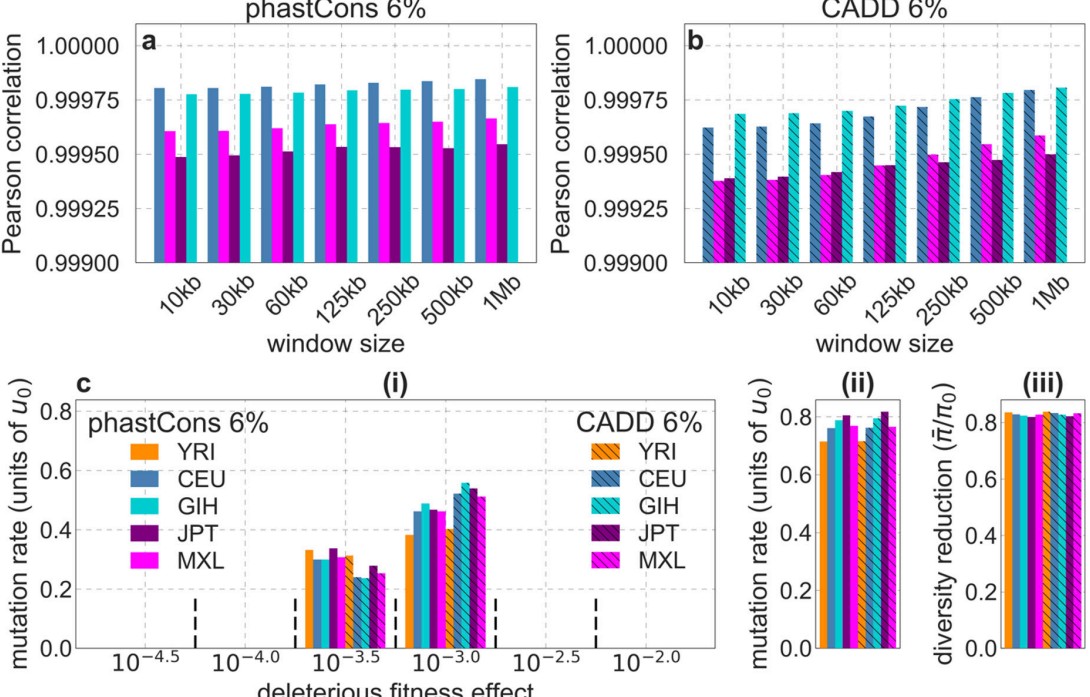

**Appendix 1—figure 35.** The maps and parameter estimates for different populations are remarkably similar. Shown are the results for our best-fitting models using data for one of 1000 Genomes Project populations from each continental group: Africa – Yoruba (YRI), Europe – North-Western European (CEU), South Asia – Gujrati Indian (GIH), East Asia – Japanese (JPT), and Americas – Mexican (MXL). (**a and b**) The Pearson correlations between predictions of relative diversity levels (compared to the population mean) in YRI vs. the other populations for the models based on phastCons (**a**) and CADD (**b**) scores. (**c**) Comparison of parameter estimates using data from these populations (panels **c** (**i-iii**) as described in panels **a** (**i-iii**) in *Appendix 1—figure 14*).

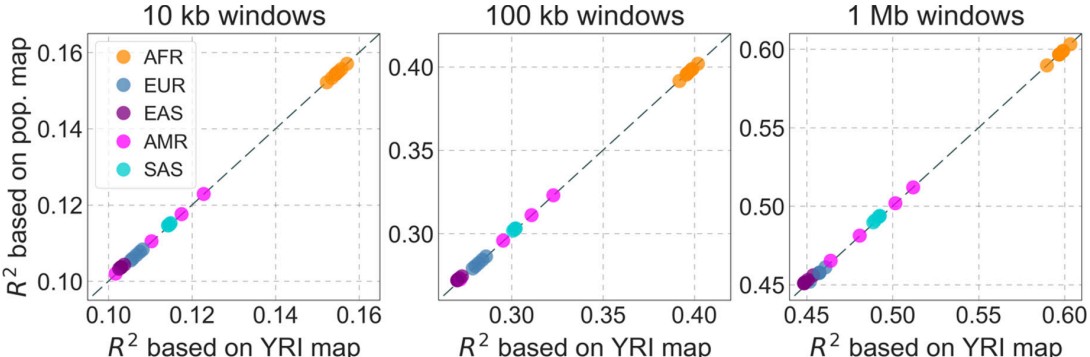

**Appendix 1—figure 36.** The proportion of variance in diversity levels explained ($R^2$) in different populations, using the population specific map vs. the YRI map. We show the results for three window sizes (10 kb, 100 kb, and 1 Mb) based on our best-fitting CADD-based model. Each point corresponds to one of the 26 populations sampled in the 1000 genomes project and is colored based on continental origin, i.e., African (AFR), European (EUR), South Asian (SAS), East Asian (EAS), and American (AMR).

While the maps inferred in different populations are highly similar, the proportion of variance explained differs substantially among populations (*Appendix 1—figure 36*). These differences can be explained by the effects of different demographic histories (e.g. historical changes in effective population sizes) on variation in diversity levels across the genome. To make this more concrete, we consider a simple model for the variance in neutral diversity levels in non-overlapping windows of a given size; for simplicity, we ignore variation in mutation rates across windows. We denote the relative (average) diversity level in window $i$ by $y_i = \pi_i/\bar{\pi}$, the predicted relative diversity level in that window by $f_i = B_i/\bar{B}$, and the corresponding residual by $e_i = y_i - f_i$, where $\bar{y} = \bar{f} = 1$ and $\bar{e} = 0$. We can now decompose the variance in relative diversity levels across windows, as

$$V\left(y\right) = V\left(f\right) + V\left(e\right) + 2Cov\left(e,f\right), \tag{15}$$

where $V(f)$ corresponds to the variance due background selection; $V(e)$, the variance of residuals, can be thought of as reflecting the effects of drift and demographic history; and the covariance, $Cov(e,f)$, can be thought of as reflecting the interaction between background selection and demographic history. Recasting the proportion of variance explained in these terms, we find that

$$R^2 \equiv 1 - \frac{V(e)}{V(y)} = \frac{V(f)}{V(y)}\left(1 + \beta\right), \tag{16}$$

where $\beta = 2 \cdot Cov(e,f)/V(f)$ is the slope of the linear regression of the residuals against the predictions, which reflects the effects of interactions between background selection and demographic history on diversity levels.

Given that we found the predicted effects of background selection to be highly similar across populations, this modeling exercise sets up a testable prediction: if the interaction terms were nil, the difference in $R^2$ among populations should come from the total variance in the denominator, $V(y) = V(f) + V(e)$, and specifically from the contribution of demographic history to this variance, $V(e)$. *Appendix 1—figure 37a–c* suggest that while most of the differences in $R^2$ among populations are indeed explained by differences in total variance due to demographic history, the interaction terms (the $\beta$s) are non-zero. To examine these interactions further, we look at the relationship between residuals, $e_i$, and predictions, $f_i$, in several populations (*Appendix 1—figure 37d–h*). We find a strong apparent dependency at the low and high ends of our predicted range, presumably reflecting artifacts due to thresholding at the low end (see Section 1.5) and possibly the effects of ancient introgression at the high end (see Section 8); removing 1.5% of the windows at each of these extremes appears to largely remove these effects. In the rest of the range, we find a weak negative correlation between residuals and predictions (which becomes somewhat stronger when we remove the ends). We also find that this correlation varies substantially among populations, for example, –0.06 to –0.1 for 1 Mb windows, which is what we would expect given differences in demographic

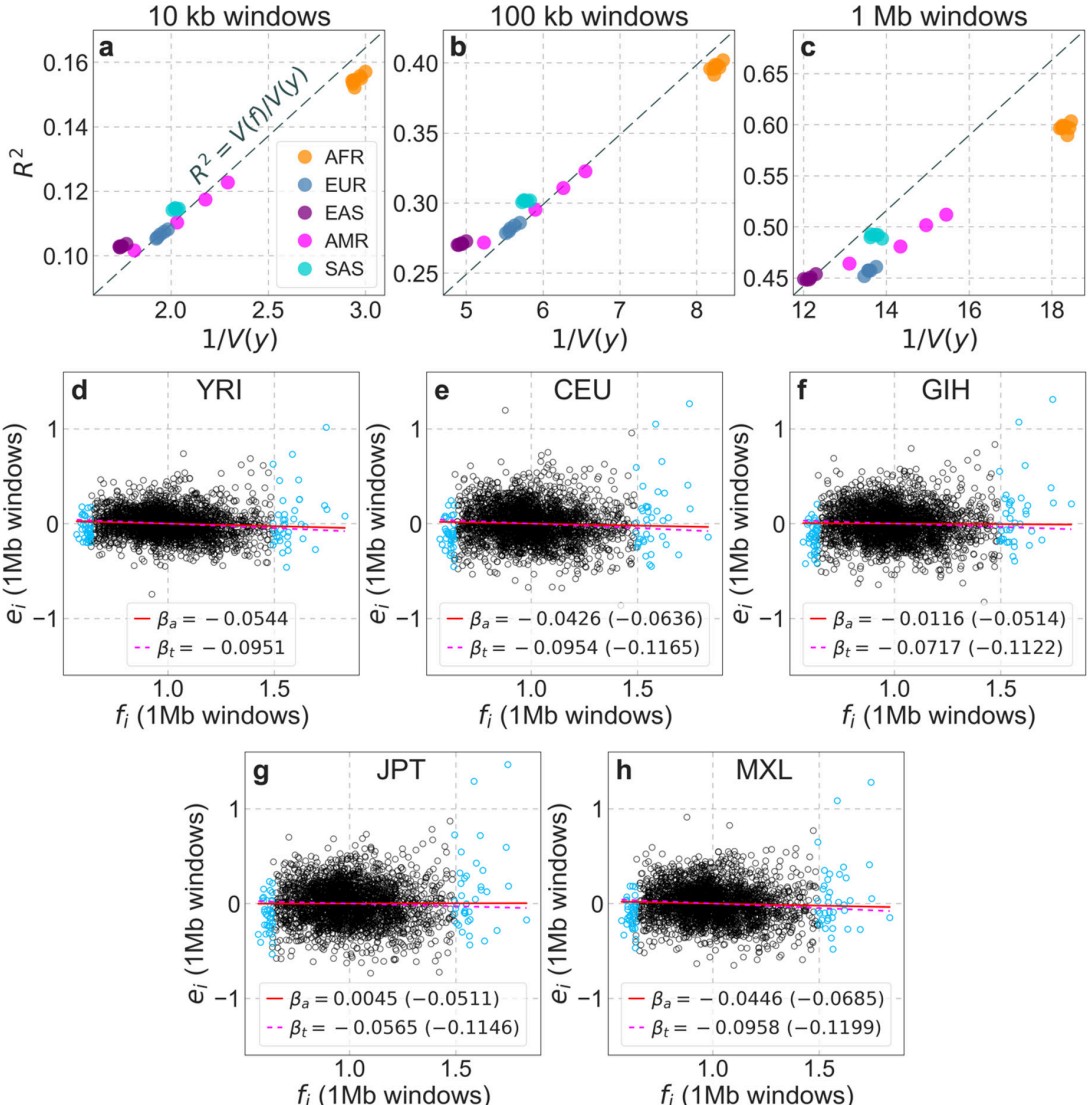

**Appendix 1—figure 37.** Differences among populations in the variance in diversity levels explained by our map of the effects background selection. (**a–c**) The variance explained ($R^2$) as a function of $1/V(y)$ (where $V(y)$ is the total variance), for three choices of window size. If the interaction terms ($\beta$) were 0, we would expect populations to fall on the dashed line $R^2 = V(f) \cdot 1/V(y)$, with slope $V(f)$ and differences in $V(y)$ due to demographic history. The distances from the dashed line reflect the interaction terms (specifically, $R^2 - V(e)/V(y) = \beta \cdot V(e)/V(y)$; *Equation 15*). The points correspond to the 26 populations sampled in the 1000 genomes project (*Auton et al., 2015*) and are colored by continental origin as described in *Appendix 1—figure 36*. Here we base our predictions on our best-fitting CADD-based map in YRI, but using other population-specific maps yields almost identical results. (**d-h**) The relationship between the residuals ($e_i$) and predictions ($f_i$) in representative populations (same as in *Appendix 1—figure 36*) on the 1 Mb scale. The 1.5% of windows with lowest and highest predicted values, where our predictions are likely off for various reasons (see Sections 1.5 and 8), are marked in blue. The $\beta$s for each population, with and without extreme points, are shown on the graph (denoted 'a' for 'all' and 't' for 'trimmed', respectively). As above, we use the predictions in YRI, but other population maps yield qualitatively similar results ($\beta$s obtained using the corresponding population specific maps are shown in parenthesis).

history. Thus, our analysis suggests that interactions between demographic history and background selection also contribute to the differences in $R^2$ among populations.

In summary, our findings suggest that the effects of background selection are similar across human populations, and that differences among populations in the proportion of variance in diversity levels that our predictions explain are likely due to differences in population demographic history.

Interestingly, there appears to be an interaction between the effects of background selection and demography on diversity levels, which varies among populations, as recently suggested by several studies (*Comeron, 2017*; *Wang et al., 2017*; *Torres et al., 2018*; *Torres et al., 2020*). Our maps of the effects of background selection have enabled us to identify evidence for these interaction effects and should facilitate a better understanding of these effects in the future.

## 8. Diversity levels where background selection is weakest ($B \approx 1$)

Our maps of background selection effects are well calibrated throughout the range of predicted effects, with two exceptions. One is in the ~5% of sites in which background selection is predicted to be strongest, where predictions are imprecise; this arises from the thresholding approximation we apply in fitting, and is discussed in Section 1.5. The other exception is for sites in which background selection is predicted to be the weakest, where observed diversity levels are markedly greater than expected (*Figure 5* and *Appendix 1—figure 38*). A close up on this region shows that observed values depart from predictions in the ~2% of sites where $B \gtrsim 0.98$ (*Appendix 1—figure 38b*). Similar behavior is seen in all 26 populations sampled in the 1000 Genomes Project (*Appendix 1—figure 52*). We consider possible explanations for it here.

First, we characterize the main covariates of the strength of background selection (quantified by $B$), such as recombination rate, base composition, and chromosomal position. Beyond the inherent interest in these covariates, they point toward processes that may explain the departure from our predictions near $B = 1$. Second, we investigate whether differences in rates of different kinds of mutations and of biased gene conversion associated with the covariates of $B$ can explain the departure near $B = 1$; our analysis suggests that they cannot. Third, we argue that the residual effect of ancient introgression between archaic humans and ancestors of extant humans may contribute to this departure.

### 8.1 Covariates of $B$

We expect the effects of background selection to be strongest in regions with low rates of recombination and high densities of functional sites, because neutral variation in such regions will be linked to more deleterious mutations. In line with these expectations, recombination rates increase

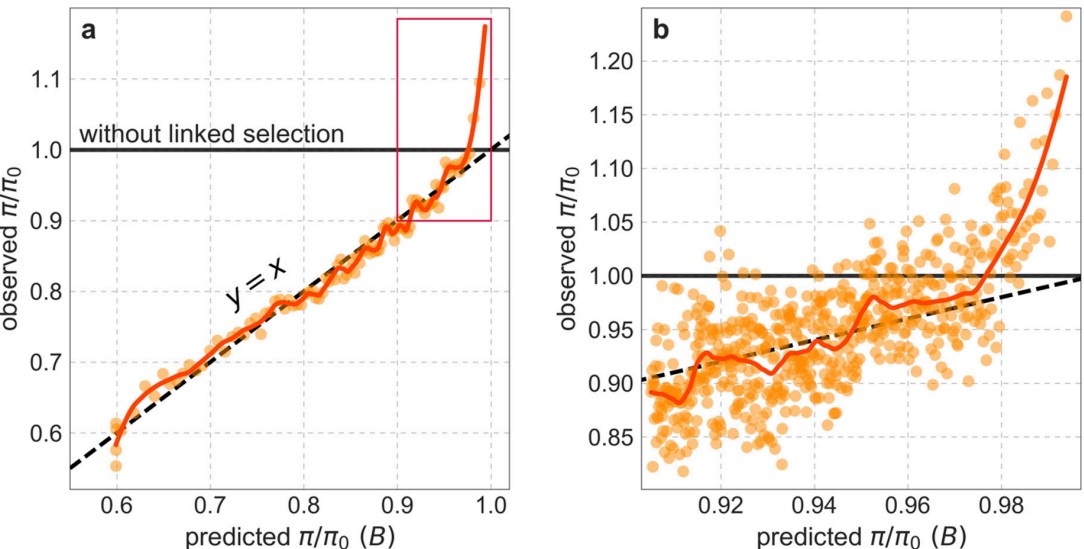

**Appendix 1—figure 38.** Observed vs. predicted neutral diversity levels across the autosomes (similar to *Figure 5*). (**a**) Light orange scatter plot: We divide putatively neutral sites into 100 equally sized bins based on the predicted effect of background selection, $B$, from the best-fitting CADD-based model. For predicted values (x-axis), we average the predicted $B$ in each bin. For observed values (y-axis), we divide the average diversity level in each bin by the average predicted diversity level in the absence of background selection, $\pi_0$, after scaling each in bin by its estimated local (relative) mutation rate ($u(x)/\bar{u}$ in *Equation 8*; Section 3.3). Dark orange curve: the LOESS curve for a similarly defined scatter plot but with 2000 rather than 100 bins (with span = 0.1). (**b**) A close-up near $B = 1$ corresponding to the boxed region in (**a**). Here, the LOESS curve has span = 0.033 and the scatter plot corresponds to 2000 bins (showing the top 30%).

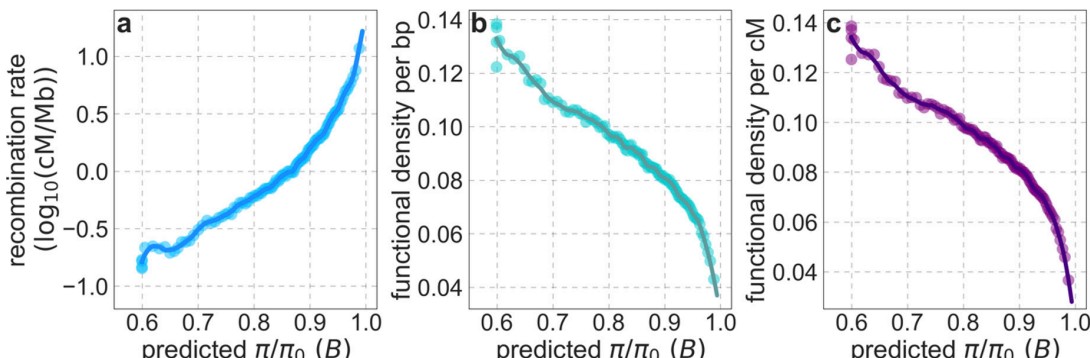

**Appendix 1—figure 39.** Average recombination rate (**a**) and functional density per bp (**b**) and per cM (**c**) as a function of predicted $B$. The predicted $B$, binning of putatively neutral sites and LOESS fitting are as described in *Appendix 1—figure 38a*. We calculate the average recombination rate in each bin based on the African-American admixture map from *Hinch et al., 2011* (Section 2.3). We measure functional density by calculating the mean phastCons score (based on the 99-vertebrate alignment) in a radius of 50 kb (**b**) or 0.05 cM (**c**) around each putatitively neutral site, and averaging these means over sites in each bin.

with greater predicted $B$ (*Appendix 1—figure 39a*); in particular, they increase sharply between the 99[th] and 100[th] percentile of predicted $B$ to >10 cM/Mb, which is tenfold the autosomal average. In addition, as expected, the average level of conservation around neutral sites decreases as $B$ increases (*Appendix 1—figure 39b and c*).

Next, we consider base composition and other factors that are known to affect rates of mutation and biased gene conversion (BGC). GC content has a J-shaped dependence on predicted $B$ (*Appendix 1—figure 40a*). The greater peak in GC content, in regions under weak background selection ($B$ near 1), is plausibly largely driven by the long-term effects of BGC due to higher rates of recombination in these regions (*Duret and Galtier, 2009*; *Li et al., 2019*; *Appendix 1—figure 39a*). Both peaks (for low and high $B$) are associated with an increase in the proportion of GC sites in CpG islands but this proportion is small throughout (<1%) (*Appendix 1—figure 40b*), suggesting that it has little effect on GC content and on mutation rates. In contrast, methylated CpG content increases sharply as predicted $B$ approaches 1 (*Appendix 1—figure 40c*), suggesting a corresponding increase in the rate of C>T transitions. The proportion of sites in C>G hypermutable regions also increases with predicted $B$ (*Appendix 1—figure 40d*).

Lastly, predicted $B$ is associated with chromosomal position, with regions under weak background selection ($B$ near 1) clustered near telomeres (*Appendix 1—figure 41*). In turn, regions near telomeres are early replicating, and replication timing is known to affect mutational patterns (*Stamatoyannopoulos et al., 2009*).

## 8.2 Mutational spectrum and biased gene conversion

As we noted, the covariates of $B$ are associated with mutational processes and with biased gene conversion that affect diversity and divergence levels (*Appendix 1—figure 42*). Specifically, we see the footprints of the following processes:

- Increased rates of **A>C/T>G** and **A>G/T>C** substitutions near $B = 1$ (*Appendix 1—figure 42a*), due to higher rates of biased gene conversion that are associated with the higher rates of recombination (*Duret and Galtier, 2009*; *Li et al., 2019*; *Appendix 1—figure 39a*). Biased gene conversion also reduces the rates of **C>A/G>T** and **C>T/G>A** substitutions, but this is not clearly visible (*Appendix 1—figure 42b*), presumably because of other processes affecting these substitutions (see below).
- Increased rate of **C>T mutations** near $B = 1$ (*Appendix 1—figure 42b*), associated with the higher methylated CpG content (*Barrett et al., 2013*; *Appendix 1—figure 40c*).
- Reduced rates of **C>A/G>T** and **A>T/T>A** mutations near $B = 1$ (*Appendix 1—figure 42a and b*), associated with improved repair of these types of mutations near origins of replication, which tend to be near telomeres (*Stamatoyannopoulos et al., 2009*; *Appendix 1—figure 41*).
- Increased rates of **C>G/G>C** mutations near $B = 1$ (*Appendix 1—figure 42b*), due to the enrichment of C>G hypermutable regions (*Jónsson et al., 2017*; *Appendix 1—figure 40d*).

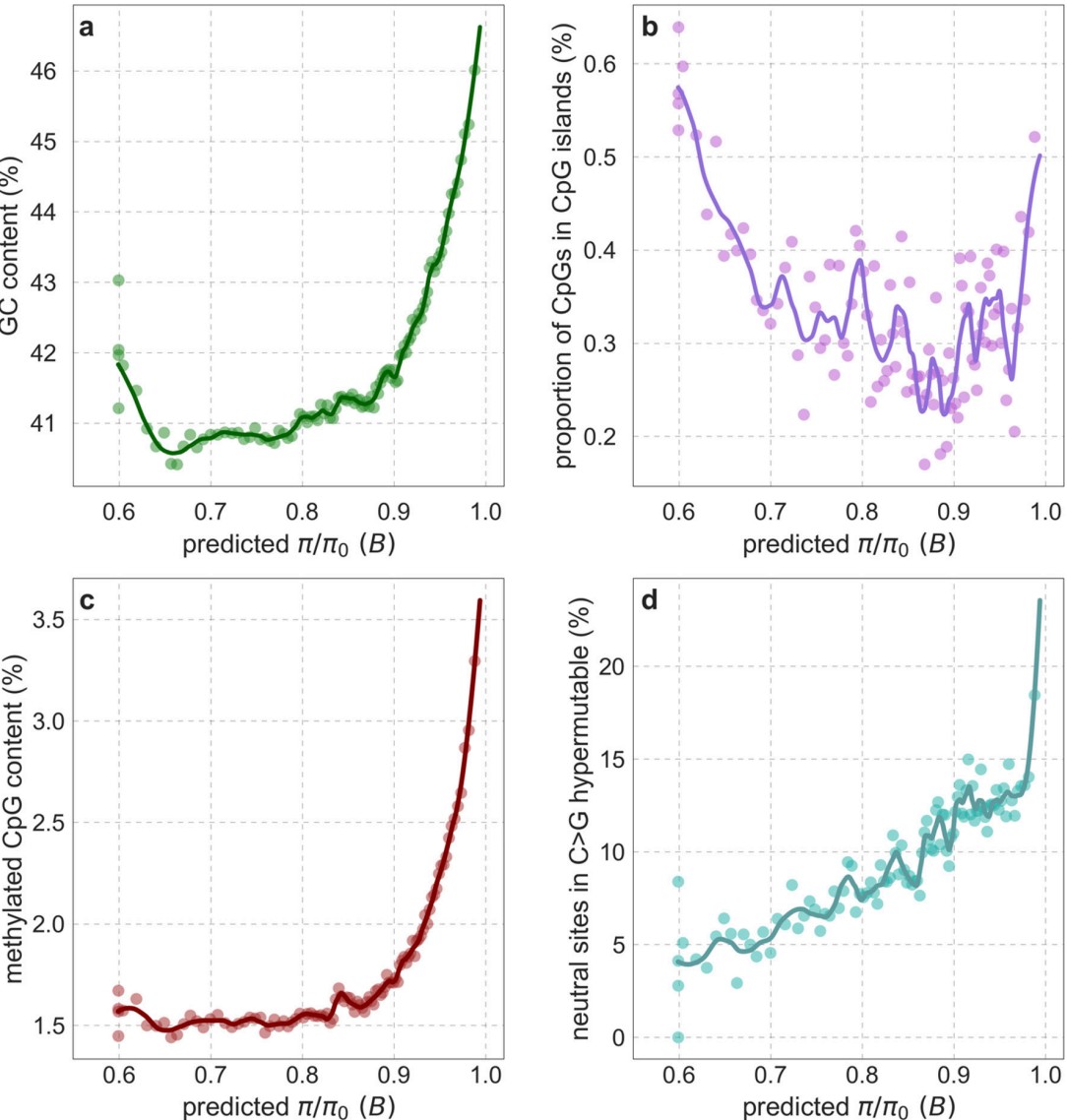

**Appendix 1—figure 40.** GC content (**a**), CpG sites in CpG islands (**b**), proportion of methylated CpGs in neutral sites (**c**), and proportion of neutral sites in C>G hypermutable regions (**d**) as a function of predicted $B$. Proportions and other quantities are measured for putatively neutral sites, whose type (i.e. GC and CpG) is defined based on the inferred state in the human-chimpanzee ancestor (Section 2.7). Data sources are detailed in Section 2.8. The predicted $B$, binning of putatively neutral sites and LOESS fitting are as described for *Appendix 1—figure 38a*.

Consequently, the rates of substitutions between any two bases covary with predicted $B$ (*Appendix 1—figure 42*). However, the different types of substitutions have markedly different contributions to levels of diversity and divergence (*Appendix 1—figure 43*), and different dependencies on predicted $B$ (*Appendix 1—figure 42*).

To investigate whether these processes can explain the departure from our predictions near $B = 1$, we break up the observed diversity levels by types of substitutions (*Appendix 1—figure 44*). We reason that if all types behave similarly near $B = 1$, the differential processes affecting them cannot explain the departure from predictions (at least not fully). Note that, up to a multiplicative constant, our observations are ratios of diversity levels and substitution rates (on the 8-primate phylogeny), implying that the signatures of processes that affect diversity and divergence similarly should cancel out. Conversely, for a process to affect our observations it must have noticeably different effects on diversity and divergence. We find that the observations associated with different types of substitutions largely align with each other and with the observations that include all types

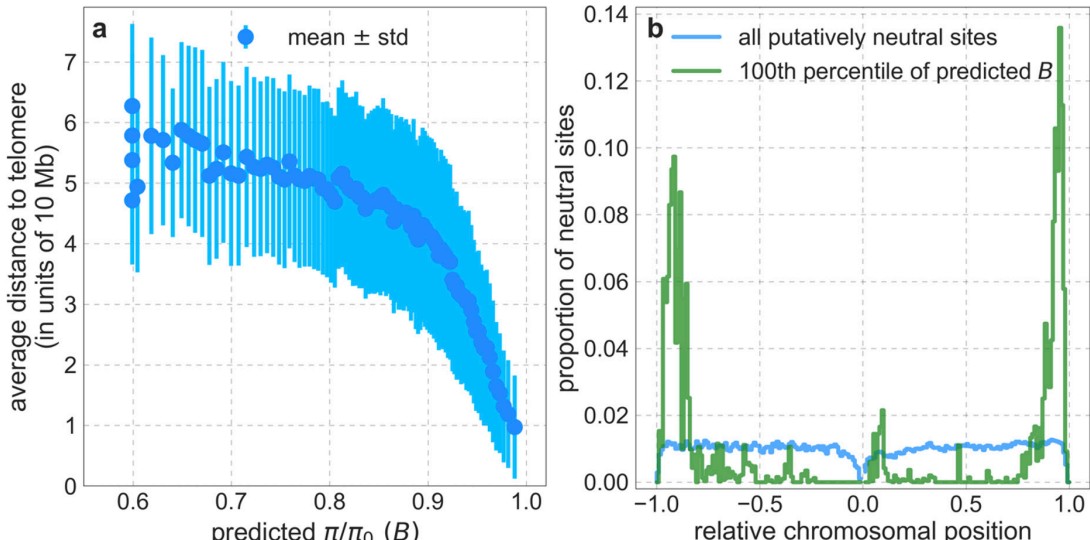

**Appendix 1—figure 41.** The relationship between chromosomal position and predicted $B$. (**a**) The distance to telomeres is measured on the same chromosome (see Section 2.8). (**b**) The distribution of putatively neutral sites by relative chromosomal position, defined as the ratio of a site's distance to the centromere and the distance between centromere and telomeres on that chromosomal arm (see Section 2.8); relative distances corresponding to the shorter chromosome arm are shown on the left (in [–1, 0]) and to the longer arm on the right (in [0, 1]). The binning of putatively neutral sites by predicted $B$ is as described for *Appendix 1—figure 38a*.

jointly (*Appendix 1—figures 38 and 44*). Specifically, they align with predictions throughout nearly the entire range of predicted $B$, and are markedly higher than predictions near $B = 1$.

Nonetheless, the observed levels associated with C>G mutations near $B = 1$ are markedly higher than for other types of substitutions (*Appendix 1—figure 44b and d*). This effect contributes negligibly to the total observed levels near $B = 1$, however, because C>G substitutions have a minor contribution to total diversity and divergence levels (*Appendix 1—figure 43*). The higher levels of C>G substitutions are associated with the enrichment of C>G hypermutable regions near $B = 1$

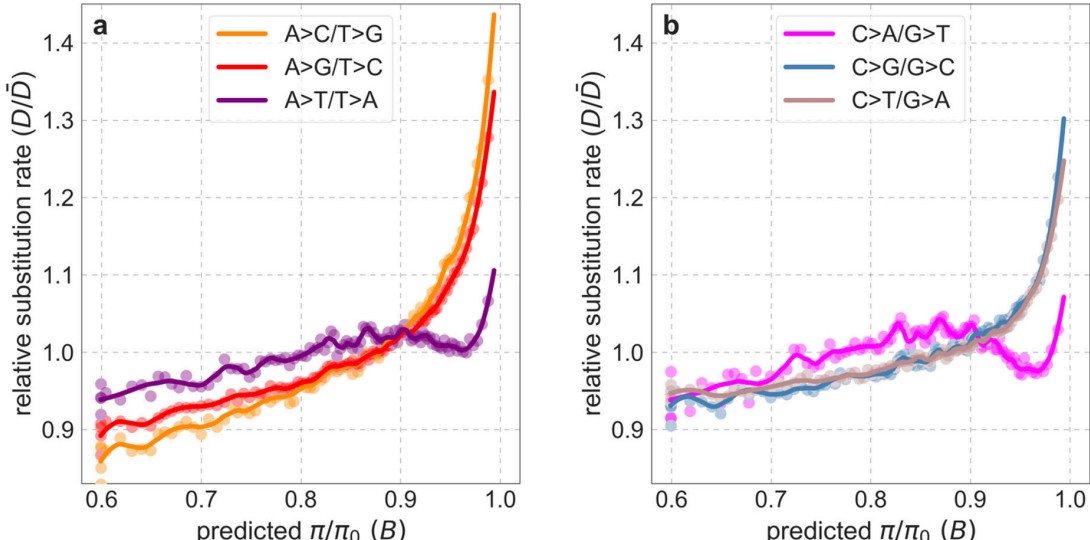

**Appendix 1—figure 42.** Rates of different types of substitutions as a function of predicted $B$. We bin putatively neutral sites by predicted $B$ as described in *Appendix 1—figure 38a*. We calculate the rate of X>Y substitutions in a bin by dividing the estimate of the number of X>Y substitutions at its sites in an 8-primate phylogeny by the estimated number of its sites with state X in the ancestor of that phylogeny (see Section 3.3). We obtain the relative rate by dividing the rate in a bin by the average rate across bins. We show the rates of substitutions with ancestral state AT in (**a**) and GC in (**b**).

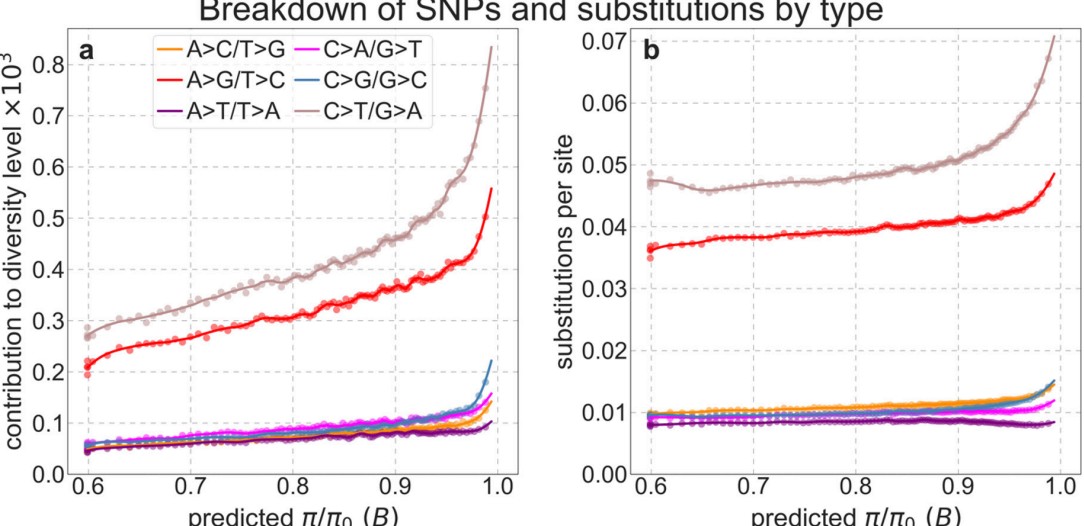

**Appendix 1—figure 43.** Contribution of different types of substitutions to diversity levels (**a**) and number of substitutions per site in the 8-primate phylogeny (**b**) as a function of predicted $B$. We bin putatively neutral sites by predicted $B$ as described in **Appendix 1—figure 38a**. (**a**) We define the ancestral state, that is, AT or GC, as the inferred state in the human-chimpanzee ancestor (see Section 2.7). We define the contribution of each type of substitution to the diversity level in a bin as the ratio of the number of pairwise differences of that type and the total number of pairwise comparisons in a bin; this way, the sum over types equals the observed diversity level in a bin ($\pi$). (**b**) We calculate the number of substitutions of each type in a bin as described in **Appendix 1—figure 42**, but here we normalize it by the number of sites, such that the sum over types equals the observed number of substitutions per site in the 8-primate phylogeny.

(**Appendix 1—figure 40d**). Notably, when we remove these regions, the observed levels associated with C>G substitutions are no longer higher than for other types of substitutions (**Appendix 1—figure 45**). Removing C>G hypermutable regions also affects the magnitude of the departure from predictions for other types of substitutions (compare **Appendix 1—figure 45c and d** with **Appendix 1—figure 44c and d**), because C>G is not the only type of mutation whose rate is higher in these regions. These hypermutable regions plausibly affect diversity more than divergence (and thus our observations) given that their effects are stronger in recent human evolution than in the more distant past and in the lineages of closely related species (Ipsita Agarwal and Molly Przeworski, personal communication). This may reflect the fact that these regions were identified in extant humans and/or a dependence of their effects on life history (**Jónsson et al., 2017**; **Gao et al., 2019**). Setting the causes aside, even when we remove these regions from the set of putatively neutral sites used in our inference, observed levels of all types are still markedly higher than the revised predictions near $B = 1$ (**Appendix 1—figure 46**).

In summary, while we cannot rule out that there are other mutational processes that contribute to the departure from predictions near $B = 1$, our analysis suggests that known mutational processes and biased gene conversion fall short of explaining these departures.

## 8.3 A footprint of archaic introgression?

Next, we consider whether the excess diversity observed near $B = 1$ could reflect a residual signal of archaic introgression. The presence of archaic alleles at a locus increases diversity because their coalescence with modern human alleles traces back to the ancestors of modern humans and the archaic hominin from which they originated. Archaic introgression could help to explain the excess diversity in regions with $B \approx 1$, if archaic alleles were more common in these regions. As we argue below, there are good reasons to believe this to be the case.

Aside from evidence for positive selection on introgressed alleles in a few cases (**Sankararaman et al., 2014**; **Vernot and Akey, 2014**; **Racimo et al., 2015**), the pattern of Neanderthal and Denisovan introgression in contemporary human populations appears to be dominated by purifying selection to remove archaic ancestry from the human genome, as evidenced by the depletion of

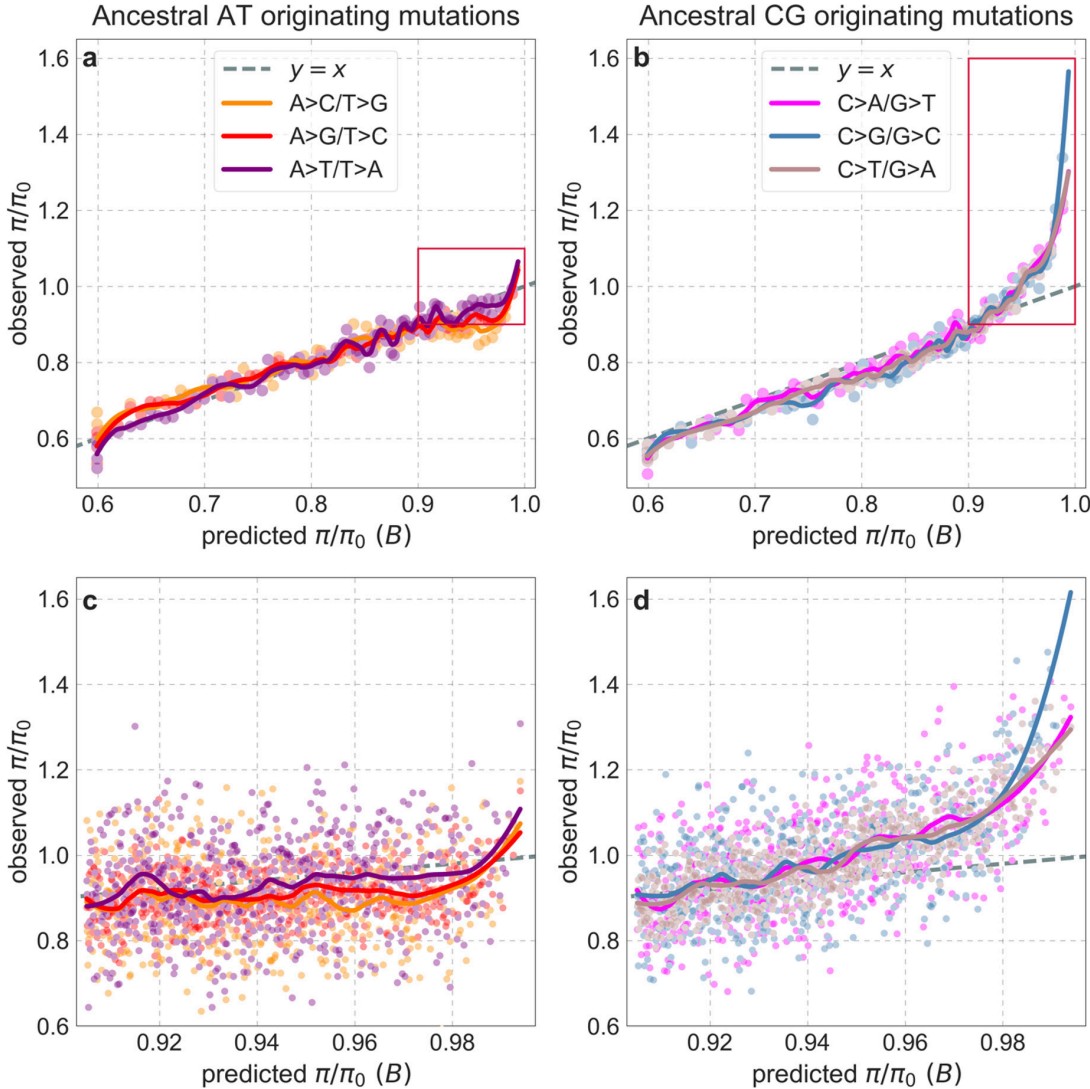

**Appendix 1—figure 44.** Observed vs. predicted neutral diversity levels for different types of substitutions. Diversity levels are presented as in *Appendix 1—figure 38*, but here, we calculate diversity levels and substitution rates for sites with a given ancestral state, that is, AT (**a and c**) or CG (**b and d**), and for each of the three types of substitutions from the ancestral state, as in *Appendix 1—figure 43*.

archaic introgression in and around genes (*Sankararaman et al., 2014*; *Vernot and Akey, 2014*; *Harris and Nielsen, 2016*; *Juric et al., 2016*; *Sankararaman et al., 2016*). The causes for this purifying selection are still being deliberated (*Harris and Nielsen, 2016*; *Juric et al., 2016*; *Schumer et al., 2018*). One hypothesis is that selection acts against introgressed alleles that are incompatible with the genetic background in modern humans, for example, alleles that are part of Dobzhansky-Muller incompatibilities between archaic hominins and modern humans (*Sankararaman et al., 2014*; *Schumer et al., 2018*). Another hypothesis is that selection acts against alleles that were deleterious in both archaic hominins and modern humans, which were more common in archaic hominins because their long-term effective population sizes were smaller than in modern humans (*Harris and Nielsen, 2016*; *Juric et al., 2016*; *Steinrücken et al., 2018*).

Regardless of its cause, we expect purifying selection to remove archaic alleles, including neutral variants, more rapidly in genomic regions under stronger background selection (*Harris and Nielsen, 2016*; *Juric et al., 2016*; *Schumer et al., 2018*). This is because these regions harbor more selected sites (*Appendix 1—figure 39b*) in which archaic alleles could be selected against, and because they have lower rates of recombination (*Appendix 1—figure 39a*) causing selection against archaic alleles to remove larger archaic segments. Conversely, we expect the highest, residual proportion

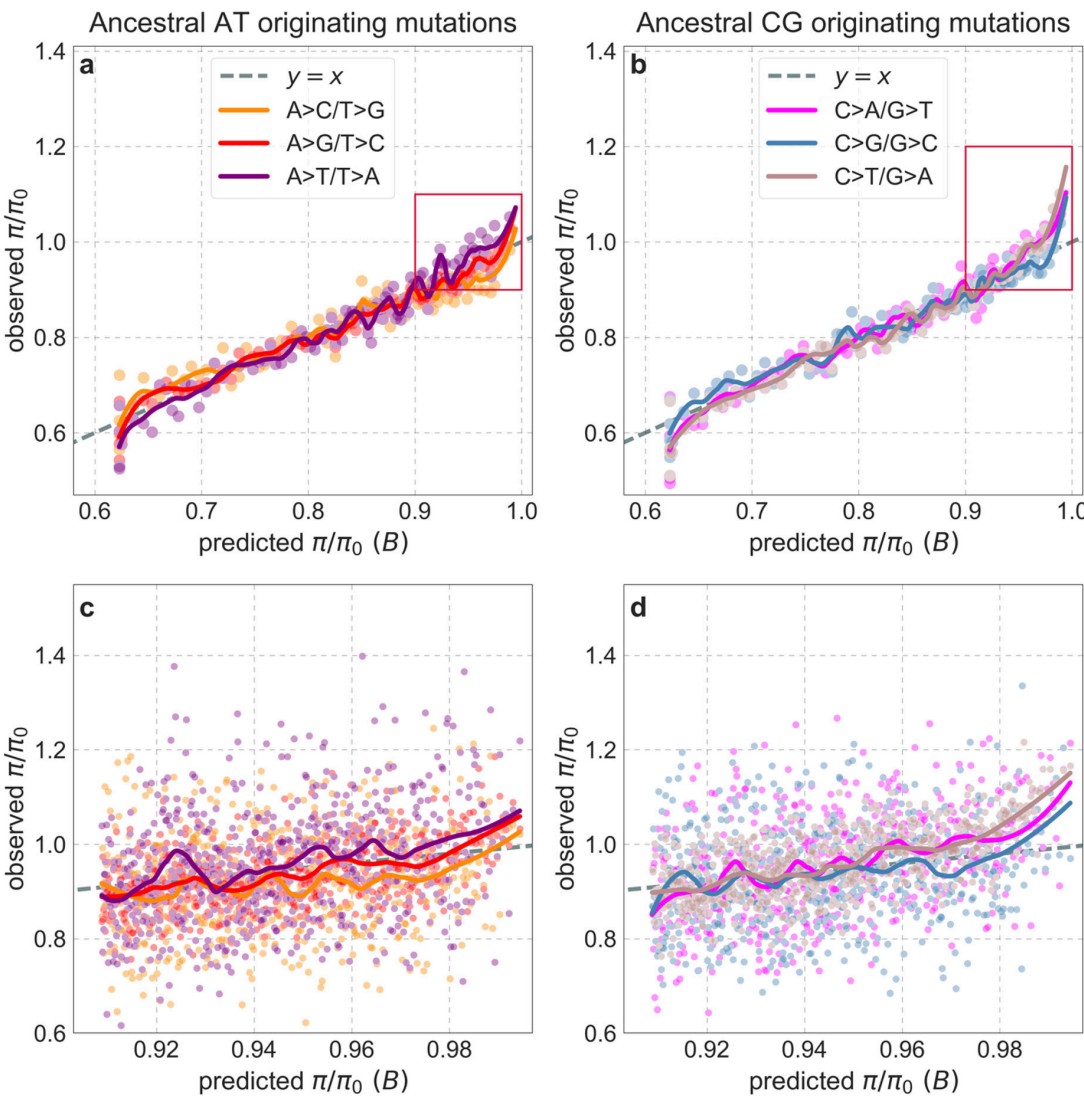

**Appendix 1—figure 45.** Observed vs. predicted neutral diversity levels for different types of substitutions after removing C>G hypermutable regions (from both the inference and observations). Other than removing ~12% of putatively neutral sites in these regions, the details are as in **Appendix 1—figure 44**. We note that while a greater proportion of sites is removed from bins near $B = 1$ (~20% for the 100th percentile), this in itself has a minor effect on the departures from predictions in these bins.

of archaic neutral variants in regions with $B \approx 1$—precisely where we observe a 10–15% excess of diversity above our predictions (**Appendix 1—figure 38**).

In order to test this expectation, we use fine-scale maps of archaic introgression inferred for European (CEU) and East-Asian (CHB/CHS) individuals from the 1000 Genomes Project (**Steinrücken et al., 2018**). These maps assign a probability of Neanderthal ancestry to contiguous 500 bp segments tiling individual genomes based on the high-coverage Altai Neanderthal genome (**Prüfer et al., 2014**). We use them to estimate the average proportion of archaic ancestry per putatively neutral site in bins of predicted $B$. As expected, we find that the estimated proportion of archaic alleles increases with predicted $B$ (**Appendix 1—figure 47**). The power to identify introgressed segments using this and other methods decreases substantially in regions with $B \approx 1$, because higher recombination rate in these regions results in much shorter archaic segments (**Skov et al., 2018**; **Steinrücken et al., 2018**). We therefore expect that the actual proportion of archaic ancestry increases more sharply near $B = 1$ than our analysis suggests, and may therefore better trace the sharp increase in diversity relative to predictions near $B = 1$.

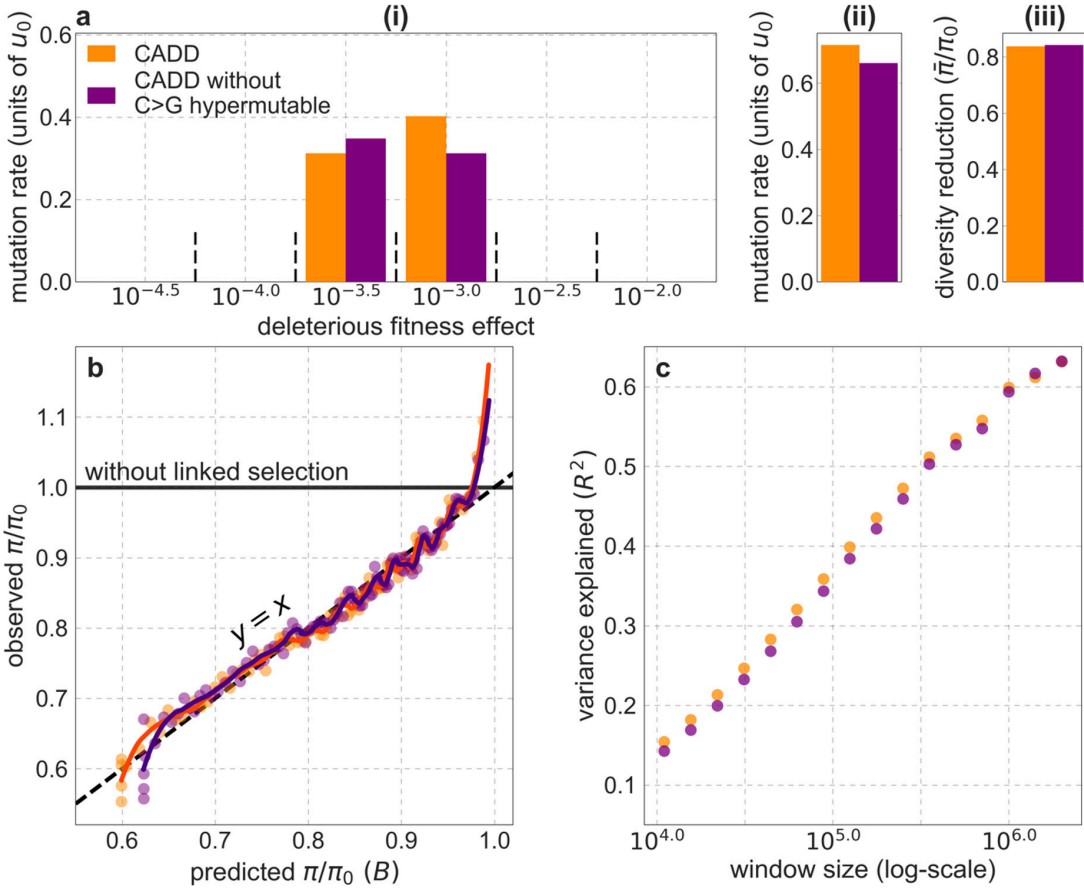

**Appendix 1—figure 46.** Comparison of the best-fitting CADD-based models with and without C>G hypermutable regions. All panels are as described for the corresponding ones in *Appendix 1—figure 14*.

Current inferences about archaic introgression are divided into those that incorporate sequenced Neanderthal and Denisovan genomes (*Green et al., 2010*; *Reich et al., 2010*; *Sankararaman et al., 2014*; *Vernot and Akey, 2014*; *Steinrücken et al., 2018*), such as the maps we used in *Appendix 1—figure 47*, and those that are based only on patterns of variation in

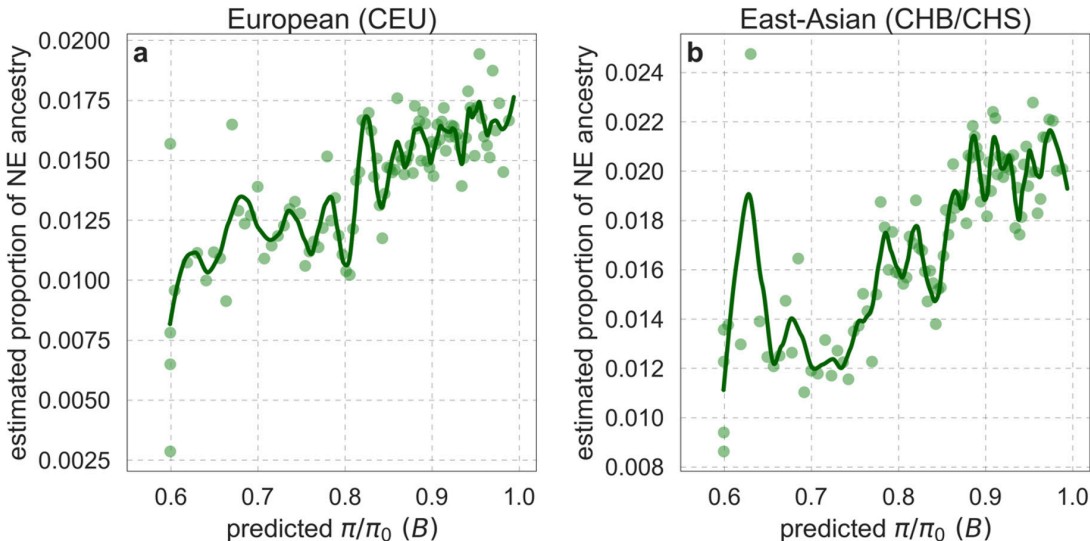

**Appendix 1—figure 47.** Estimated proportion of Neanderthal (NE) ancestry as a function of predicted $B$ in Europeans (CEU) (**a**) and East-Asians (CHB/CHS) (**b**). See text for the estimation procedure. The bins and LOESS curves were calculated as in *Appendix 1—figure 38*.

contemporary humans (*Plagnol and Wall, 2006*; *Wall et al., 2009*; *Skov et al., 2018*; *Durvasula and Sankararaman, 2020*). When we repeat the analysis in *Appendix 1—figure 47* using ancestry-maps based on the latter approach in both Africans and non-Africans (*Skov et al., 2018*; *Durvasula and Sankararaman, 2020*), we find that levels of archaic ancestry either increase and level off at intermediate values of *B*, or peak at intermediate values and decrease as *B* approaches 1. We believe that this departure from our expectation reflects a decrease in the power of these methods near *B* = 1 (due to higher rates of recombination), which is greater than the decrease for methods based on sequenced archaic genomes. An additional caveat is that the evidence for the contribution of archaic introgression to the African gene pool is based solely on patterns in contemporary genetic variation (*Plagnol and Wall, 2006*; *Wall and Hammer, 2006*; *Wall et al., 2009*; *Durvasula and Sankararaman, 2020*) and remain more speculative in lieu of more direct evidence. Thus, while it seems plausible that the greater retention of neutral archaic variants in regions with the highest ~2% of predicted *B* values contributes substantially to the departure from our predictions in both African and non-African populations, at present, the evidence for such a contribution remains equivocal.

## 9. Additional figures

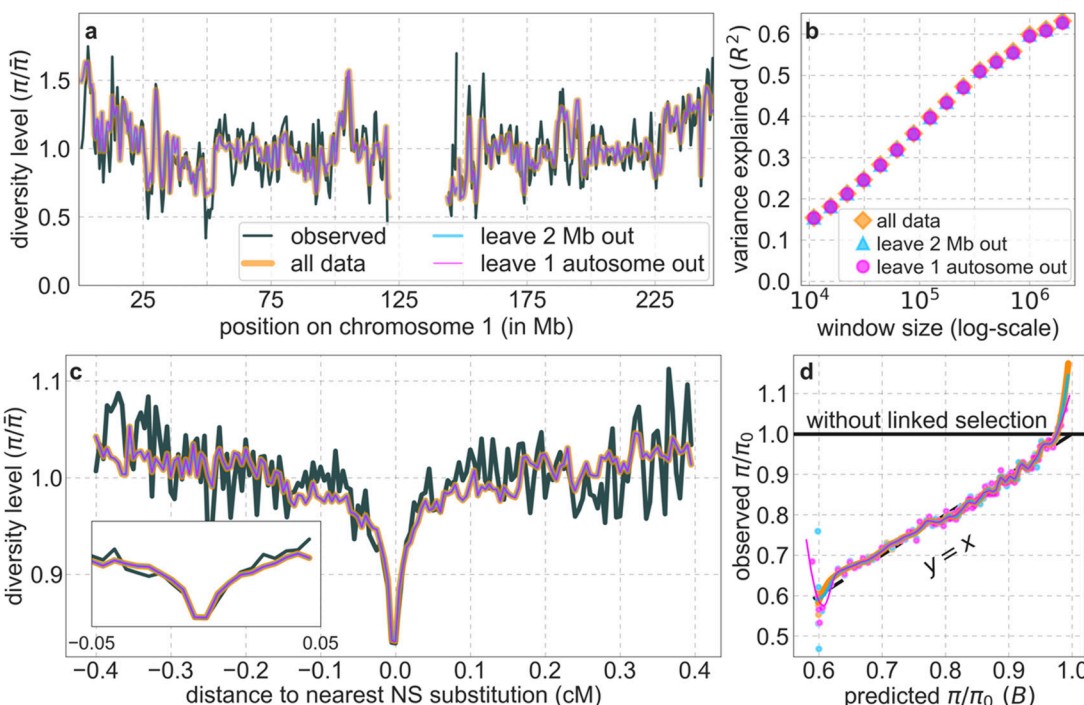

**Appendix 1—figure 48.** Overfitting has negligible effects on our results (see also Section 6.1). As an illustration, we compare the results corresponding to our best-fitting CADD-based model, using all the data jointly (orange), out-of-sample predictions in non-overlapping, contiguous, 2 Mb windows (light blue) and out-of-sample predictions for each autosome (pink). (**a**) Predicted and observed diversity levels along chromosome 1 in the YRI sample; (details as in *Figure 2A* in Main Text). (**b**) The proportion of variance in YRI diversity levels explained by background selection at different spatial scales (details as in *Figure 2B* in Main Text). (**c**) Predicted and observed neutral diversity levels around human-specific nonsynonymous (NS) substitutions (details as in *Figure 3* in Main Text). (**d**) Observed vs. predicted neutral diversity levels across the autosomes (details as in *Figure 5* in Main Text). As an example, the explained variance in diversity levels over 1 Mb windows is 59.9% without excluding data, 59.8% when we exclude 2 Mb windows, and 59.5% when we leave out one chromosome at a time. Our statistical analysis in Section 6.2 suggests that these minute difference are not significant. Moreover, the reduction in explained variance in the case in which we exclude one autosome at a time is plausibly due to the reduced amount of data rather than overfitting (e.g. chromosomes 1 and 2 correspond to 8% and 9.3% of our putatively neutral sites, respectively).

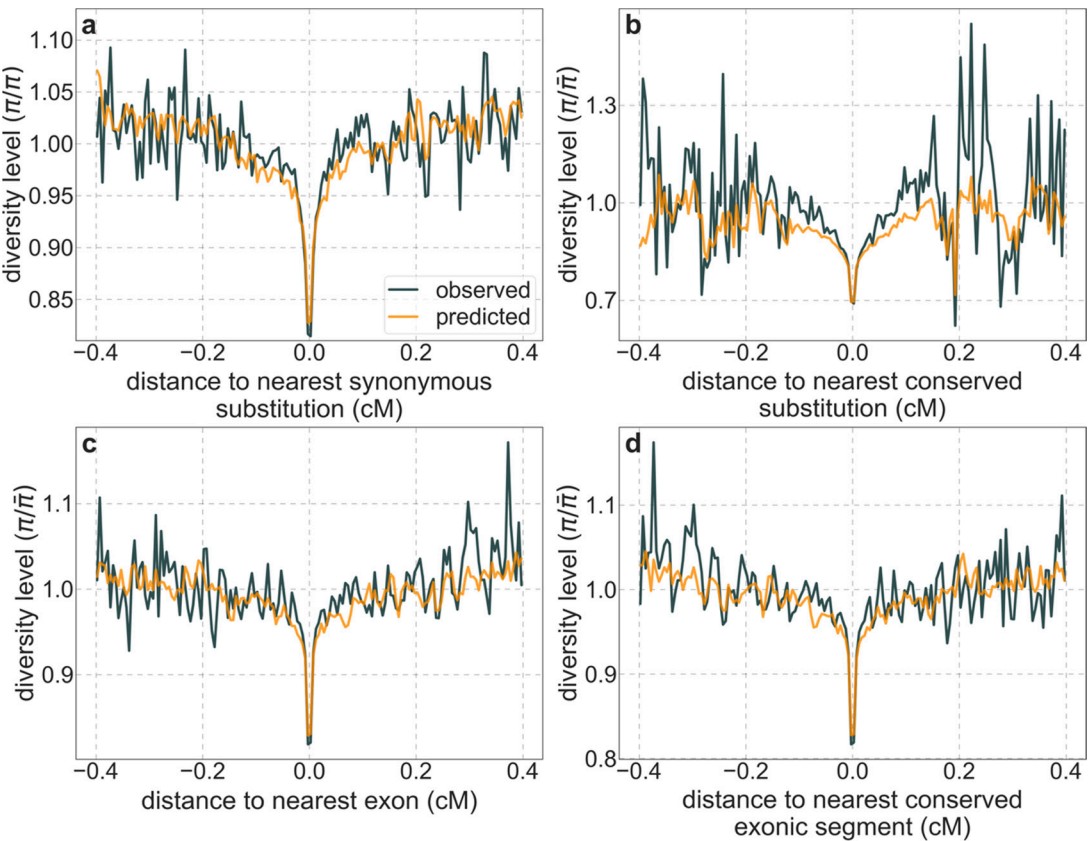

**Appendix 1—figure 49.** A background selection model predicts neutral diversity levels around different genomic features. Here we use our best-fitting CADD-based model and show diversity levels around: (**a**) human-specific synonymous substitutions; (**b**) human-specific substitutions in conserved regions; (**c**) exons; and (**d**) conserved exonic regions. The inference of human-specific substitutions is described in Section 2.7. Conserved regions are based on autosomal sites with the top 6% phastCons scores in the 99-vertebrate alignment (Section 4.1). The set of exons is described in Section 2.4. The genetic distance to the nearest element (e.g. exon) is measured to its closest edge. Other details are similar to *Figure 3*.

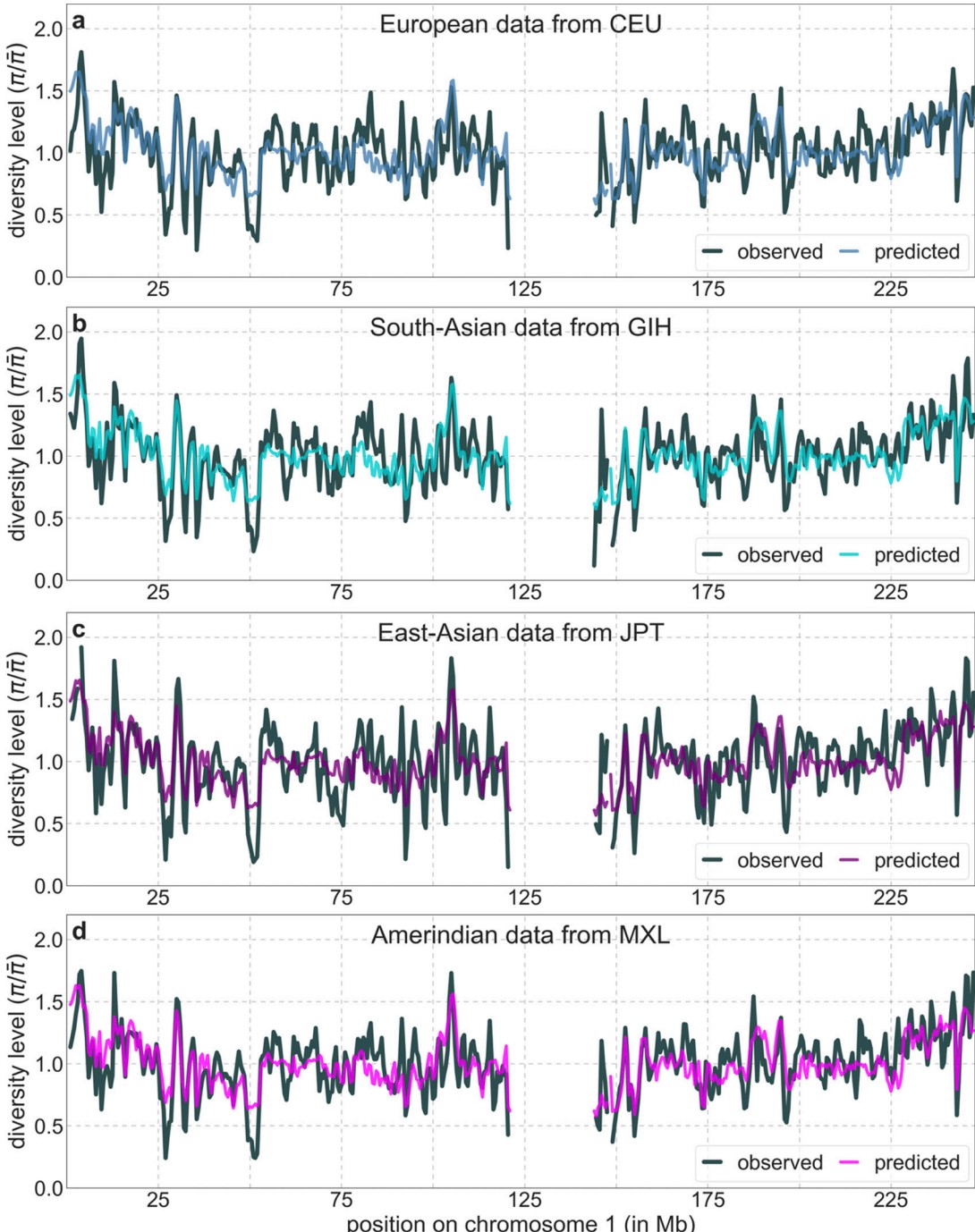

**Appendix 1—figure 50.** Predicted and observed neutral diversity levels along chromosome 1 based on data from representative continental populations. Plots are generated as detailed in *Figure 2A*.

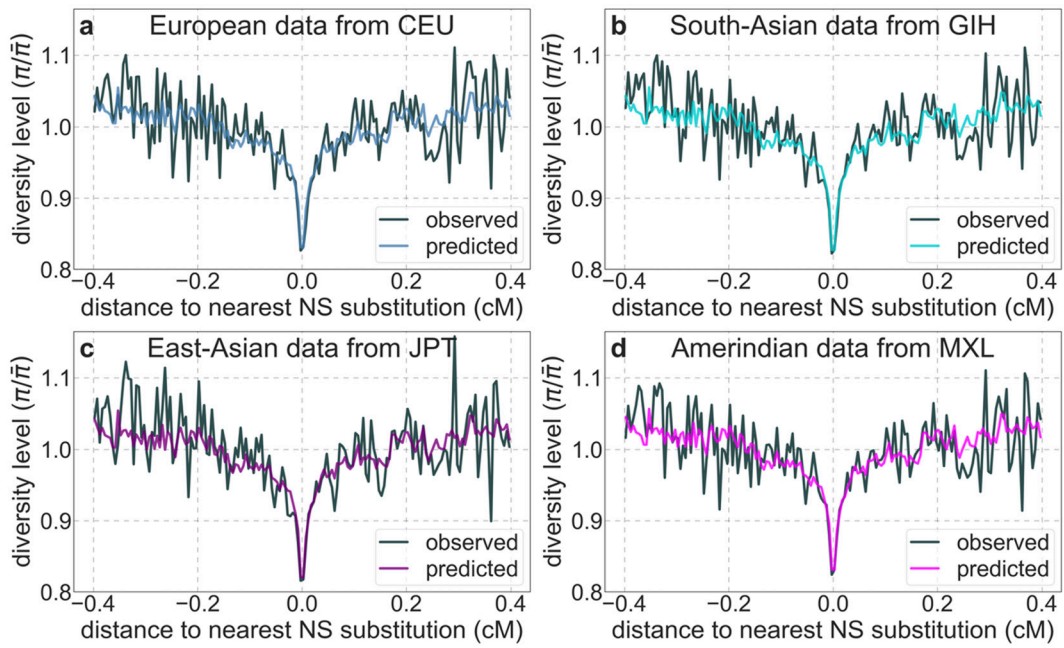

**Appendix 1—figure 51.** A background selection model predicts neutral diversity levels around human-specific nonsynonymous (NS) substitutions in representative continental populations. Plots are constructed as detailed in *Figure 3*, using polymorphism data from each population for both inferences and observations.

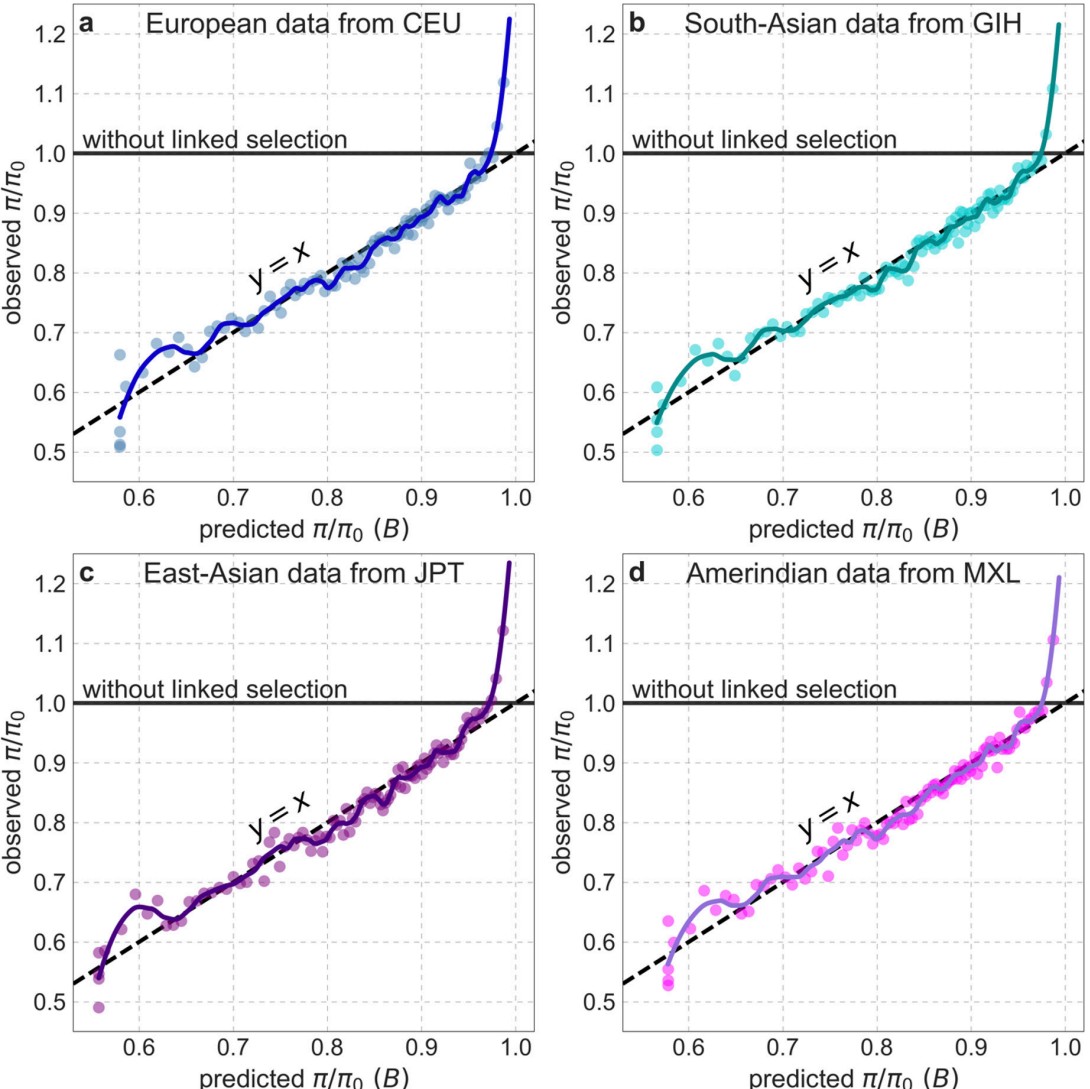

**Appendix 1—figure 52.** Observed vs. predicted neutral diversity levels across the autosomes in representative continental populations. Plots are constructed as detailed in *Figure 5*, using polymorphism data from each population for both inferences and observations.

