## [Editor Report]

This paper uses state-of-the-art methods and the latest data to answer the question of whether variation in polymorphism levels along the human genome is mostly driven by linked purifying selection or selective sweeps. It makes a very strong case for the former. The paper is exceptionally well written and should be of interest to anyone wishing to understand patterns of polymorphism.

---

## [Decision Letter]

**Decision letter after peer review:**

Thank you for submitting your article "Broad-scale variation in human genetic diversity levels is predicted by purifying selection on coding and non-coding elements" for consideration by *eLife*. Your article has been reviewed by 3 peer reviewers, including Magnus Nordborg as Reviewing Editor and Reviewer #1, and the evaluation has been overseen by Detlef Weigel as the Senior Editor.

The reviewers are unanimous in considering this an exceptionally clear and well-written paper. If published as-is, it would be better than most published papers. Respect!

Nonetheless, it can be improved, like all papers, and the reviews attached provide lots of suggestions along these lines. In addition, there are two things we would like to see done, and that we believe should not be too onerous.

Essential Revisions (both points are well-described by Reviewer 3 [indeed] and related to model fitting):

1) A simple neutral simulation to check the extent to which the composite likelihood model can over-fit to neutral variation.

2) An out-of-sample prediction, e.g. by leaving chromosomes out.

*Reviewer #1 (Recommendations for the authors):*

This is an exceptionally well-written paper, and I have no major suggestions for improvement.

*Reviewer #3 (Recommendations for the authors):*

I am reasonably happy with the paper in its current form – I can see the incredible care the authors took to make this study robust. I would recommend some forward simulations using SLiM, but I will not require them (as I said, mostly since this itself would be a sizable task to do). However, I would like to see some simple neutral simulations with realistic recombination maps as a negative control as mentioned. This may seem like an unnecessary negative control – however, my main concern here is that I have heard others critique the Eylashiv et al. model, claiming it could be fitting neutral "noise" along the chromosome. I do not think this is the case, as I think the functional form doesn't have that many degrees of freedom, but I would still like to see this concern addressed.

My primary concern is that the out-sample prediction comparison analysis is conducted (beyond the current leave one window out approach) and included in the main text. I think this would strengthen the claim that BGS models alone closely match diversity data (and the present results are not just a consequence of overfitting). I don't think these out-sample models need to include the positive selection model since it has little effect.

---

## [Author Response]

Essential Revisions (both points are well-described by Reviewer 3 [indeed] and related to model fitting):1) A simple neutral simulation to check the extent to which the composite likelihood model can over-fit to neutral variation.

We do not expect over-fitting to be an issue, for the reasons detailed in our reply to Reviewer 3. Nonetheless, we ran simulations detailed below, and confirmed that assumption: our inference explains almost no variation in diversity levels and infers approximately zero effects of background selection when applied to data simulated under neutrality.

Specifically, we performed the inference on simulated data as follows:

– First, we used *msprime* (Baumdicker et al., 2022) to simulate a neutral diversity dataset that mimics the data we used in our inferences. Specifically, we simulated diversity on all autosomes in a sample size of 216 chromosomes (akin to the sample from 108 individuals in YRI we used), assuming *N_e_* of 20,000, the genetic map from Hinch et al. (2011), and a mutation rate of 1.25·10^-8^ per bp per generation.

– Second, we ran the inference corresponding to our best fitting CADD model, i.e., using the 6% of sites with the highest CADD scores as targets of selection.

We describe the results in the same terms that we used in the manuscript.

**Author response image 1. sa2fig1:** 

Author response image 1 is analogous to Figure 2. As expected, we infer practically no effects of background selection, and predicted diversity levels along chromosome 1 are flat (A: orange curve). The explained variance in diversity levels is indistinguishable from 0 over different spatial scales (B: orange circles), where for comparison, we also included the corresponding results based on the real data. For example, on the 1 Mb scale, the predicted map explains ~0.2% of the variance compared to ~60% for the map based on the same model with real data.

Author response image 2 are analogous to Figures 3 and 5. As expected, diversity levels around nonsynonymous substitutions are nearly flat (on the left); the tiny dip at the center arises inferring a tiny rather than a strictly 0 rate of deleterious mutations at selected regions. The difference between this deleterious mutation rate and 0 is plausibly not statistically significant: ~2.7·10^-10^ per bp per generation at selected regions compared to ~10^-8^ for the same model with real data. The relationship between observed and predicted diversity levels (on the right) reflects noise within a narrow range of (0.985, 1.0) compared to calibrated predications in a range of (0.6, 1.0) for the real data (see Figure 5 in the paper).

2) An out-of-sample prediction, e.g. by leaving chromosomes out.

Following this suggestion, we replaced all the results of the main text for our best-fitting background selection models with maps based on out-of-sample prediction. Specifically, we tile autosomes with non-overlapping 2 Mb windows, where the predicted diversity level in a window is inferred while excluding the data from that window. The 2 Mb scale is substantially greater than the scale of LD blocks in human populations, so any correlation among tMRCAs at the edges of windows should have at most minute effects (see, e.g., Wall and Pritchard NRG 2003). We also include the main analyses based on leaving one chromosome out. The results of these analyses are shown together in the newly added Figure A48. With either approach, the results are indistinguishable from the results we obtained beforehand, as expected given the few parameters that we infer relative to the enormous amount of data (see reply to reviewer 3 on p. 10-11). As an example, the explained variance in diversity levels over 1 Mb windows using our best-fitting CADD-based model is 59.9% without excluding data; 59.8% when we exclude 2 Mb windows; and 59.5% when we leave out one chromosome at a time. Our statistical analysis in Appendix 6 Section 6.2 suggests that these tiny differences are not statistically significant. Moreover, the reduction in explained variance when one chromosome is excluded at a time is plausibly due to the reduction in the amount of data used rather than overfitting (e.g., chromosomes 1 and 2 correspond to 8% and 9.3% of our putatively neutral sites, respectively).

The results of our inference corresponding to leaving out one chromosome at a time are presented as Author response image 3, using analogous figures to those in the main text:

**Author response image 3. sa2fig3:** Fig. 1 equivalent: orange curve (**A**) and points (**B**) correspond to predictions in which data from the focal chromosome was excluded.

**Author response image 4. sa2fig4:** Figures 3 and 5 equivalents.

We hope that these additional analyses alleviate any concerns about overfitting.

Reviewer #3 (Recommendations for the authors):I am reasonably happy with the paper in its current form – I can see the incredible care the authors took to make this study robust. I would recommend some forward simulations using SLiM, but I will not require them (as I said, mostly since this itself would be a sizable task to do). However, I would like to see some simple neutral simulations with realistic recombination maps as a negative control as mentioned. This may seem like an unnecessary negative control – however, my main concern here is that I have heard others critique the Eylashiv et al. model, claiming it could be fitting neutral "noise" along the chromosome. I do not think this is the case, as I think the functional form doesn't have that many degrees of freedom, but I would still like to see this concern addressed.My primary concern is that the out-sample prediction comparison analysis is conducted (beyond the current leave one window out approach) and included in the main text. I think this would strengthen the claim that BGS models alone closely match diversity data (and the present results are not just a consequence of overfitting). I don't think these out-sample models need to include the positive selection model since it has little effect.

The results of the requested neutral simulations and leave-one-out analyses are described above, in the response to essential revisions (pp. 2-3). In brief, and as anticipated by the reviewer, there is no overfitting of neutral diversity patterns or otherwise.

We note further that the concerns regarding both Elyashiv et al., (2016) and the current manuscript, which we were not aware of previously, are puzzling, for a number of reasons:

– First, we are fitting models with very few parameters, e.g., seven parameters in our best fitting CADD and phastCons based models, to highly variable diversity levels along the genome (see, e.g., Figure 2A) and relying on an enormous amount of data (diversity levels estimated from a sample of ~108 individuals at ~653M putatively neutral autosomal sites, spread over ~2600 LD blocks; Berisa and Pickrell, 2016). Based on first principles alone then, it seems hard to imagine how overfitting could account for substantial variance in diversity levels (e.g., ~60% of the variance in diversity levels in 2580 widows of 1 Mb along human autosomes).

– Regardless of the recombination landscape, given a uniform mutation rate, the expected heterozygosity under a neutral model should be constant throughout autosomes. Consequently, our method should infer no linked selection effects—which is indeed what we see in the neutral simulations above.

– Lastly, the models we fit are based on functional forms specific to the effects of linked selection, and they are fit based on the genetic distance to putative targets of selection. There is no reason to assume that these functional forms (alongside the spatial organization of selected regions) would capture neutral effects, and many reasons to suppose otherwise. If anything, it is remarkable that simple models of linked selection, based on rough annotations of selected regions and assuming mutation and selection parameters that are fixed within each annotation, should fit the data as well as they do.

Thus, based on first principle considerations, there is no reason to expect overfitting to be an issue here, and indeed simulations confirm that there is no overfitting. Hopefully, these additions will put the concern to rest.